# Inference for Batched Bandits

**Kelly W. Zhang**
Department of Computer Science
Harvard University
kellywzhang@seas.harvard.edu

**Lucas Janson**
Departments of Statistics
Harvard University
ljanson@fas.harvard.edu

**Susan A. Murphy**
Departments of Statistics and Computer Science
Harvard University
samurphy@fas.harvard.edu

## Abstract

As bandit algorithms are increasingly utilized in scientific studies and industrial applications, there is an associated increasing need for reliable inference methods based on the resulting adaptively-collected data. In this work, we develop methods for inference on data collected in batches using a bandit algorithm. We first prove that the ordinary least squares estimator (OLS), which is asymptotically normal on independently sampled data, is *not* asymptotically normal on data collected using standard bandit algorithms when there is no unique optimal arm. This asymptotic non-normality result implies that the naive assumption that the OLS estimator is approximately normal can lead to Type-1 error inflation and confidence intervals with below-nominal coverage probabilities. Second, we introduce the Batched OLS estimator (BOLS) that we prove is (1) asymptotically normal on data collected from both multi-arm and contextual bandits and (2) robust to non-stationarity in the baseline reward.

## 1   Introduction

Due to their regret minimizing guarantees bandit algorithms have been increasingly used in in real-world sequential decision-making problems, like online advertising [27], mobile health [42], and online education [34]. However, for many real-world problems it is not enough to just minimize regret on a particular problem instance. For example, suppose we have run an online education experiment using a bandit algorithm where we test different types of teaching strategies. When designing a new online course, ideally we could use the data from the previous experiment to inform the design, e.g., under-performing arms could be eliminated or modified. Moreover, to help others designing online courses we would like to be able to publish our findings about how different teaching strategies compare in their performance. This example demonstrates the need for statistical inference methods on bandit data, which allow practitioners to draw generalizable knowledge from the data they have collected (e.g., how much better one teaching strategy is compared to another) for the sake of scientific discovery and informed decision making.

In this work we will focus on methods to construct confidence intervals for the margin—the difference in expected rewards of two bandit arms—from batched bandit data. Rather than constructing high probability confidence intervals, we are interested in constructing confidence intervals by using the asymptotic distribution of estimators to approximate their finite sample distribution. Asymptotic approximation methods for statistical inference has a long history of being successful in science and leads to much narrower confidence intervals than those constructed using high probability bounds. Most statistical inference methods based on asymptotic approximation assume that treatments

are assigned independently [15]. **However, bandit data violates this independence assumption because it is collected *adaptively*, meaning previous actions and rewards inform future action selections.** The non-independence makes statistical inference more challenging, e.g., estimators like the sample mean are often biased on bandit data [32, 37].

Throughout, we focus on the batched bandit setting, in which arms of the bandit are pulled in batches. For our asymptotic analysis we fix the total number of batches, $T$, and allow the arm pulls in each batch, $n$, to go to infinity. *Note that we do not need or expect $n$ to go to infinity for real-world experiments; we use the asymptotic distribution of estimators to approximate their finite-sample distribution when constructing confidence intervals.* We focus on the batched setting because it closely reflects many of the problem settings where bandit algorithms are applied. For example, in many mobile health [42, 23, 28] and online education problems [22, 34] multiple users use apps / take courses simultaneously, so a batch corresponds to the number of unique users the bandit algorithm acts on at once. The batched setting is even common in online recommendations and advertising because it is impractical to update the bandit after every action if many users visit the site simultaneously [38, 36, 12, 26]. In many such experimental settings the length of the study, $T$, cannot be arbitrarily adjusted, e.g., in online education, courses generally cannot be made arbitrarily long, and clinical trials often run for a standard amount of time that depends on the domain science (e.g. the length of mobile health studies is a function of the scientific community's belief in how long it should take for users to form a habit). On the other hand, the number of users, $n$, can in principle grow as large as funding allows.

Additionally, in our batched setting, we assume that the means of the arms can change over time, i.e., from batch to batch, which reflects the temporal non-stationarity that is prevalent in many real world bandit application problems. For example, in online recommendation systems, the click through rate of a given recommendation typically varies over time, e.g., breaking news articles become less popular over time [38, 26]. Online education and mobile health are also highly non-stationary problems because users tend to disengage over time, so the same notification may be much less effective if sent near the end of an experiment than sent near the beginning [9, 21, 7]. Our statistical inference method does not need to assume that the number of stationary time periods in the experiment is large and is robust to temporal non-stationarity from batch to batch.

**The first contribution of this work is proving that on bandit data, rather surprisingly, whether standard estimators are asymptotically normal can depend on whether the margin is zero.** We prove that for common bandit algorithms, the arm selection probabilities only concentrate if there is a unique optimal arm. Thus, for two-arm bandits, the arm selection probabilities do not concentrate when the margin—the difference in the expected rewards between the arms—is zero. We show that this leads the ordinary least squares (OLS) estimator to be asymptotically normal when the margin is non-zero, and asymptotically *not* normal when the margin is zero. *Since the OLS estimator does not converge uniformly (over values of the margin), standard inference methods (normal approximations, bootstrap[1]) can lead to inflated Type-1 error and unreliable confidence intervals on bandit data.*

**The second contribution of this work is introducing the Batched OLS (BOLS) estimator, which can be used for reliable inference—even in non-stationary settings—on data collected with batched bandits.** We prove that, regardless of whether the margin is zero or not, the BOLS estimator for the margin for both multi-arm and contextual bandits is asymptotically normal and thus can be used for both hypothesis testing and obtaining confidence intervals. Moreover, BOLS is also automatically robust to non-stationarity in the rewards and can be used for constructing valid confidence intervals even if there is non-stationarity in the baseline reward, i.e., if the rewards of the arms change from batch to batch, but the margin remains constant. If the margin itself is also non-stationary, BOLS can also be used for constructing simultaneous confidence intervals for the margins for each batch.

## 2  Related Work

**Batched Bandits**  Much work on batched bandits focuses on minimizing regret [33, 10] or identifying the best arm with high probability [2, 18]. The best arm identification literature utilizes high probability confidence bounds to construct confidence intervals for bandit parameters; we will discuss this method in the next section. Note that in contrast to other batched bandit literature that allow batch sizes to be adjusted adaptively [33], here we do not have adaptive control over the batch sizes.

Batched bandits are closely related to multistage adaptive clinical trials, in which between each batch (or stage of the trial) the data collection procedure can be adjusted depending on the outcome of the previous batches. Our Batched OLS estimator is most closely related to "stage-wise" p-values for group sequential trials that are computed on each stage separately [40]. p-value combination tests are commonly used to combine stage-wise p-values, when the sequence of p-values are shown to be independent or *p-clud*, meaning that under the null each p-value has a Uniform(0,1) distribution conditional on past p-values [40]. [29] formally establish the independence of stage-wise p-values for two-stage trials in which there are a countable number of adaptive rules; note that this rules out bandit algorithms with real-valued arm selection probabilities, like Thompson Sampling. [4] establishes the p-clud property for two-stage adaptive clinical trials under the assumption that the distribution of the second stage data is known conditioned on the decision rule and first stage data under the null hypothesis. Neither of these methods are sufficient for obtaining independent p-values for adaptive trials (1) with an arbitrary number of stages, (2) where exact distribution of rewards is unknown, and (3) where the action selection probabilities can be real numbers, like for Thompson Sampling.

**High Probability Confidence Intervals**   High probability confidence intervals provide stronger guarantees than those constructed using asymptotic approximations. In particular, these bounds are guaranteed to hold for finite number of observations and often even hold uniformly over all $n$ and $T$. These types of bounds are used throughout the bandit and reinforcement learning literature to construct confidence intervals for bandit parameters [14, 20], prove regret bounds [1, 25], and provide guarantees regarding best arm identification [16, 17]. The primary drawback of high probability confidence intervals is that they are much more conservative than those constructed using asymptotic approximations. This means that many more observations will be needed to get a confidence interval of the same width or for a statistical test to have the same power when using high probability confidence intervals compared to those constructed using asymptotic approximation. Since the cost of increasing the the number of users in a study can be large, being able to construct narrow—yet reliable—confidence intervals is crucial to many applications.

In our simulations we compare our method to high probability confidence bounds constructed using the self-normalized martingale bound of [1]. This bound is guaranteed to hold on adaptively collected data and is commonly used in the proof of regret bounds for bandit algorithms. We find that all the approaches based on asymptotic approximations (which we discuss next), significantly outperform the statistical test constructed using a self-normalized martingale bound in terms of power. Moreover, despite the weaker guarantees of statistical inference based on asymptotic approximations, they are generally able to provide reliable coverage of confidence intervals and type-1 error control.

**Adaptive Inference based on Asymptotic Approximations**   A common approach in the literature for performing inference on bandit data is to use adaptive weights, which are weights that are a function of the history. An early example of using adaptive weights is that of [31] and [30], who use adaptive weights in estimating the expected reward under the optimal policy when one has access to i.i.d. observational data. They use an Augmented-Inverse-Probability-Weighted estimator with adaptive weights that are a function of the estimated standard deviation of the reward. [30] conjecture that their approach can be adapted to the adaptive sampling case. Subsequently [11] developed the adaptively weighted method for inference on bandit data to produce the Adaptively-Weighted Augmented-Inverse-Probability-Weighted Estimator (AW-AIPW) for data collected via multi-arm bandits. They prove a central limit theorem (CLT) for AW-AIPW when the adaptive weights satisfy certain conditions. Note, however, the AW-AIPW estimator does not have guarantees in non-stationary settings.

Adaptive weights are also used by [6] to form the W-decorrelated estimator, a debiased version of OLS, that is asymptotically normal. In the multi-arm bandit setting, the adaptive weights are a function of the number of times an arm was chosen previously. We found that in the two-arm setting, the W-decorrelated estimator down-weights rewards from later in the study (Appendix F). [5] introduce the Online Debiased Estimator that also has bias guarantees on adaptive data, but in the more challenging high-dimensional linear regression setting. They prove the asymptotic normality of their estimator in the Gaussian autoregressive time series and the two-batch settings. Note that none of these estimation methods have guarantees in non-stationary bandit settings.

[24] provide conditions under which the OLS estimator is asymptotically normal on adaptively collected data. However, as noted in [39, 6, 11], classical inference techniques developed for i.i.d.

data often empirically have inflated Type-1 error on bandit data. In Section 4.1, we discuss the restrictive nature of [24]'s CLT conditions.

# 3   Problem Formulation

**Setup and Notation**   Though our results generalize to $K$-arm, contextual bandits (see Section 5.2), we first focus on the two-arm bandit for expositional simplicity. Suppose there are $T$ timesteps or batches in a study. In each batch $t \in [1\colon T]$, we select $n$ binary actions $\{A_{t,i}\}_{i=1}^n \in \{0,1\}^n$. We then observe independent rewards $\{R_{t,i}\}_{i=1}^n$, one for each action selected. Note that the distribution of these random variables changes with the batch size, $n$. For example, the distribution of the actions one chooses for the 2nd batch, $\{A_{2,i}\}_{i=1}^n$, may change if one has observed $n = 10$ vs. $n = 100$ samples $\{A_{1,i}, R_{1,i}\}_{i=1}^n$ in the first batch. For readability, we omit indexing random variables by $n$, except for the variables $H_{t-1}^{(n)}$ and $\pi_t^{(n)}$, and filtrations like $\mathcal{G}_{t-1}^{(n)}$ to be introduced next.

For each $t \in [1\colon T]$, the bandit selects actions $\{A_{t,i}\}_{i=1}^n \overset{i.i.d.}{\sim} \mathrm{Bernoulli}(\pi_t^{(n)})$ conditional on $H_{t-1}^{(n)} := \{A_{t',i}, R_{t',i}\}_{i=1,t'=1}^{i=n,t'=t-1}$, the history prior to batch $t$. Note, the *action selection probability* $\pi_t^{(n)} := \mathbb{P}(A_{t,i} = 1 | H_{t-1}^{(n)})$ depends on the history $H_{t-1}^{(n)}$. We assume the following conditional mean for rewards:

$$\mathbb{E}\big[R_{t,i} \big| H_{t-1}^{(n)}, A_{t,i}\big] = (1 - A_{t,i})\beta_{t,0} + A_{t,i}\beta_{t,1}. \tag{1}$$

Note in equation (1) we condition on $H_{t-1}^{(n)}$ because the conditional mean of the reward does not depend on prior rewards or actions. Let $\mathbf{X}_{t,i} := [1 - A_{t,i}, A_{t,i}]^\top \in \mathbb{R}^2$; note $\mathbf{X}_{t,i}$ is higher dimensional when we add more arms and/or context variables. We define the errors as $\epsilon_{t,i} := R_{t,i} - (\mathbf{X}_{t,i})^\top \boldsymbol{\beta}_t$. Equation (1) implies that $\{\epsilon_{t,i} : i \in [1\colon n], t \in [1\colon T]\}$ are a martingale difference array with respect to the filtration $\{\mathcal{G}_t^{(n)}\}_{t=1}^T$, where $\mathcal{G}_t^{(n)} := \sigma\big(H_{t-1}^{(n)} \cup \{A_{t,i}\}_{i=1}^n\big)$; thus, $\mathbb{E}[\epsilon_{t,i}|\mathcal{G}_{t-1}^{(n)}] = 0, \forall t, i, n$. The parameters $\boldsymbol{\beta}_t = (\beta_{t,0}, \beta_{t,1})$ can change across batches $t \in [1\colon T]$, which allows for non-stationarity between batches. Assuming that $\boldsymbol{\beta}_t = \boldsymbol{\beta}_{t'}$ for all $t, t' \in [1\colon T]$ simplifies to the stationary mean case.

**Action Selection Probability Constraint (Clipping)**   In order to perform inference on bandit data it is necessary to guarantee that the bandit algorithm explores sufficiently. For example, the CLTs for both the W-decorrelated [6] and the AW-AIPW [11] estimators have conditions that implicitly require that the bandit algorithms cannot sample any given action with probability that goes to zero or one arbitrarily fast. Greater exploration also increases the power of statistical tests regarding the margin [41]. Moreover, if there is non-stationarity in the margin between batches, it is desirable for the bandit algorithm to continue exploring. We explicitly guarantee exploration by constraining the probability that any given action can be sampled (see Definition 1). We allow the action selection probabilities $\pi_t^{(n)}$ to converge to 0 and/or 1 at some rate.

**Definition 1.** *A clipping constraint with rate $f(n)$ means that $\pi_t^{(n)}$ satisfies the following:*

$$\lim_{n \to \infty} \mathbb{P}\big(\pi_t^{(n)} \in [f(n), 1 - f(n)]\big) = 1 \tag{2}$$

# 4   Asymptotic Distribution of the Ordinary Least Squares Estimator

Suppose we are in the stationary case, and we would like to estimate $\boldsymbol{\beta}$. Consider the OLS estimator: $\hat{\boldsymbol{\beta}}^{\mathrm{OLS}} = (\underline{\mathbf{X}}^\top \underline{\mathbf{X}})^{-1} \underline{\mathbf{X}}^\top \mathbf{R}$, where $\underline{\mathbf{X}} := [\mathbf{X}_{1,1}, .., \mathbf{X}_{1,n}, .., \mathbf{X}_{T,1}, .., \mathbf{X}_{T,n}]^\top \in \mathbb{R}^{nT \times 2}$ and $\mathbf{R} := [R_{1,1}, .., R_{1,n}, .., R_{T,1}, .., R_{T,n}]^\top \in \mathbb{R}^{nT}$. Note that $\underline{\mathbf{X}}^\top \underline{\mathbf{X}} = \sum_{t=1}^T \sum_{i=1}^n \mathbf{X}_{t,i} \mathbf{X}_{t,i}^\top$.

## 4.1   Conditions for Asymptotically Normality of the OLS estimator

If $(\mathbf{X}_{t,i}, \epsilon_{t,i})$ are i.i.d., $\mathbb{E}[\epsilon_{t,i}] = 0$, $\mathbb{E}[\epsilon_{t,i}^2] = \sigma^2$, and the first two moments of $\mathbf{X}_{t,i}$ exist, a classical result from statistics [3] is that the OLS estimator is asymptotically normal, i.e., as $n \to \infty$,

$$(\underline{\mathbf{X}}^\top \underline{\mathbf{X}})^{1/2}(\hat{\boldsymbol{\beta}}^{\mathrm{OLS}} - \boldsymbol{\beta}) \overset{D}{\to} \mathcal{N}(\mathbf{0}, \sigma^2 \underline{\boldsymbol{I}}_p).$$

[24] generalize this result by proving that the OLS estimator is still asymptotically normal in the adaptive sampling case when $\underline{\mathbf{X}}^\top \underline{\mathbf{X}}$ satisfies a certain stability condition. To show that a similar result

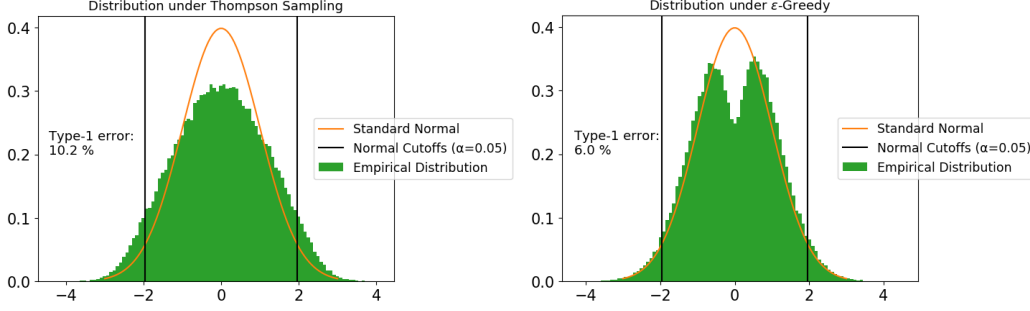

**Figure 1:** Empirical distribution of the Z-statistic ($\sigma^2$ is known) of the OLS estimator for the margin. All simulations are with no margin ($\beta_1 = \beta_0 = 0$); $\mathcal{N}(0,1)$ rewards; $T = 25$; and $n = 100$. For $\epsilon$-greedy, $\epsilon = 0.1$.

holds for the batched setting, we generalize the CLT of [24] to triangular arrays (required since the distribution of our random variables vary as the batch size, $n$, changes), as stated in Theorem 5.

**Condition 1** (Moments). *For all $t, n, i$, $\mathbb{E}\left[\epsilon_{t,i}^2 \big| \mathcal{G}_{t-1}^{(n)}\right] = \sigma^2$ and $\mathbb{E}\left[\epsilon_{t,i}^4 \big| \mathcal{G}_{t-1}^{(n)}\right] < M < \infty$.*

**Condition 2** (Stability). *For some non-random sequence of scalars $\{a_i\}_{i=1}^\infty$, as $n \to \infty$,*

$$a_n \cdot \frac{1}{nT} \sum_{t=1}^T \sum_{i=1}^n A_{t,i} \xrightarrow{P} 1$$

**Theorem 1** (Triangular array version [24], Theorem 3). *Assuming Conditions 1 and 2, as $n \to \infty$,*

$$\left(\underline{\boldsymbol{X}}^\top \underline{\boldsymbol{X}}\right)^{1/2}(\hat{\boldsymbol{\beta}}^{\text{OLS}} - \boldsymbol{\beta}) \xrightarrow{D} \mathcal{N}(0, \sigma^2 \underline{\boldsymbol{I}}_p).$$

Note that in the bandit setting, Condition 2 means that prior to running the experiment, the asymptotic rate at which arms will be selected is predictable. We will show that Condition 2 is in a sense necessary for the asymptotic normality of OLS. In Corollary 1 below we state that Conditions 1 and 3, and a non-zero margin are sufficient for stability Condition 2. Later, we will show that when the margin is zero, Condition 2 does not hold for many common bandit algorithms and prove that this leads the OLS estimator to be asymptotically non-normal.

**Condition 3** (Conditionally i.i.d. actions). *For each $t \in [1\colon T]$, $\{A_{t,i}\}_{i=1}^n \overset{i.i.d.}{\sim} \text{Bernoulli}(\pi_t^{(n)})$ i.i.d. over $i \in [1\colon n]$ conditional on $H_{t-1}^{(n)}$.*

**Corollary 1** (Sufficient conditions for Theorem 5). *If Conditions 1 and 3 hold, and **the margin is non-zero**, data collected in batches using $\epsilon$-greedy, Thompson Sampling, or UCB with clipping constraint with $f(n) = c$ for some $0 < c \leq \frac{1}{2}$ (see Definition 1) satisfy Theorem 5 conditions.*

### 4.2 Asymptotic Non-Normality under No Margin

We prove the conjecture of [6] that when the margin is zero, the OLS estimator is asymptotically non-normal under common bandit algorithms, including Thompson Sampling, $\epsilon$-greedy, and UCB. Thus as seen in Figure 1, assuming the OLS estimator is approximately Normal on bandit data can lead to inflated Type-1 error, even asymptotically. The asymptotic non-normality of OLS occurs when the margin is zero because when there is no unique optimal arm, $\pi_t^{(n)}$ does not concentrate as $n \to \infty$ (Appendix C).

We state the asymptotic non-normality result for Thompson Sampling in Theorem 2; see Appendix C for the proof and similar results for $\epsilon$-greedy and UCB. It is sufficient to prove asymptotic non-normality for $T = 2$. Note, $\hat{\Delta}^{\text{OLS}}$ is the difference in the sample means for each arm, so $\hat{\Delta}^{\text{OLS}} = \hat{\beta}_1^{\text{OLS}} - \hat{\beta}_0^{\text{OLS}}$. The Z-statistic of $\hat{\Delta}^{\text{OLS}}$, which is asymptotically normal under i.i.d. sampling, is as follows:

$$\sqrt{\frac{\sum_{t=1}^2 \sum_{i=1}^n A_{t,i}(\sum_{t=1}^2 \sum_{i=1}^n 1 - A_{t,i})}{2\sigma^2 n}} \left(\hat{\Delta}^{\text{OLS}} - \Delta\right). \tag{3}$$

**Theorem 2** (Asymptotic non-normality of OLS estimator under zero margin for Thompson Sampling). *Let $T = 2$ and $\pi_1^{(n)} = \frac{1}{2}$. If $\epsilon_{t,i} \overset{i.i.d.}{\sim} \mathcal{N}(0,1)$, we have independent normal priors on arm means*

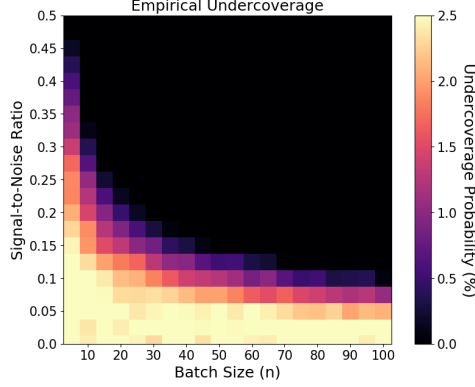

**Figure 2:** Empirical undercoverage probabilities (coverage probability below $95\%$) of confidence intervals using on a normal approximation for the OLS estimator. We use Thompson Sampling with $\mathcal{N}(0,1)$ priors, a clipping constraint of $0.05 \le \pi_t^{(n)} \le 0.95$, $\mathcal{N}(0,1)$ rewards, $T = 25$, and known $\sigma^2$. Standard errors are $< 0.001$.

$\tilde{\beta}_0, \tilde{\beta}_1 \overset{i.i.d.}{\sim} \mathcal{N}(0,1)$, and $\pi_2^{(n)} = \pi_{\min} \vee [(1 - \pi_{\max}) \wedge \mathbb{P}(\tilde{\beta}_1 > \tilde{\beta}_0 | H_1^{(n)})]$ for constants $\pi_{\min}, \pi_{\max}$ with $0 < \pi_{\min} \le \pi_{\max} < 1$, then (3) is asymptotically **not** normal when the margin is zero.

Since the OLS estimator is asymptotically normal when $\Delta \neq 0$ (Corollary 1) and asymptotically *not* Normal when $\Delta = 0$, the OLS estimator does not converge *uniformly* on data collected under standard bandit algorithms. The non-uniform convergence of the OLS estimator precludes us from using a normal approximation to perform hypothesis testing and construct confidence intervals (see [19]). In real-world applications, there is rarely exactly zero margin. However, the non-uniform convergence of the OLS estimator at zero margin is still practically important because the asymptotic distribution of the OLS estimator when the margin is zero is indicative of the finite-sample distribution when the margin is statistically difficult to differentiate from zero, i.e., when the signal-to-noise ratio, $\frac{|\Delta|}{\sigma}$, is low. Figure 2 shows that *even when the margin is non-zero, when the signal-to-noise ratio is low, confidence intervals constructed using a normal approximation have coverage probabilities below the nominal level. Moreover, for any batch size $n$ and noise variance $\sigma^2$, there exists a non-zero margin size with a finite-sample distribution that is poorly approximated by a normal distribution.*

## 5 Batched OLS Estimator

### 5.1 Batched OLS Estimator for Multi-Arm Bandits

We now introduce the Batched OLS (BOLS) estimator that is asymptotically normal under a large class of bandit algorithms, even when the margin is zero. Instead of computing the OLS estimator on all the data, we compute the OLS estimator for each batch and normalize it by the variance estimated from that batch. For each $t \in [1\!:\!T]$, the BOLS estimator of the margin $\Delta_t := \beta_{t,1} - \beta_{t,0}$ is:

$$\hat{\Delta}_t^{\text{BOLS}} = \frac{\sum_{i=1}^n (1 - A_{t,i}) R_{t,i}}{\sum_{i=1}^n 1 - A_{t,i}} - \frac{\sum_{i=1}^n A_{t,i} R_{t,i}}{\sum_{i=1}^n A_{t,i}}.$$

**Theorem 3** (Asymptotic Normality of Batched OLS estimator for multi-arm bandits)**.** *Assuming Conditions 1 (moments) and 3 (conditionally i.i.d. actions), and a clipping rate $f(n) = \frac{1}{n^\alpha}$ for some $0 \le \alpha < 1$ (see Definition 1),*

$$\begin{bmatrix} \sqrt{\frac{(\sum_{i=1}^n 1 - A_{1,i})(\sum_{i=1}^n A_{1,i})}{n}} (\hat{\Delta}_1^{\text{BOLS}} - \Delta_1) \\ \sqrt{\frac{(\sum_{i=1}^n 1 - A_{2,i})(\sum_{i=1}^n A_{2,i})}{n}} (\hat{\Delta}_2^{\text{BOLS}} - \Delta_2) \\ \vdots \\ \sqrt{\frac{(\sum_{i=1}^n 1 - A_{T,i})(\sum_{i=1}^n A_{T,i})}{n}} (\hat{\Delta}_T^{\text{BOLS}} - \Delta_T) \end{bmatrix} \overset{D}{\to} \mathcal{N}(0, \sigma^2 \boldsymbol{I}_T)$$

It is straightforward to generalize Theorem 3 to the case that batches are different sizes but the size of the smallest batch goes to infinity and the batch size is independent of the history.

By Theorem 3, for the stationary margin case, we can test $H_0 : \Delta = c$ vs. $H_1 : \Delta \neq c$ with the following statistic, which is asymptotically normal under the null:

$$\frac{1}{\sqrt{T}} \sum_{t=1}^{T} \sqrt{\frac{(\sum_{i=1}^{n} 1 - A_{t,i})(\sum_{i=1}^{n} A_{t,i})}{n\sigma^2}} (\hat{\Delta}_t^{\text{BOLS}} - c). \tag{4}$$

This type of test statistic—a weighted combination of asymptotically independent normals—a special case of the inverse normal p-value combination test, has been used in simple settings in which the studies (e.g., batches) are independent (e.g., when conducting meta-analyses across multiple studies) [26]. Here the ability to use this type of test statistic is novel since, due to the bandit algorithm, the batches are *not* independent. The work here demonstrates asymptotic independence and thus for large $n$ the Z-statistics from each batch should be approximately independently distributed.

The key to proving asymptotic normality for BOLS is that the following ratio converges in probability to one: $\frac{(\sum_{i=1}^{n} 1 - A_{t,i})(\sum_{i=1}^{n} A_{t,i})}{n} \frac{1}{n\pi_t^{(n)}(1-\pi_t^{(n)})} \xrightarrow{P} 1$. Since $\pi_t^{(n)} \in \mathcal{G}_{t-1}^{(n)}$, $\frac{1}{n\pi_t^{(n)}(1-\pi_t^{(n)})}$ is a constant given $\mathcal{G}_{t-1}^{(n)}$. Thus, even if $\pi_t^{(n)}$ does not concentrate, we are still able to apply the martingale CLT [8] to prove asymptotic normality. See Appendix B for more details.

## 5.2 Batched OLS Estimator for Contextual Bandits

For contextual $K$-arm bandits, for any two arms $x, y \in [0: K - 1]$, we can estimate the margin between them $\boldsymbol{\Delta}_{t,x-y} := \boldsymbol{\beta}_{t,x} - \boldsymbol{\beta}_{t,y} \in \mathbb{R}^d$. In each batch, we observe context vectors $\{\mathbf{C}_{t,i}\}_{i=1}^{n}$ for $\mathbf{C}_{t,i} \in \mathbb{R}^d$. We redefine the history $H_{t-1}^{(n)} := \{\mathbf{C}_{t',i}, A_{t',i}, R_{t',i}\}_{i=1, t'=1}^{i=1, t'=t-1}$ and define the filtration $\mathcal{F}_t^{(n)} := \sigma(H_{t-1}^{(n)} \cup \{A_{t,i}, \mathbf{C}_{t,i}\}_{i=1}^{n})$. The action selection probabilities $\boldsymbol{\pi}_t^{(n)}$ are now functions of the context, so $\boldsymbol{\pi}_t^{(n)}(\mathbf{C}_{t,i}) \in [0, 1]^K$ is a vector where the $k^{\text{th}}$ dimension equals $\mathbb{P}(A_{t,i} = k | \mathcal{H}_{t-1}^{(n)}, \mathbf{C}_{t,i})$. We assume the following conditional mean model of the reward: $\mathbb{E}[R_{t,i} | \mathcal{F}_{t-1}^{(n)}] = \sum_{k=0}^{K-1} \mathbb{I}_{(A_{t,i}=k)} \mathbf{C}_{t,i}^\top \boldsymbol{\beta}_{t,k}$ and let $\epsilon_{t,i} := R_{t,i} - \sum_{k=0}^{K-1} \mathbb{I}_{(A_{t,i}=k)} \mathbf{C}_{t,i}^\top \boldsymbol{\beta}_{t,k}$.

**Condition 4** (Conditionally i.i.d. contexts). *For each $t$, $\mathbf{C}_{t,1}, \mathbf{C}_{t,2}, ..., \mathbf{C}_{t,n}$ are i.i.d. and its first two moments, $\boldsymbol{\mu}_t, \underline{\boldsymbol{\Sigma}}_t$, are non-random given $H_{t-1}^{(n)}$, i.e., $\boldsymbol{\mu}_t, \underline{\boldsymbol{\Sigma}}_t \in \sigma(H_{t-1}^{(n)})$.*

**Condition 5** (Bounded context). *$\|\mathbf{C}_{t,i}\|_{\max} \leq u$ for all $i, t, n$ for some constant $u$. Also, the minimum eigenvalue of $\underline{\boldsymbol{\Sigma}}_t$ is lower bounded, i.e., $\lambda_{\min}(\underline{\boldsymbol{\Sigma}}_t) > l > 0$.*

**Definition 2.** *A conditional clipping constraint with rate $f(n)$ means that the action selection probabilities $\boldsymbol{\pi}_t^{(n)} : \mathbb{R}^d \to [0, 1]^K$ satisfy the following:*

$$\mathbb{P}(\forall \boldsymbol{c} \in \mathbb{R}^d, \boldsymbol{\pi}_t^{(n)}(\boldsymbol{c}) \in [f(n), 1 - f(n)]^K) \to 1$$

For each $t \in [1: T]$, we have the OLS estimator for $\boldsymbol{\Delta}_{t,x-y}$: $\hat{\boldsymbol{\Delta}}_t^{\text{OLS}} := [\underline{\mathbf{C}}_{t,x}^{-1} + \underline{\mathbf{C}}_{t,y}^{-1}]^{-1} (\hat{\boldsymbol{\beta}}_{t,x}^{\text{OLS}} - \hat{\boldsymbol{\beta}}_{t,y}^{\text{OLS}})$, where $\underline{\mathbf{C}}_{t,k} := \sum_{i=1}^{n} \mathbb{I}_{A_{t,i}^{(n)}=k} \mathbf{C}_{t,i} \mathbf{C}_{t,i}^\top \in \mathbb{R}^{d \times d}$, $\hat{\boldsymbol{\beta}}_{t,k}^{\text{OLS}} = \underline{\mathbf{C}}_{t,k}^{-1} \sum_{i=1}^{n} \mathbb{I}_{A_{t,i}^{(n)}=k} \mathbf{C}_{t,i} R_{t,i}$.

**Theorem 4** (Asymptotic Normality of Batched OLS estimator for contextual bandits). *Assuming Conditions 1 (moments)[2], 3 (conditionally i.i.d. actions), 4, and 5, and a conditional clipping rate $f(n) = c$ for some $0 \leq c < \frac{1}{2}$ (see Definition 2),*

$$\begin{bmatrix} [\underline{\mathbf{C}}_{1,x}^{-1} + \underline{\mathbf{C}}_{1,y}^{-1}]^{1/2} (\hat{\boldsymbol{\Delta}}_1^{\text{OLS}} - \boldsymbol{\Delta}_{1,x-y}) \\ [\underline{\mathbf{C}}_{2,x}^{-1} + \underline{\mathbf{C}}_{2,y}^{-1}]^{1/2} (\hat{\boldsymbol{\Delta}}_2^{\text{OLS}} - \boldsymbol{\Delta}_{2,x-y}) \\ \vdots \\ [\underline{\mathbf{C}}_{T,x}^{-1} + \underline{\mathbf{C}}_{T,y}^{-1}]^{1/2} (\hat{\boldsymbol{\Delta}}_T^{\text{OLS}} - \boldsymbol{\Delta}_{T,x-y}) \end{bmatrix} \xrightarrow{D} \mathcal{N}(0, \sigma^2 \underline{\boldsymbol{I}}_{Td}).$$

## 5.3 Batched OLS Statistic for Non-Stationary Bandits

Many real-world problems we would like to use bandit algorithms for have non-stationary over time. For example, in online advertising, the effectiveness of an ad may change over time due to exposure

to competing ads and general societal changes that could affect perceptions of an ad. We may believe that the expected reward for a given action may vary over time, but that the margin is constant from batch to batch. In the online advertising setting, this would mean whether one ad is always better than another is stable, but the overall effectiveness of both ads may change over time. In this case, we can simply use the BOLS test statistic described earlier in equation (4) to test $H_0: \Delta = 0$ vs. $H_1: \Delta \neq 0$. Note that the BOLS test statistic for the margin is robust to non-stationarity in the baseline reward without any adjustment. Moreover, in our simulation settings we estimate the variance $\sigma^2$ separately for each batch, which allows for non-stationarity in the variance between batches as well; see Appendix A for variance estimation details and see Section 6 for simulation results. Additionally, in the case that we believe that the margin itself may vary from batch to batch, the BOLS test statistic can also be used to construct confidence regions that contain the true margin $\Delta_t$ for each batch simultaneously; see Appendix A.5 for details.

## 6 Simulation Experiments

**Procedure**   We focus on the two-arm bandit setting and test whether the margin is zero, specifically $H_0: \Delta = 0$ vs. $H_1: \Delta \neq 0$. We perform experiments for when the noise variance $\sigma^2$ is estimated. We assume homoscedastic errors throughout. See Appendix A.4 for more details about how we estimate the noise variance and more details regarding our experimental setup. In Figures 6 and 7, we display results for stationary bandits and in Figure 5 we show results for bandits with non-stationary baseline rewards. See Appendix A.5 for results for bandits with non-stationary margins.

In our simulations, we found that OLS and AW-AIPW have inflated Type-1 error. Since Type-1 control is a hard constraint, solutions with inflated Type-1 error are *infeasible* solutions. In the power plots, we adjust the cutoffs of the estimators to ensure proper Type-1 error control; if an estimator has inflated Type-1 error under the null, in the power simulations we use a critical value estimated using the simulations under the null. Note that it is unfeasible to make these cutoff adjustment for real experiments (unless one found the worst case setting), as there are many nuisance parameters—like the expected rewards for each arm and the noise variance—which can affect cutoff values.

**Results**   Figure 3 shows that for small sample sizes ($nT \lesssim 300$), BOLS has more reliable Type-1 error control than AW-AIPW with variance stabilizing weights. After $nT \geq 500$ samples, AW-AIPW has proper Type-1 error, and by Figure 4 it always has slightly greater power than BOLS in the stationary setting. The W-decorrelated estimator has reliable Type-1 error control, but very low power compared to AW-AIPW and BOLS. Finally the high probability, self-normalized martingale bound of [1], which we use for hypothesis testing, has very low power compared to the asymptotic approximation statistical inference methods.

In Figure 5, we display simulation results for the non-stationary baseline reward setting. *Whereas other estimators have no Type-1 error guarantees, BOLS still has proper Type-1 error control in the non-stationary baseline reward setting. Moreover, BOLS can have much greater power than other estimators when there is non-stationarity in the baseline reward.* Overall, BOLS is favorable over other estimators in small-sample settings or when one wants to be robust to non-stationarity in the baseline reward—at the cost of losing a little power if the environment is stationary.

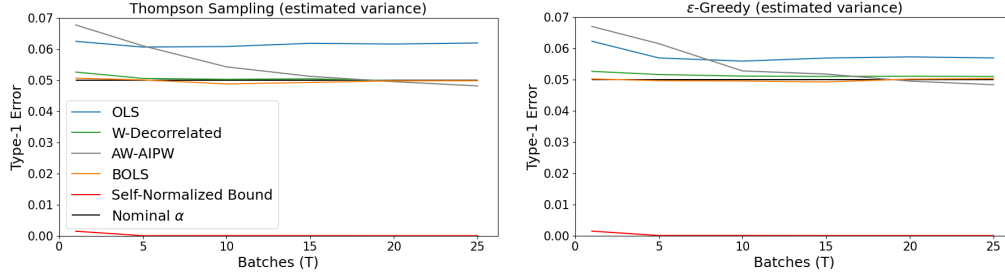

**Figure 3: Stationary Setting:** Type-1 error for a two-sided test of $H_0 : \Delta = 0$ vs. $H_1 : \Delta \neq 0$ ($\alpha = 0.05$). We set $\beta_1 = \beta_0 = 0$, $n = 25$, and a clipping constraint of $0.1 \leq \pi_t^{(n)} \leq 0.9$. We use 100k Monte Carlo simulations and standard errors are $< 0.001$.

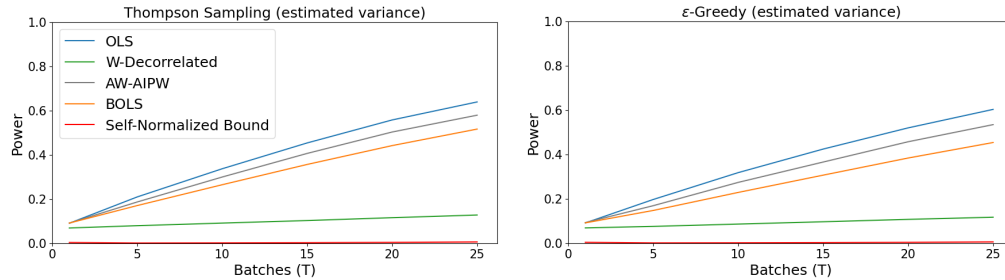

**Figure 4: Stationary Setting:** Power for a two-sided test of $H_0 : \Delta = 0$ vs. $H_1 : \Delta \neq 0$ ($\alpha = 0.05$). We set $\beta_1 = 0$, $\beta_0 = 0.25$, $n = 25$, and a clipping constraint of $0.1 \leq \pi_t^{(n)} \leq 0.9$. We use 100k Monte Carlo simulations and standard errors are $< 0.002$. We account for Type-1 error inflation as described in Section 6.

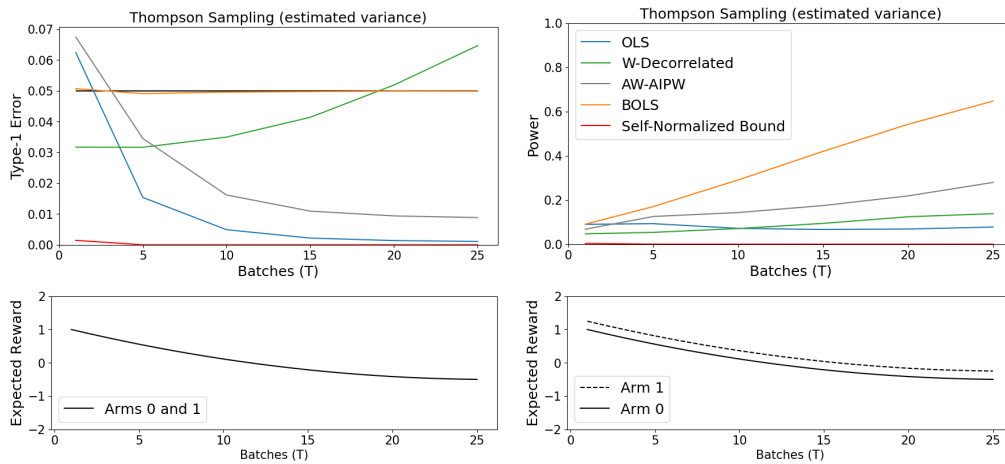

**Figure 5: Non-stationary baseline reward setting:** Type-1 error (upper left) and power (upper right) for a two-sided test of $H_0 : \Delta = 0$ vs. $H_1 : \Delta \neq 0$ ($\alpha = 0.05$). In the lower two plots we plot the expected rewards for each arm; note the margin is constant across batches. We use $n = 25$ and a clipping constraint of $0.1 \leq \pi_t^{(n)} \leq 0.9$. We use 100k Monte Carlo simulations and standard errors are $< 0.002$.

## 7   Discussion

We found that the OLS estimator is asymptotically non-normal when the margin is zero due to the non-concentration of the action selection probabilities. Since the OLS estimator is a canonical example of a method-of-moments estimator [13], our results suggest that the inferential guarantees of standard method-of-moments estimators may fail to hold on adaptively collected data when there is no unique optimal, regret-minimizing policy. We develop the Batched OLS estimator, which is asymptotically normal even when the action selection probabilities do not concentrate. An open question is whether batched versions of general method-of-moments estimators could similarly be used for adaptive inference.

## Broader Impact

Our work has the positive impact of encouraging the use of valid statistical inference methods on bandit data, which ultimately leads to more reliable scientific conclusions. In addition, by providing a valid statistical inference method on bandit data, our work facilitates the use of bandit algorithms in experimentation.

## Acknowledgments and Disclosure of Funding

Research reported in this paper was supported by National Institute on Alcohol Abuse and Alcoholism (NIAAA) of the National Institutes of Health under award number R01AA23187, National Institute on Drug Abuse (NIDA) of the National Institutes of Health under award number P50DA039838, National Cancer Institute (NCI) of the National Institutes of Health under award number U01CA229437, and by NIH/NIBIB and OD award number P41EB028242. The content is solely the responsibility of the authors and does not necessarily represent the official views of the National Institutes of Health

This material is based upon work supported by the National Science Foundation Graduate Research Fellowship Program under Grant No. DGE1745303. Any opinions, findings, and conclusions or recommendations expressed in this material are those of the author(s) and do not necessarily reflect the views of the National Science Foundation.

## Footnotes

[1]Note that the validity of bootstrap methods rely on uniform convergence [35].

[2] Assume an analogous moment condition for the contextual bandit case, where $\mathcal{G}_t^{(n)}$ is replaced by $\mathcal{F}_t^{(n)}$.

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
