[Supplementary Material]

# A  Simulation Details

## A.1  W-Decorrelated Estimator

For the $W$-decorrelated estimator [6], for a batch size of $n$ and for $T$ batches, we set $\lambda$ to be the $\frac{1}{nT}$ quantile of $\lambda_{\min}(\mathbf{X}^\top \mathbf{X})/\log(nT)$, where $\lambda_{\min}(\mathbf{X}^\top \mathbf{X})$ denotes the minimum eigenvalue of $\mathbf{X}^\top \mathbf{X}$. This procedure of choosing $\lambda$ is motivated by the conditions of Theorem 4 of [6] and follows the methods used by [6] in their simulation experiments. We had to adjust the original procedure for choosing $\lambda$ used by [6] (who set $\lambda$ to the $0.15$ quantile of $\lambda_{\min}(\mathbf{X}^\top \mathbf{X})$), because they only evaluated the W-decorrelated method for when the total number of samples was $nT = 1000$ and valid values of $\lambda$ changes with the sample size.

## A.2  AW-AIPW Estimator

Since the AW-AIPW test statistic for the treatment effect is not explicitly written in the original paper [11], we now write the formulas for the AW-AIPW estimator of the treatment effect: $\hat{\Delta}^{\text{AW-AIPW}} := \hat{\beta}_1^{\text{AW-AIPW}} - \hat{\beta}_0^{\text{AW-AIPW}}$. We use the variance stabilizing weights, equal to the square root of the sampling probabilities, $\sqrt{\pi_t^{(n)}}$ and $\sqrt{1 - \pi_t^{(n)}}$. Below, $N_{t,1} = \sum_{i=1}^n A_{t,i}$ and $N_{t,0} = \sum_{i=1}^n (1 - A_{t,i})$.

$$Y_{t,1} := \frac{A_{t,i}}{\pi_t^{(n)}} R_{t,i} + \left(1 - \frac{A_{t,i}^{(n)}}{\pi_t^{(n)}}\right) \frac{\sum_{t'=1}^{t-1} \sum_{i=1}^n A_{t,i}^{(n)} R_{t,i}}{\sum_{t'=1}^{t-1} N_{t',1}}$$

$$Y_{t,0} := \frac{1 - A_{t,i}^{(n)}}{1 - \pi_t^{(n)}} R_{t,i} + \left(1 - \frac{1 - A_{t,i}}{1 - \pi_t^{(n)}}\right) \frac{\sum_{t'=1}^{t-1} \sum_{i=1}^n (1 - A_{t,i}) R_{t,i}}{\sum_{t'=1}^{t-1} N_{t',0}}$$

$$\hat{\beta}_1^{\text{AW-AIPW}} := \frac{\sum_{t=1}^T \sum_{i=1}^n \sqrt{\pi_t^{(n)}} Y_{t,1}}{\sum_{t=1}^T \sum_{i=1}^n \sqrt{\pi_t^{(n)}}} \quad \text{and} \quad \hat{\beta}_0^{\text{AW-AIPW}} := \frac{\sum_{t=1}^T \sum_{i=1}^n \sqrt{1 - \pi_t^{(n)}} Y_{t,0}}{\sum_{t=1}^T \sum_{i=1}^n \sqrt{1 - \pi_t^{(n)}}}$$

The variance estimator for $\hat{\Delta}^{\text{AW-AIPW}}$ is $\hat{V}_0 + \hat{V}_1 + 2\hat{C}_{0,1}$ where

$$\hat{V}_1 := \frac{\sum_{t=1}^T \sum_{i=1}^n \pi_t^{(n)} (Y_{t,1} - \hat{\beta}_1^{\text{AW-AIPW}})^2}{\left(\sum_{t=1}^T \sum_{i=1}^n \sqrt{\pi_t^{(n)}}\right)^2} \quad \text{and} \quad \hat{V}_0 := \frac{\sum_{t=1}^T \sum_{i=1}^n (1 - \pi_t^{(n)}) (Y_{t,0} - \hat{\beta}_0^{\text{AW-AIPW}})^2}{\left(\sum_{t=1}^T \sum_{i=1}^n \sqrt{1 - \pi_t^{(n)}}\right)^2}$$

$$\hat{C}_{0,1} := -\frac{\sum_{t=1}^T \sum_{i=1}^n \sqrt{\pi_t^{(n)} (1 - \pi_t^{(n)})} (Y_{t,1} - \hat{\beta}_1^{\text{AW-AIPW}})(Y_{t,0} - \hat{\beta}_0^{\text{AW-AIPW}})}{\left(\sum_{t=1}^T \sum_{i=1}^n \sqrt{\pi_t^{(n)}}\right)\left(\sum_{t=1}^T \sum_{i=1}^n \sqrt{1 - \pi_t^{(n)}}\right)}$$

## A.3  Self-Normalized Martingale Bound

By the self-normalized martingale bound of [1], specifically Theorem 1 and Lemma 6, we have that in the two arm bandit setting,

$$\mathbb{P}\left(\forall T, n \geq 1, \left|\hat{\beta}_1^{\text{OLS}} - \beta_1\right| \leq c_{1,T} \text{ and } \left|\hat{\beta}_0^{\text{OLS}} - \beta_0\right| \leq c_{0,T}\right) \geq 1 - \delta$$

where

$$c_{a,T} = \sqrt{\sigma^2 \frac{1 + \sum_{t=1}^T N_{t,a}}{\left(\sum_{t=1}^T N_{t,a}\right)^2}\left(1 + 2\log\left(\frac{2\sqrt{1 + \sum_{t=1}^T N_{t,a}}}{\delta}\right)\right)}$$

We estimate $\sigma^2$ using the procedure stated below for the OLS estimator. We reject the null hypothesis that $\Delta = 0$ whenever either the confidence bounds for the two arms are non-overlapping. Specifically when

$$\hat{\beta}_1^{\text{OLS}} + c_{1,T} \leq \hat{\beta}_0^{\text{OLS}} - c_{0,T} \quad \text{or} \quad \hat{\beta}_0^{\text{OLS}} + c_{0,T} \leq \hat{\beta}_1^{\text{OLS}} - c_{1,T}$$

## A.4 Estimating Noise Variance

**OLS Estimator** Given the OLS estimators for the means of each arm, $\hat{\beta}_1^{\text{OLS}}, \hat{\beta}_0^{\text{OLS}}$, we estimate the noise variance $\sigma^2$ as follows:

$$\hat{\sigma}^2 := \frac{1}{nT-2} \sum_{t=1}^{T} \sum_{i=1}^{n} \left( R_{t,i} - A_{t,i}\hat{\beta}_1^{\text{OLS}} - (1 - A_{t,i})\hat{\beta}_0^{\text{OLS}} \right)^2.$$

We use a degrees of freedom bias correction by normalizing by $nT - 2$ rather than $nT$. Since the W-decorrelated estimator is a modified version of the OLS estimator, we also use this same noise variance estimator for the W-decorrelated estimator; we found that this worked well in practice, in terms of Type-1 error control.

**Batched OLS** Given the Batched OLS estimators for the means of each arm for each batch, $\hat{\beta}_{t,1}^{\text{BOLS}}, \hat{\beta}_{t,0}^{\text{BOLS}}$, we estimate the noise variance for each batch $\sigma_t^2$ as follows:

$$\hat{\sigma}_t^2 := \frac{1}{n-2} \sum_{i=1}^{n} \left( R_{t,i} - A_{t,i}\hat{\beta}_{t,1}^{\text{BOLS}} - (1 - A_{t,i})\hat{\beta}_{t,0}^{\text{BOLS}} \right)^2.$$

Again, we use a degrees of freedom bias correction by normalizing by $n - 2$ rather than $n$. We prove the consistency of $\hat{\sigma}_t^2$ (meaning $\hat{\sigma}_t^2 \xrightarrow{P} \sigma^2$) in Corollary 4. Using BOLS to test $H_0 : \Delta = a$ vs. $H_1 : \Delta \neq a$, we use the following test statistic:

$$\frac{1}{\sqrt{T}} \sum_{t=1}^{T} \sqrt{\frac{N_{t,0}N_{t,1}}{n\hat{\sigma}_t^2}} (\hat{\Delta}_t^{\text{BOLS}} - a).$$

Above, $N_{t,1} = \sum_{i=1}^{n} A_{t,i}$ and $N_{t,0} = \sum_{i=1}^{n}(1 - A_{t,i})$. For this test statistic, we use cutoffs based on the Student-t distribution, i.e., for $Y_t \overset{i.i.d.}{\sim} t_{n-2}$ we use a cutoff $c_{\alpha/2}$ such that

$$\mathbb{P}\left( \left| \frac{1}{\sqrt{T}} \sum_{t=1}^{T} Y_t \right| > c_{\alpha/2} \right) = \alpha.$$

We found $c_{\alpha/2}$ by simulating draws from the Student-t distribution.

## A.5 Non-Stationary Treatment Effect

When we believe that the margin itself varies from batch to batch, we are able to construct a confidence region that contains the true margin $\Delta_t$ for each batch simultaneously with probability $1 - \alpha$.

**Corollary 2** (Confidence band for margin for non-stationary bandits). *Assume the same conditions as Theorem 3. Let $z_x$ be $x^{\text{th}}$ quantile of the standard Normal distribution, i.e., for $Z \sim \mathcal{N}(0, 1)$, $\mathbb{P}(Z < z_\alpha) = \alpha$. For each $t \in [1 : T]$, we define the interval*

$$\boldsymbol{L}_t = \hat{\Delta}_t^{\text{OLS}} \pm z_{1-\frac{\alpha}{2T}} \sqrt{\frac{\sigma^2 n}{N_{t,0}N_{t,1}}}.$$

$\lim_{n\to\infty} \mathbb{P}\big(\forall t \in [1 : T], \Delta_t \in \boldsymbol{L}_t\big) \geq 1 - \alpha$. *Above, $N_{t,1} = \sum_{i=1}^{n} A_{t,i}$ and $N_{t,0} = \sum_{i=1}^{n}(1 - A_{t,i})$.*

**Proof:** Note that by Corollary 3,

$$\mathbb{P}\big(\text{exists some } t \in [1 : T] \text{ s.t. } \Delta_t \notin \boldsymbol{L}_t\big) \leq \sum_{t=1}^{T} \mathbb{P}\big(\Delta_t \notin \boldsymbol{L}_t\big) \to \sum_{t=1}^{T} \frac{\alpha}{T} = \alpha$$

where the limit is as $n \to \infty$. Since

$$\mathbb{P}\big(\forall t \in [1 : T], \Delta_t \in \boldsymbol{L}_t\big) = 1 - \mathbb{P}\big(\text{exists some } t \in [1 : T] \text{ s.t. } \Delta_t \notin \boldsymbol{L}_t\big)$$

Thus,

$$\lim_{n\to\infty} \mathbb{P}\big(\forall t \in [1 : T], \Delta_t \in \boldsymbol{L}_t\big) \geq 1 - \alpha \qquad \square$$

We can also test the null hypothesis of no margin against the alternative that at least one batch has non-zero margin, i.e., $H_0\colon \forall t \in [1\colon T], \Delta_t = 0$ vs. $H_1\colon \exists t \in [1\colon T]\ s.t.\ \Delta_t \neq 0$. Note that the global null stated above is of great interest in the mobile health literature [?, ?]. Specifically we use the following test statistic:

$$\sum_{t=1}^{T} \frac{N_{t,0} N_{t,1}}{\sigma^2 n} (\hat{\Delta}_t^{\text{OLS}} - 0)^2,$$

which by Theorem 3 converges in distribution to a chi-squared distribution with $T$ degrees of freedom under the null $\Delta_t = 0$ for all $t$.

To account for estimating noise variance $\sigma^2$, in our simulations for this test statistic, we use cutoffs based on the Student-t distribution, i.e., for $Y_t \overset{i.i.d.}{\sim} t_{n-2}$ we use a cutoff $c_{\alpha/2}$ such that

$$\mathbb{P}\left(\frac{1}{T}\sum_{t=1}^{T} Y_t^2 > c_\alpha\right) = \alpha.$$

We found $c_\alpha$ by simulating draws from the Student-t distribution.

In the plots below we call the test statistic in (A.5) "BOLS Non-Stationary Treatment Effect" (BOLS NSTE). BOLS NSTE performs poorly in terms of power compared to other test statistics in the stationary setting; however, *in the non-stationary setting, BOLS NSTE significantly outperforms all other test statistics, which tend to have low power when the average treatment effect is close to zero.* Note that the W-decorrelated estimator performs well in the left plot of Figure 8; this is because as we show in Appendix F, the W-decorrelated estimator upweights samples from the earlier batches in the study. So when the treatment effect is large in the beginning of the study, the W-decorrelated estimator has high power and when the treatment effect is small or zero in the beginning of the study, the W-decorrelated estimator has low power.

**Figure 6: Stationary Setting:** Type-1 error for a two-sided test of $H_0\colon \Delta = 0$ vs. $H_1\colon \Delta \neq 0$ $(\alpha = 0.05)$. We set $\beta_1 = \beta_0 = 0$, $n = 25$, and a clipping constraint of $0.1 \leq \pi_t^{(n)} \leq 0.9$. We use 100k Monte Carlo simulations and standard errors are $< 0.001$.

**Figure 7: Stationary Setting:** Power for a two-sided test of $H_0\colon \Delta = 0$ vs. $H_1\colon \Delta \neq 0$ $(\alpha = 0.05)$. We set $\beta_1 = 0$, $\beta_0 = 0.25$, $n = 25$, and a clipping constraint of $0.1 \leq \pi_t^{(n)} \leq 0.9$. We use 100k Monte Carlo simulations and standard errors are $< 0.002$. We account for Type-1 error inflation as described in Section 6.

**Figure 8: Nonstationary setting:** The two upper plots display the power of estimators for a two-sided test of $H_0 \colon \forall t \in [1 \colon T],\ \beta_{t,1} - \beta_{t,0} = 0$ vs. $H_1 \colon \exists t \in [1 \colon T], \beta_{t,1} - \beta_{t,0} \neq 0$ ($\alpha = 0.05$). The two lower plots display two treatment effect trends; the left plot considers a decreasing trend (quadratic function) and the right plot considers a oscillating trend (sin function). We set $n = 25$, and a clipping constraint of $0.1 \leq \pi_t^{(n)} \leq 0.9$. We use 100k Monte Carlo simulations and standard errors are $< 0.002$.

# B  Asymptotic Normality of the OLS Estimator

**Condition 6** (Weak moments). $\forall t, n, i$, $\mathbb{E}[\epsilon_{t,i}^2|\mathcal{G}_{t-1}^{(n)}] = \sigma^2$ and for all $\forall t, n, i$, $\mathbb{E}[\varphi(\epsilon_{t,i}^2)|\mathcal{G}_{t-1}^{(n)}] < M < \infty$ a.s. for some function $\varphi$ where $\lim_{x\to\infty} \frac{\varphi(x)}{x} \to \infty$.

**Condition 7** (Stability). *There exists a sequence of nonrandom positive-definite symmetric matrices, $\underline{\boldsymbol{V}}_n$, such that*

(a) $\underline{\boldsymbol{V}}_n^{-1} \big( \sum_{t=1}^T \sum_{i=1}^n \boldsymbol{X}_{t,i}\boldsymbol{X}_{t,i}^\top \big)^{\frac{1}{2}} = \underline{\boldsymbol{V}}_n^{-1}(\underline{\boldsymbol{X}}^\top\underline{\boldsymbol{X}})^{\frac{1}{2}} \xrightarrow{P} \boldsymbol{I}_p$

(b) $\max_{i\in[1:\,n], t\in[1:\,T]} \|\underline{\boldsymbol{V}}_n^{-1}\boldsymbol{X}_{t,i}\|_2 \xrightarrow{P} 0$

**Theorem 5** (Triangular array version of Lai & Wei (1982), Theorem 3). *Let $\boldsymbol{X}_{t,i} \in \mathbb{R}^p$ be non-anticipating with respect to filtration $\{\mathcal{G}_t^{(n)}\}_{t=1}^T$, so $\boldsymbol{X}_{t,i}$ is $\mathcal{G}_{t-1}^{(n)}$ measurable. We assume the following conditional mean model for rewards:*

$$\mathbb{E}\big[R_{t,i}|\mathcal{G}_{t-1}^{(n)}\big] = \boldsymbol{X}_{t,i}^\top\boldsymbol{\beta}.$$

*We define $\epsilon_{t,i} := R_{t,i} - \boldsymbol{X}_{t,i}^\top\boldsymbol{\beta}$. Note that $\{\epsilon_{t,i}\}_{i=1,t=1}^{i=n,t=T}$ is a martingale difference array with respect to filtration $\{\mathcal{G}_t^{(n)}\}_{t=1}^T$.*

*Assuming Conditions 6 and 7, as $n \to \infty$,*

$$(\underline{\boldsymbol{X}}^\top\underline{\boldsymbol{X}})^{1/2}(\hat{\boldsymbol{\beta}}^{\text{OLS}} - \boldsymbol{\beta}) \xrightarrow{D} \mathcal{N}(0, \sigma^2\boldsymbol{I}_p)$$

*Note, in the body of the paper we state that this theorem holds in the two-arm bandit case assuming Conditions 2 and 1. Note that Condition 1 is sufficient for Condition 6 and Condition 2 is sufficient for Condition 7 in the two-arm bandit case.*

**Proof:**

$$\hat{\boldsymbol{\beta}}^{\text{OLS}} = ((\underline{\boldsymbol{X}}^\top\underline{\boldsymbol{X}})^{-1}\underline{\boldsymbol{X}}^{(n),\top}\boldsymbol{R}^{(n)} = (\underline{\boldsymbol{X}}^\top\underline{\boldsymbol{X}})^{-1}\underline{\boldsymbol{X}}^\top(\underline{\boldsymbol{X}}\boldsymbol{\beta} + \boldsymbol{\epsilon})$$

$$\hat{\boldsymbol{\beta}}^{\text{OLS}} - \boldsymbol{\beta} = (\underline{\boldsymbol{X}}^\top\underline{\boldsymbol{X}})^{-1}\underline{\boldsymbol{X}}^\top\boldsymbol{\epsilon} = \left( \sum_{t=1}^T \sum_{i=1}^n \boldsymbol{X}_{t,i}\boldsymbol{X}_{t,i}^\top \right)^{-1} \sum_{t=1}^T \sum_{i=1}^n \boldsymbol{X}_{t,i}\epsilon_{t,i}$$

It is sufficient to show that as $n \to \infty$:

$$(\underline{\boldsymbol{X}}^\top\underline{\boldsymbol{X}})^{-1/2} \sum_{t=1}^T \sum_{i=1}^n \boldsymbol{X}_{t,i}\epsilon_{t,i} \xrightarrow{D} \mathcal{N}(0, \sigma^2\boldsymbol{I}_p)$$

By Slutsky's Theorem and Condition 7 (a), it is also sufficient to show that as $n \to \infty$,

$$\underline{\boldsymbol{V}}_n^{-1} \sum_{t=1}^T \sum_{i=1}^n \boldsymbol{X}_{t,i}\epsilon_{t,i} \xrightarrow{D} \mathcal{N}(0, \sigma^2\underline{\boldsymbol{I}}_p)$$

By Cramer-Wold device, it is sufficient to show multivariate normality if for any fixed $\mathbf{c} \in \mathbb{R}^p$ s.t. $\|\mathbf{c}\|_2 = 1$, as $n \to \infty$,

$$\mathbf{c}^\top\underline{\boldsymbol{V}}_n^{-1} \sum_{t=1}^T \sum_{i=1}^n \boldsymbol{X}_{t,i}\epsilon_{t,i} \xrightarrow{D} \mathcal{N}(0, \sigma^2)$$

We will prove this central limit theorem by using a triangular array martingale central limit theorem, specifically Theorem 2.2 of [8]. We will do this by letting $Y_{t,i} = \mathbf{c}^T\underline{\boldsymbol{V}}_n^{-1}\boldsymbol{X}_{t,i}\epsilon_{t,i}$. The theorem states that as $n \to \infty$, $\sum_{t=1}^T \sum_{i=1}^n Y_{t,i} \xrightarrow{D} \mathcal{N}(0, \sigma^2)$ if the following conditions hold as $n \to \infty$:

(a) $\sum_{t=1}^T \sum_{i=1}^n E[Y_{t,i}|\mathcal{G}_{t-1}^{(n)}] \xrightarrow{P} 0$

(b) $\sum_{t=1}^T \sum_{i=1}^n E[Y_{t,i}^2|\mathcal{G}_{t-1}^{(n)}] \xrightarrow{P} \sigma^2$

(c) $\forall \delta > 0, \sum_{t=1}^T \sum_{i=1}^n E\big[Y_{t,i}^2\mathbb{I}_{(|Y_{t,i}|>\delta)}|\mathcal{G}_{t-1}^{(n)}\big] \xrightarrow{P} 0$

**Useful Properties**   Note that by Cauchy-Schwartz and Condition 7 (b), as $n \to \infty$,

$$\max_{i \in [1:\, n], t \in [1:\, T]} \left| \mathbf{c}^\top \underline{\mathbf{V}}_n^{-1} \mathbf{X}_{t,i} \right| \leq \max_{i \in [1:\, n], t \in [1:\, T]} \|\mathbf{c}\|_2 \|\underline{\mathbf{V}}_n^{-1} \mathbf{X}_{t,i}\|_2 \xrightarrow{P} 0$$

By continuous mapping theorem and since the square function on non-negative inputs is order preserving, as $n \to \infty$,

$$\left( \max_{i \in [1:\, n], t \in [1:\, T]} \left| \mathbf{c}^\top \underline{\mathbf{V}}_n^{-1} \mathbf{X}_{t,i} \right| \right)^2 = \max_{i \in [1:\, n], t \in [1:\, T]} \left( \mathbf{c}^\top \underline{\mathbf{V}}_n^{-1} \mathbf{X}_{t,i} \right)^2 \xrightarrow{P} 0 \tag{5}$$

By Condition 7 (a) and continuous mapping theorem, $\mathbf{c}^\top \underline{\mathbf{V}}_n^{-1} (\mathbf{X}_{t,i}^\top \mathbf{X}_{t,i})^{1/2} \xrightarrow{P} \mathbf{c}^\top$, so

$$\mathbf{c}^\top \underline{\mathbf{V}}_n^{-1} (\mathbf{X}_{t,i}^\top \mathbf{X}_{t,i})^{1/2} (\mathbf{X}_{t,i}^\top \mathbf{X}_{t,i})^{1/2} \underline{\mathbf{V}}_n^{-1} \mathbf{c} \xrightarrow{P} \mathbf{c}^\top \mathbf{c} = 1$$

Thus,

$$\mathbf{c}^\top \underline{\mathbf{V}}_n^{-1} \left( \sum_{t=1}^T \sum_{i=1}^n \mathbf{X}_{t,i} \mathbf{X}_{t,i}^\top \right) \underline{\mathbf{V}}_n^{-1} \mathbf{c} = \sum_{t=1}^T \sum_{i=1}^n \mathbf{c}^\top \underline{\mathbf{V}}_n^{-1} \mathbf{X}_{t,i} \mathbf{X}_{t,i}^\top \underline{\mathbf{V}}_n^{-1} \mathbf{c} \xrightarrow{P} 1$$

Since $\mathbf{c}^\top \underline{\mathbf{V}}_n^{-1} \mathbf{X}_{t,i}$ is a scalar, as $n \to \infty$,

$$\sum_{t=1}^T \sum_{i=1}^n (\mathbf{c}^\top \underline{\mathbf{V}}_n^{-1} \mathbf{X}_{t,i})^2 \xrightarrow{P} 1 \tag{6}$$

**Condition (a): Martingale**

$$\sum_{t=1}^T \sum_{i=1}^n \mathbb{E}[\mathbf{c}^\top \underline{\mathbf{V}}_n^{-1} \mathbf{X}_{t,i} \epsilon_{t,i} | \mathcal{G}_{t-1}^{(n)}] = \sum_{t=1}^T \sum_{i=1}^n \mathbf{c}^\top \underline{\mathbf{V}}_n^{-1} \mathbf{X}_{t,i} \mathbb{E}[\epsilon_{t,i} | \mathcal{G}_{t-1}^{(n)}] = 0$$

**Condition (b): Conditional Variance**

$$\sum_{t=1}^T \sum_{i=1}^n \mathbb{E}[(\mathbf{c}^\top \underline{\mathbf{V}}_n^{-1} \mathbf{X}_{t,i})^2 \epsilon_{t,i}^2 | \mathcal{G}_{t-1}^{(n)}] = \sum_{t=1}^T \sum_{i=1}^n (\mathbf{c}^\top \underline{\mathbf{V}}_n^{-1} \mathbf{X}_{t,i})^2 \mathbb{E}[\epsilon_{t,i}^2 | \mathcal{G}_{t-1}^{(n)}] = \sigma^2 \sum_{t=1}^T \sum_{i=1}^n (\mathbf{c}^\top \underline{\mathbf{V}}_n^{-1} \mathbf{X}_{t,i})^2 \xrightarrow{P} \sigma^2$$

where the last equality holds by Condition 6 and the limit holds by (6) as $n \to \infty$.

**Condition (c): Lindeberg Condition**   Let $\delta > 0$. We want to show that as $n \to \infty$,

$$\sum_{t=1}^T \sum_{i=1}^n Z_{t,i}^2 \mathbb{E}\left[ \epsilon_{t,i}^2 \mathbb{I}_{(Z_{t,i}^2 \epsilon_{t,i}^2 > \delta^2)} \Big| \mathcal{G}_{t-1}^{(n)} \right] \xrightarrow{P} 0$$

where above, we define $Z_{t,i}^{(n)} := \mathbf{c}^\top \underline{\mathbf{V}}_n^{-1} \mathbf{X}_{t,i}$. By Condition 6, we have that for all $n \geq 1$,

$$\max_{t \in [1:\, T], i \in [1:\, n]} \mathbb{E}[\varphi(\epsilon_{t,i}^2) | \mathcal{G}_{-1}^{(n)}] < M$$

Since we assume that $\lim_{x \to \infty} \frac{\varphi(x)}{x} = \infty$, for all $m \geq 1$, there exists a $b_m$ s.t. $\varphi(x) \geq mMx$ for all $x \geq b_m$. So, for all $n, t, i$,

$$M \geq \mathbb{E}[\varphi(\epsilon_{t,i}^2) | \mathcal{G}_{t-1}^{(n)}] \geq \mathbb{E}[\varphi(\epsilon_{t,i}^2) \mathbb{I}_{(\epsilon_{t,i}^2 \geq b_m)} | \mathcal{G}_{t-1}^{(n)}] \geq mM \mathbb{E}[\epsilon_{t,i}^2 \mathbb{I}_{(\epsilon_{t,i}^2 \geq b_m)} | \mathcal{G}_{t-1}^{(n)}]$$

Thus,

$$\max_{t \in [1:\, T], i \in [1:\, n]} \mathbb{E}[\epsilon_{t,i}^2 \mathbb{I}_{(\epsilon_{t,i}^2 \geq b_m)} | \mathcal{G}_{t-1}^{(n)}] \leq \frac{1}{m}$$

So we have that

$$\sum_{t=1}^T \sum_{i=1}^n Z_{t,i}^2 \mathbb{E}\left[ \epsilon_{t,i}^2 \mathbb{I}_{(Z_{t,i}^2 \epsilon_{t,i}^2 > \delta^2)} \Big| \mathcal{G}_{t-1}^{(n)} \right]$$

$$= \sum_{t=1}^{T} \sum_{i=1}^{n} Z_{t,i}^2 \left( \mathbb{E}\left[ \epsilon_{t,i}^2 \mathbb{I}_{(Z_{t,i}^2 \epsilon_{t,i}^2 > \delta^2)} \Big| \mathcal{G}_{t-1}^{(n)} \right] \mathbb{I}_{(Z_{t,i}^2 \leq \delta^2/b_m)} + \mathbb{E}\left[ \epsilon_{t,i}^2 \mathbb{I}_{(Z_{t,i}^2 \epsilon_{t,i}^2 > \delta^2)} \Big| \mathcal{G}_{t-1}^{(n)} \right] \mathbb{I}_{(Z_{t,i}^2 > \delta^2/b_m)} \right)$$

$$\leq \sum_{t=1}^{T} \sum_{i=1}^{n} Z_{t,i}^2 \left( \mathbb{E}\left[ \epsilon_{t,i}^2 \mathbb{I}_{(\epsilon_{t,i}^2 > b_m)} \Big| \mathcal{G}_{t-1}^{(n)} \right] + \sigma^2 \mathbb{I}_{(Z_{t,i}^2 > \delta^2/b_m)} \right)$$

$$\leq \left( \frac{1}{m} + \sigma^2 \mathbb{I}_{(\max_{t' \in [1:\, T], j \in [1:\, n]} Z_{t',j}^2 > \delta^2/b_m)} \right) \sum_{t=1}^{T} \sum_{i=1}^{n} Z_{t,i}^2$$

By Slutsky's Theorem and (6), it is sufficient to show that as $n \to \infty$,

$$\frac{1}{m} + \sigma^2 \mathbb{I}_{(\max_{t' \in [1:\, T], j \in [1:\, n]} Z_{t',j}^2 > \delta^2/b_m)} \xrightarrow{P} 0$$

For any $\epsilon > 0$,

$$\mathbb{P}\left( \frac{1}{m} + \sigma^2 \mathbb{I}_{(\max_{t' \in [1:\, T], j \in [1:\, n]} Z_{t',j}^2 > \delta^2/b_m)} > \epsilon \right) \leq \mathbb{I}_{(\frac{1}{m} > \frac{\epsilon}{2})} + \mathbb{P}\left( \sigma^2 \mathbb{I}_{(\max_{t' \in [1:\, T], j \in [1:\, n]} Z_{t',j}^2 > \delta^2/b_m)} > \frac{\epsilon}{2} \right)$$

We can choose $m$ such that $\frac{1}{m} \leq \frac{\epsilon}{2}$, so $\mathbb{P}\left( \frac{1}{m} > \frac{\epsilon}{2} \right) = 0$. For the second term (note that $m$ is now fixed),

$$\mathbb{P}\left( \sigma^2 \mathbb{I}_{(\max_{t' \in [1:\, T], j \in [1:\, n]} Z_{t',j}^2 > \delta^2/b_m)} > \frac{\epsilon}{2} \right) \leq \mathbb{P}\left( \max_{t' \in [1:\, T], j \in [1:\, n]} Z_{t',j}^2 > \delta^2/b_m \right) \to 0$$

where the last limit holds by (5) as $n \to \infty$. $\quad\square$

## B.1   Corollary 1 (Sufficient conditions for Theorem 5)

*Under Conditions 1 and 3, when **the treatment effect is non-zero** data collected in batches using $\epsilon$-greedy, Thompson Sampling, or UCB with a fixed clipping constraint (see Definition 1) will satisfy Theorem 5 conditions.*

**Proof:**   The only condition of Theorem 5 that needs to verified is Condition 2. To satisfy Condition 2, it is sufficient to show that for any given $\Delta$, for some constant $c \in (0, T)$,

$$\frac{1}{n} \sum_{t=1}^{T} \sum_{i=1}^{n} A_{t,i} = \frac{1}{n} \sum_{t=1}^{T} N_{t,1} \xrightarrow{P} c.$$

$\epsilon$-**greedy** We assume without loss of generality that $\Delta > 0$ and $\pi_1^{(n)} = \frac{1}{2}$. Recall that for $\epsilon$-greedy, for $a \in [2:\, T]$,

$$\pi_a^{(n)} = \begin{cases} 1 - \frac{\epsilon}{2} & \text{if } \frac{\sum_{t=1}^{a} \sum_{i=1}^{n} A_{t,i} R_{t,i}}{\sum_{t'=1}^{a} N_{t',1}} > \frac{\sum_{t=1}^{a} \sum_{i=1}^{n} (1 - A_{t,i}) R_{t,i}}{\sum_{t'=1}^{a} N_{t',0}} \\ \frac{\epsilon}{2} & \text{otherwise} \end{cases}$$

Thus to show that $\pi_a^{(n)} \xrightarrow{P} 1 - \frac{\epsilon}{2}$ for all $a \in [2:\, T]$, it is sufficient to show that

$$\mathbb{P}\left( \frac{\sum_{t=1}^{a} \sum_{i=1}^{n} A_{t,i} R_{t,i}}{\sum_{t'=1}^{a} N_{t',1}} > \frac{\sum_{t=1}^{a} \sum_{i=1}^{n} (1 - A_{t,i}) R_{t,i}}{\sum_{t'=1}^{a} N_{t',0}} \right) \to 1 \tag{7}$$

To show (7), it is equivalent to show that

$$\mathbb{P}\left( \Delta > \frac{\sum_{t=1}^{a} \sum_{i=1}^{n} (1 - A_{t,i}) \epsilon_{t,i}}{\sum_{t'=1}^{a} N_{t',0}} - \frac{\sum_{t=1}^{a} \sum_{i=1}^{n} A_{t,i} \epsilon_{t,i}}{\sum_{t'=1}^{a} N_{t',1}} \right) \to 1 \tag{8}$$

To show (8), it is sufficient to show that

$$\frac{\sum_{t=1}^{a} \sum_{i=1}^{n} (1 - A_{t,i}) \epsilon_{t,i}}{\sum_{t'=1}^{a} N_{t',0}} - \frac{\sum_{t=1}^{a} \sum_{i=1}^{n} A_{t,i} \epsilon_{t,i}}{\sum_{t'=1}^{a} N_{t',1}} \xrightarrow{P} 0. \tag{9}$$

To show (9), it is equivalent to show that

$$\sum_{t=1}^{a} \frac{\sqrt{N_{t,0}}}{\sum_{t'=1}^{a} N_{t',0}} \frac{\sum_{i=1}^{n} (1 - A_{t,i}) \epsilon_{t,i}}{\sqrt{N_{t,0}}} - \sum_{t=1}^{a} \frac{\sqrt{N_{t,1}}}{\sum_{t'=1}^{a} N_{t',1}} \frac{\sum_{i=1}^{n} A_{t,i} \epsilon_{t,i}}{\sqrt{N_{t,1}}} \xrightarrow{P} 0. \tag{10}$$

By Lemma 1, for all $t \in [1:T]$,

$$\frac{N_{t,1}}{\pi_t^{(n)} n} \xrightarrow{P} 1$$

Thus by Slutsky's Theorem, to show (10), it is sufficient to show that

$$\sum_{t=1}^{a} \frac{\sqrt{n(1 - \pi_t^{(n)})}}{n \sum_{t'=1}^{a}(1 - \pi_{t'}^{(n)})} \frac{\sum_{i=1}^{n}(1 - A_{t,i})\epsilon_{t,i}}{\sqrt{N_{t,0}}} - \sum_{t=1}^{a} \frac{\sqrt{n\pi_t^{(n)}}}{n \sum_{t'=1}^{a} \pi_{t'}^{(n)}} \frac{\sum_{i=1}^{n} A_{t,i}\epsilon_{t,i}}{\sqrt{N_{t,1}}} \xrightarrow{P} 0. \quad (11)$$

Since $\pi_t^{(n)} \in [\frac{\epsilon}{2}, 1 - \frac{\epsilon}{2}]$ for all $t, n$, the left hand side of (11) equals the following:

$$\sum_{t=1}^{a} o_p(1) \frac{\sum_{i=1}^{n}(1 - A_{t,i})\epsilon_{t,i}}{\sqrt{N_{t,0}}} - \sum_{t=1}^{a} o_p(1) \frac{\sum_{i=1}^{n} A_{t,i}\epsilon_{t,i}}{\sqrt{N_{t,1}}} \xrightarrow{P} 0.$$

The above limit holds because by Thereom 3, we have that

$$\left( \frac{\sum_{i=1}^{n} A_{1,i}\epsilon_{1,i}}{\sqrt{N_{1,1}}}, \frac{\sum_{i=1}^{n}(1 - A_{1,i})\epsilon_{1,i}}{\sqrt{N_{1,0}}}, ..., \frac{\sum_{i=1}^{n} A_{T,i}\epsilon_{T,i}}{\sqrt{N_{T,1}}}, \frac{\sum_{i=1}^{n}(1 - A_{T,i})\epsilon_{T,i}}{\sqrt{N_{T,0}}} \right) \xrightarrow{D} \mathcal{N}(\mathbf{0}, \sigma^2 \underline{\mathbf{I}}_{2T}). \quad (12)$$

Thus, by Slutsky's Theorem and Lemma 1, we have that

$$\frac{1}{n} \sum_{t=1}^{T} N_{t,1} \xrightarrow{P} \frac{1}{2} + (T - 1)(1 - \frac{\epsilon}{2}) \qquad \text{and} \qquad \frac{1}{n} \sum_{t=1}^{T} N_{t,0} \xrightarrow{P} \frac{1}{2} + (T - 1)\frac{\epsilon}{2}$$

**Thompson Sampling** We assume without loss of generality that $\Delta > 0$ and $\pi_1^{(n)} = \frac{1}{2}$. Recall that for Thompson Sampling with independent standard normal priors ($\tilde{\beta}_1, \tilde{\beta}_0 \overset{i.i.d.}{\sim} \mathcal{N}(0,1)$) for $a \in [2:T]$,

$$\pi_a^{(n)} = \pi_{\min} \vee \left[ \pi_{\max} \wedge \mathbb{P}(\tilde{\beta}_1 > \tilde{\beta}_0 \mid H_{a-1}^{(n)}) \right]$$

Given the independent standard normal priors on $\tilde{\beta}_1, \tilde{\beta}_0$, we have the following posterior distribution:

$$\tilde{\beta}_1 - \tilde{\beta}_0 \mid H_{a-1}^{(n)} \sim \mathcal{N}\left( \frac{\sum_{t=1}^{a-1} \sum_{i=1}^{n} A_{t,i} R_{t,i}}{\sigma^2 + \sum_{t=1}^{a-1} N_{t,1}} - \frac{\sum_{t=1}^{a-1} \sum_{i=1}^{n}(1 - A_{t,i}) R_{t,i}}{\sigma^2 + \sum_{t=1}^{a-1} N_{t,0}}, \right.$$

$$\left. \frac{\sigma^2(\sigma^2 + \sum_{t=1}^{a-1} N_{t,1}) + \sigma^2(\sigma^2 + \sum_{t=1}^{a-1} N_{t,0})}{(\sigma^2 + \sum_{t=1}^{a-1} N_{t,0})(\sigma^2 + \sum_{t=1}^{a-1} N_{t,1})} \right)$$

$$=: \mathcal{N}\left( \mu_{a-1}^{(n)}, (\sigma_{a-1}^{(n)})^2 \right)$$

Thus to show that $\pi_a^{(n)} \xrightarrow{P} \pi_{\max}$ for all $a \in [2:T]$, it is sufficient to show that $\mu_{a-1}^{(n)} \xrightarrow{P} \Delta$ and $(\sigma_{a-1}^{(n)})^2 \xrightarrow{P} 0$ for all $a \in [2:T]$. By Lemma 1, for all $t \in [1:T]$,

$$\frac{N_{t,1}}{\pi_t^{(n)} n} \xrightarrow{P} 1$$

Thus, to show $(\sigma_{a-1}^{(n)})^2 \xrightarrow{P} 0$, it is sufficient to show that

$$\frac{\sigma^2(\sigma^2 + n \sum_{t=1}^{a-1} \pi_t^{(n)}) + \sigma^2(\sigma^2 + n \sum_{t=1}^{a-1}(1 - \pi_t^{(n)}))}{(\sigma^2 + n \sum_{t=1}^{a-1}(1 - \pi_t^{(n)}))(\sigma^2 + n \sum_{t=1}^{a-1} \pi_t^{(n)})} \xrightarrow{P} 0$$

The above limit holds because $\pi_t^{(n)} \in [\pi_{\min}, \pi_{\max}]$ for $0 < \pi_{\min} \le \pi_{\max} < 1$ by the clipping condition.

We now show that $\mu_{a-1}^{(n)} \xrightarrow{P} \Delta$, which is equivalent to showing that the following converges in probability to $\Delta$

$$\frac{\sum_{t=1}^{a-1} \sum_{i=1}^{n} A_{t,i} R_{t,i}}{\sigma^2 + \sum_{t=1}^{a-1} N_{t,1}} - \frac{\sum_{t=1}^{a-1} \sum_{i=1}^{n}(1 - A_{t,i}) R_{t,i}}{\sigma^2 + \sum_{t=1}^{a-1} N_{t,0}}$$

$$= \frac{\sum_{t=1}^{a-1} N_{t,1}}{\sigma^2 + \sum_{t=1}^{a-1} N_{t,1}} \frac{\sum_{t=1}^{a-1} \sum_{i=1}^{n} A_{t,i} R_{t,i}}{\sum_{t=1}^{a-1} N_{t,1}} - \frac{\sum_{t=1}^{a-1} N_{t,0}}{\sigma^2 + \sum_{t=1}^{a-1} N_{t,0}} \frac{\sum_{t=1}^{a-1} \sum_{i=1}^{n} (1 - A_{t,i}) R_{t,i}}{\sum_{t=1}^{a-1} N_{t,0}}$$

$$= \frac{\sum_{t=1}^{a-1} N_{t,1}}{\sigma^2 + \sum_{t=1}^{a-1} N_{t,1}} \left( \beta_1 + \frac{\sum_{t=1}^{a-1} \sum_{i=1}^{n} A_{t,i} \epsilon_{t,i}}{\sum_{t=1}^{a-1} N_{t,1}} \right)$$
$$- \frac{\sum_{t=1}^{a-1} N_{t,0}}{\sigma^2 + \sum_{t=1}^{a-1} N_{t,0}} \left( \beta_0 + \frac{\sum_{t=1}^{a-1} \sum_{i=1}^{n} (1 - A_{t,i}) \epsilon_{t,i}}{\sum_{t=1}^{a-1} N_{t,0}} \right) \quad (13)$$

Note that

$$\frac{\sum_{t=1}^{a-1} N_{t,1}}{\sigma^2 + \sum_{t=1}^{a-1} N_{t,1}} \beta_1 - \frac{\sum_{t=1}^{a-1} N_{t,0}}{\sigma^2 + \sum_{t=1}^{a-1} N_{t,0}} \beta_0 \xrightarrow{P} \Delta \quad (14)$$

Equation (14) above holds by Lemma 1, because

$$\frac{n \sum_{t=1}^{a-1} \pi_t^{(n)}}{\sigma^2 + n \sum_{t=1}^{a-1} \pi_t^{(n)}} \xrightarrow{P} 1 \qquad \frac{n \sum_{t=1}^{a-1} (1 - \pi_t^{(n)})}{\sigma^2 + n \sum_{t=1}^{a-1} (1 - \pi_t^{(n)})} \xrightarrow{P} 1 \quad (15)$$

which hold because $\pi_t^{(n)} \in [\pi_{\min}, \pi_{\max}]$ due to our clipping condition.

By Slutsky's Theorem and (14), to show (13), it is sufficient to show that

$$\frac{\sum_{t=1}^{a-1} \sum_{i=1}^{n} A_{t,i} \epsilon_{t,i}}{\sigma^2 + \sum_{t=1}^{a-1} N_{t,1}} - \frac{\sum_{t=1}^{a-1} \sum_{i=1}^{n} (1 - A_{t,i}) \epsilon_{t,i}}{\sigma^2 + \sum_{t=1}^{a-1} N_{t,0}} \xrightarrow{P} 0. \quad (16)$$

Equation (16) is equivalent to the following:

$$\sum_{t=1}^{a-1} \frac{\sqrt{N_{t,1}}}{\sigma^2 + \sum_{t'=1}^{a-1} N_{t',1}} \frac{\sum_{i=1}^{n} A_{t,i} \epsilon_{t,i}}{\sqrt{N_{t,1}}} - \sum_{t=1}^{a-1} \frac{\sqrt{N_{t,0}}}{\sigma^2 + \sum_{t'=1}^{a-1} N_{t',0}} \frac{\sum_{i=1}^{n} (1 - A_{t,i}) \epsilon_{t,i}}{\sqrt{N_{t,0}}} \xrightarrow{P} 0 \quad (17)$$

By Lemma 1, to show (17) it is sufficient to show that

$$\sum_{t=1}^{a-1} \frac{\sqrt{n \pi_t^{(n)}}}{\sigma^2 + n \sum_{t'=1}^{a-1} \pi_{t'}^{(n)}} \frac{\sum_{i=1}^{n} A_{t,i} \epsilon_{t,i}}{\sqrt{N_{t,1}}} - \sum_{t=1}^{a-1} \frac{\sqrt{n(1 - \pi_t^{(n)})}}{\sigma^2 + n \sum_{t'=1}^{a-1} (1 - \pi_{t'}^{(n)})} \frac{\sum_{i=1}^{n} (1 - A_{t,i}) \epsilon_{t,i}}{\sqrt{N_{t,0}}} \xrightarrow{P} 0 \quad (18)$$

Since $\pi_t^{(n)} \in [\pi_{\min}, \pi_{\max}]$ due to our clipping condition, the left hand side of (18) equals the following

$$\sum_{t=1}^{a-1} o_p(1) \frac{\sum_{i=1}^{n} A_{t,i} \epsilon_{t,i}}{\sqrt{N_{t,1}}} - \sum_{t=1}^{a-1} o_p(1) \frac{\sum_{i=1}^{n} (1 - A_{t,i}) \epsilon_{t,i}}{\sqrt{N_{t,0}}} \xrightarrow{P} 0$$

The above limit holds by (12).

Thus, by Slutsky's Theorem and Lemma 1, we have that

$$\frac{1}{n} \sum_{t=1}^{T} N_{t,1} \xrightarrow{P} \frac{1}{2} + (T - 1)\pi_{\max} \qquad \text{and} \qquad \frac{1}{n} \sum_{t=1}^{T} N_{t,0} \xrightarrow{P} \frac{1}{2} + (T - 1)\pi_{\min} \qquad \square$$

**UCB** We assume without loss of generality that $\Delta > 0$ and $\pi_1^{(n)} = \frac{1}{2}$. Recall that for UCB, for $a \in [2\!:\!T]$,

$$\pi_a^{(n)} = \begin{cases} \pi_{\max} & \text{if } U_{a-1,1} > U_{a-1,0} \\ 1 - \pi_{\max} & \text{otherwise} \end{cases}$$

where we define the upper confidence bounds $U$ for any confidence level $\delta$ with $0 < \delta < 1$ as follows:

$$U_{a-1,1} = \begin{cases} \infty & \text{if } \sum_{t=1}^{a-1} N_{t,1} = 0 \\ \frac{\sum_{t=1}^{a-1} \sum_{i=1}^{n} A_{t,i} R_{t,i}}{\sum_{t=1}^{a-1} N_{t,1}} + \sqrt{\frac{2 \log 1/\delta}{\sum_{t=1}^{a-1} N_{t,1}}} & \text{otherwise} \end{cases}$$

$$U_{a-1,0} = \begin{cases} \infty & \text{if } N_{1,0} = 0 \\ \frac{\sum_{t=1}^{a-1}\sum_{i=1}^{n}(1-A_{t,i})R_{t,i}}{\sum_{t=1}^{a-1} N_{t,0}} + \sqrt{\frac{2\log 1/\delta}{\sum_{t=1}^{a-1} N_{t,0}}} & \text{otherwise} \end{cases}$$

Thus to show that $\pi_a^{(n)} \overset{P}{\to} \pi_{\max}$ for all $a \in [2\colon T]$, it is sufficient to show that $\mathbb{I}_{(U_{a,1}>U_{a,0})} \overset{P}{\to} 1$, which is equivalent to showing that the following converges in probability to 1:

$$\mathbb{I}_{(\sum_{t=1}^{a} N_{t,1}>0, \sum_{t=1}^{a} N_{t,0}>0)} \mathbb{I}_{\left(\frac{\sum_{t=1}^{a}\sum_{i=1}^{n} A_{t,i} R_{t,i}}{\sum_{t=1}^{a} N_{t,1}} + \sqrt{\frac{2\log 1/\delta}{\sum_{t=1}^{a} N_{t,1}}} > \frac{\sum_{t=1}^{a}\sum_{i=1}^{n}(1-A_{t,i})R_{t,i}}{\sum_{t=1}^{a} N_{t,1}} + \sqrt{\frac{2\log 1/\delta}{\sum_{t=1}^{a} N_{t,0}}}\right)}$$
$$+ \mathbb{I}_{(\sum_{t=1}^{a} N_{t,1}=0, \sum_{t=1}^{a} N_{t,0}>0)}$$

$$= \mathbb{I}_{\left((\beta_1-\beta_0)+\frac{\sum_{t=1}^{a}\sum_{i=1}^{n} A_{t,i}\epsilon_{t,i}}{\sum_{t=1}^{a} N_{t,1}} + \sqrt{\frac{2\log 1/\delta}{\sum_{t=1}^{a} N_{t,1}}} > \frac{\sum_{t=1}^{a}\sum_{i=1}^{n}(1-A_{t,i})\epsilon_{t,i}}{\sum_{t=1}^{a} N_{t,1}} + \sqrt{\frac{2\log 1/\delta}{\sum_{t=1}^{a} N_{t,0}}}\right)} + o_p(1)$$

Note that to show that the above converges in probability to 1, it is sufficient to show that the following:

$$\frac{\sum_{t=1}^{a}\sum_{i=1}^{n}(1-A_{t,i})\epsilon_{t,i}}{\sum_{t=1}^{a} N_{t,1}} + \sqrt{\frac{2\log 1/\delta}{\sum_{t=1}^{a} N_{t,0}}} - \frac{\sum_{t=1}^{a}\sum_{i=1}^{n} A_{t,i}\epsilon_{t,i}}{\sum_{t=1}^{a} N_{t,1}} - \sqrt{\frac{2\log 1/\delta}{\sum_{t=1}^{a} N_{t,1}}} \overset{P}{\to} 0$$

Note that for fixed $\delta$, we have that $\frac{2\log 1/\delta}{\sum_{t=1}^{a} N_{t,0}} \overset{P}{\to} 0$, since $\frac{N_{t,0}}{n/2} \overset{P}{\to} 1$. Also note that $\frac{\sum_{t=1}^{a}\sum_{i=1}^{n}(1-A_{t,i})\epsilon_{t,i}}{\sum_{t=1}^{a} N_{t,1}} - \frac{\sum_{t=1}^{a}\sum_{i=1}^{n} A_{t,i}\epsilon_{t,i}}{\sum_{t=1}^{a} N_{t,1}} \overset{P}{\to} 0$, by the same argument made in the $\epsilon$-greedy case to show (9).

Thus, by Slutsky's Theorem and Lemma 1, we have that

$$\frac{1}{n}\sum_{t=1}^{T} N_{t,1} \overset{P}{\to} \frac{1}{2} + (T-1)\pi_{\max} \qquad \text{and} \qquad \frac{1}{n}\sum_{t=1}^{T} N_{t,0} \overset{P}{\to} \frac{1}{2} + (T-1)(1-\pi_{\max})$$

# C  Non-uniform convergence of the OLS Estimator

**Definition 3** (Non-concentration of a sequence of random variables). *For a sequence of random variables $\{Y_i\}_{i=1}^n$ on probability space $(\Omega, \mathcal{F}, \mathbb{P})$, we say $Y_n$ does not concentrate if for each $a \in \mathbb{R}$ there exists an $\epsilon_a > 0$ with*

$$P\big(\{\omega \in \Omega : |Y_n(\omega) - a| > \epsilon_a\}\big) \not\to 0.$$

## C.1  Thompson Sampling

**Proposition 1** (Non-concentration of sampling probabilities under Thompson Sampling). *Under the assumptions of Theorem 2, the posterior distribution that arm 1 is better than arm 0 converges as follows:*

$$\mathbb{P}(\tilde{\beta}_1 > \tilde{\beta}_0 \mid H_1^{(n)}) \xrightarrow{D} \begin{cases} 1 & \text{if } \Delta > 0 \\ 0 & \text{if } \Delta < 0 \\ \text{Uniform}[0,1] & \text{if } \Delta = 0 \end{cases}$$

*Thus, the sampling probabilities $\pi_t^{(n)}$ do not concentrate when $\Delta = 0$.*

**Proof:**  Below, $N_{t,1} = \sum_{i=1}^n A_{t,i}$ and $N_{t,0} = \sum_{i=1}^n (1 - A_{t,i})$. Posterior means:

$$\tilde{\beta}_0 | H_1^{(n)} \sim \mathcal{N}\left( \frac{\sum_{i=1}^n (1 - A_{1,i}) R_{1,i}}{\sigma_a^2 + N_{1,0}}, \frac{\sigma^2}{\sigma_a^2 + N_{0,1}} \right)$$

$$\tilde{\beta}_1 | H_1^{(n)} \sim \mathcal{N}\left( \frac{\sum_{i=1}^n A_{1,i} R_{1,i}}{\sigma_a^2 + N_{1,1}}, \frac{\sigma_a^2}{\sigma_a^2 + N_{1,1}} \right)$$

$$\tilde{\beta}_1 - \tilde{\beta}_0 \mid H_1^{(n)} \sim \mathcal{N}(\mu_n, \sigma_n^2)$$

for $\mu_n := \frac{\sum_{i=1}^n A_{1,i} R_{1,i}}{\sigma_a^2 + N_{1,1}} - \frac{\sum_{i=1}^n (1 - A_{1,i}) R_{1,i}}{\sigma_a^2 + N_{1,0}}$ and $\sigma_n^2 := \frac{\sigma_a^2(\sigma_a^2 + N_{1,1}) + \sigma_a^2(\sigma_a^2 + N_{1,0})}{(\sigma_a^2 + N_{1,0})(\sigma_a^2 + N_{1,1})}$.

$$P(\tilde{\beta}_1 > \tilde{\beta}_0 \mid H_1^{(n)}) = P(\tilde{\beta}_1 - \tilde{\beta}_0 > 0 \mid H_1^{(n)}) = P\left( \frac{\tilde{\beta}_1 - \tilde{\beta}_0 - \mu_n}{\sigma_n} > -\frac{\mu_n}{\sigma_n} \,\bigg|\, H_1^{(n)} \right)$$

For $Z \sim \mathcal{N}(0,1)$ independent of $\mu_n, \sigma_n$.

$$= P\left( Z > -\frac{\mu_n}{\sigma_n} \,\bigg|\, H_1^{(n)} \right) = P\left( Z < \frac{\mu_n}{\sigma_n} \,\bigg|\, H_1^{(n)} \right) = \Phi\left( \frac{\mu_n}{\sigma_n} \,\bigg|\, H_1^{(n)} \right)$$

$$\frac{\mu_n}{\sigma_n} = \left( \frac{\sum_{i=1}^n A_{1,i} R_{1,i}}{\sigma_a^2 + N_{1,1}} - \frac{\sum_{i=1}^n (1 - A_{1,i}) R_{1,i}}{\sigma_a^2 + N_{1,0}} \right) \sqrt{ \frac{(\sigma_a^2 + N_{1,0})(\sigma_a^2 + N_{1,1})}{2\sigma_a^4 + \sigma_a^2 n} }$$

$$= \left( \frac{\beta_1 N_{1,1} + \sum_{i=1}^n A_{1,i} \epsilon_{1,i}}{\sigma_a^2 + N_{1,1}} - \frac{\beta_0 N_{1,0} + \sum_{i=1}^n (1 - A_{1,i}) \epsilon_{1,i}}{\sigma_a^2 + N_{1,0}} \right) \sqrt{ \frac{(\sigma_a^2 + N_{1,0})(\sigma_a^2 + N_{1,1})}{2\sigma_a^4 + \sigma_a^2 n} }$$

$$= \frac{\sum_{i=1}^n A_{1,i} \epsilon_{1,i}}{\sqrt{N_{1,1}}} \sqrt{ \frac{N_{1,1}(\sigma_a^2 + N_{1,0})}{(2\sigma_a^4 + \sigma_a^2 n)(\sigma_a^2 + N_{1,1})} } - \frac{\sum_{i=1}^n (1 - A_{1,i}) \epsilon_{1,i}}{\sqrt{N_{1,0}}} \sqrt{ \frac{N_{1,0}(\sigma_a^2 + N_{1,1})}{(2\sigma_a^4 + \sigma_a^2 n)(\sigma_a^2 + N_{1,0})} }$$

$$+ \left( \beta_1 \frac{N_{1,1}}{\sigma_a^2 + N_{1,1}} - \beta_0 \frac{N_{1,0}}{\sigma_a^2 + N_{1,0}} \right) \sqrt{ \frac{(\sigma_a^2 + N_{1,0})(\sigma_a^2 + N_{1,1})}{2\sigma_a^4 + \sigma_a^2 n} } =: B_n + C_n$$

Let's first examine $C_n$. Note that $\beta_1 = \beta_0 + \Delta$, so $\beta_1 \frac{N_{1,1}}{\sigma_a^2 + N_{1,1}} - \beta_0 \frac{N_{1,0}}{\sigma_a^2 + N_{1,0}}$ equals

$$= (\beta_0 + \Delta) \frac{N_{1,1}}{\sigma_a^2 + N_{1,1}} - \beta_0 \frac{N_{1,0}}{\sigma_a^2 + N_{1,0}} = \Delta \frac{N_{1,1}}{\sigma_a^2 + N_{1,1}} + \beta_0 \left( \frac{N_{1,1}}{\sigma_a^2 + N_{1,1}} - \frac{N_{1,0}}{\sigma_a^2 + N_{1,0}} \right)$$

$$= \Delta \frac{N_{1,1}/n}{(\sigma_a^2 + N_{1,1})/n} + \beta_0 \left( \frac{N_{1,1}(\sigma_a^2 + N_{1,0}) - N_{1,0}(\sigma_a^2 + N_{1,1})}{(\sigma_a^2 + N_{1,1})(\sigma_a^2 + N_{1,1})} \right)$$

$$= \Delta \frac{\frac{1}{2} + o(1)}{\frac{1}{2} + o(1)} + \beta_0 \sigma_a^2 \left( \frac{N_{1,1} - N_{1,0}}{(\sigma_a^2 + N_{1,1})(\sigma_a^2 + N_{1,1})} \right) = \Delta[1 + o(1)] + o\left( \frac{1}{n} \right)$$

where the last equality holds by the Strong Law of Large Numbers because

$$\frac{\frac{1}{n^2}(N_{1,1} - N_{1,0})}{\frac{1}{n^2}(\sigma_a^2 + N_{1,1})(\sigma_a^2 + N_{1,1})} = \frac{\frac{1}{n}[\frac{1}{2} - \frac{1}{2} + o(1)]}{[\frac{1}{2} + o(1)][\frac{1}{2} + o(1)]} = \frac{\frac{1}{n}o(1)}{\frac{1}{4} + o(1)} = o\left( \frac{1}{n} \right)$$

Thus,

$$C_n = \left[ \Delta[1 + o(1)] + o\left( \frac{1}{n} \right) \right] \sqrt{\frac{(\sigma_a^2 + N_{1,0})(\sigma_a^2 + N_{1,1})}{2\sigma_a^4 + \sigma_a^2 n}}$$

$$= \left[ \Delta[1 + o(1)] + o\left( \frac{1}{n} \right) \right] \sqrt{\frac{n[\frac{1}{2} + o(1)][\frac{1}{2} + o(1)]}{o(1) + \sigma_a^2}} = \sqrt{n}\Delta \left[ 1/(2\sigma_a) + o(1) \right] + o\left( \frac{1}{\sqrt{n}} \right)$$

Let's now examine $B_n$.

$$\sqrt{\frac{N_{1,1}(\sigma_a^2 + N_{1,0})}{(2\sigma_a^4 + \sigma_a^2 n)(\sigma_a^2 + N_{1,1})}} = \sqrt{\frac{[\frac{1}{2} + o(1)][\frac{1}{2} + o(1)]}{[\sigma_a^2 + o(1)][\frac{1}{2} + o(1)]}} = \sqrt{\frac{1}{2\sigma_a^2}} + o(1)$$

$$\sqrt{\frac{N_{1,0}(\sigma_a^2 + N_{1,1})}{(2\sigma_a^4 + \sigma_a^2 n)(\sigma_a^2 + N_{1,0})}} = \sqrt{\frac{[\frac{1}{2} + o(1)][\frac{1}{2} + o(1)]}{[\sigma_a^2 + o(1)][\frac{1}{2} + o(1)]}} = \sqrt{\frac{1}{2\sigma_a^2}} + o(1)$$

Note that by Theorem 3, $\left[ \frac{1}{\sqrt{N_{1,1}}} \sum_{i=1}^n \epsilon_{1,i} A_{1,i}, \frac{1}{\sqrt{N_{1,0}}} \sum_{i=1}^n \epsilon_{1,i}(1 - A_{1,i}) \right] \xrightarrow{D} \mathcal{N}(\mathbf{0}, \mathbf{I}_2)$. Thus by Slutky's Theorem,

$$\begin{bmatrix} \frac{\sum_{i=1}^n A_{1,i}\epsilon_{1,i}}{\sqrt{N_{1,1}}} \sqrt{\frac{N_{1,1}(\sigma_a^2 + N_{1,0})}{(2\sigma_a^4 + \sigma_a^2 n)(\sigma_a^2 + N_{1,1})}} \\ \frac{\sum_{i=1}^n (1-A_{1,i})\epsilon_{1,i}}{\sqrt{N_{1,0}}} \sqrt{\frac{N_{1,0}(\sigma_a^2 + N_{1,1})}{(2\sigma_a^4 + \sigma_a^2 n)(\sigma_a^2 + N_{1,0})}} \end{bmatrix} = \begin{bmatrix} \frac{\sum_{i=1}^n A_{1,i}\epsilon_{1,i}}{\sqrt{N_{1,1}}} \left[ \sqrt{\frac{1}{2\sigma_a^2}} + o(1) \right] \\ \frac{\sum_{i=1}^n (1-A_{1,i})\epsilon_{1,i}}{\sqrt{N_{1,0}}} \left[ \sqrt{\frac{1}{2\sigma_a^2}} + o(1) \right] \end{bmatrix} \xrightarrow{D} \mathcal{N}\left( \mathbf{0}, \frac{1}{2\sigma_a^2} \mathbf{I}_2 \right)$$

Thus, we have that, $B_n \xrightarrow{D} \mathcal{N}\left( 0, \frac{1}{\sigma_a^2} \right)$. Since we assume that the algorithm's variance is correctly specified, so $\sigma_a^2 = 1$,

$$B_n + C_n \xrightarrow{D} \begin{cases} \infty & \text{if } \Delta > 0 \\ -\infty & \text{if } \Delta < 0 \\ \mathcal{N}(0, 1) & \text{if } \Delta = 0 \end{cases}$$

Thus, by continuous mapping theorem,

$$\mathbb{P}\big( \tilde{\beta}_1 > \tilde{\beta}_0 | H_1^{(n)} \big) = \Phi\left( \frac{\mu_n}{\sigma_n} \right) = \Phi(B_n + C_n) \xrightarrow{D} \begin{cases} 1 & \text{if } \Delta > 0 \\ 0 & \text{if } \Delta < 0 \quad \square \\ \text{Uniform}[0,1] & \text{if } \Delta = 0 \end{cases}$$

**Proof of Theorem 2 (Non-uniform convergence of the OLS estimator of the treatment effect for Thompson Sampling):** The normalized errors of the OLS estimator for $\Delta$, which are asymptotically normal under i.i.d. sampling are as follows:

$$\sqrt{\frac{(N_{1,1} + N_{2,1})(N_{1,0} + N_{2,0})}{2n}} \left( \hat{\beta}_1^{\text{OLS}} - \hat{\beta}_0^{\text{OLS}} - \Delta \right)$$

$$= \sqrt{\frac{(N_{1,1} + N_{2,1})(N_{1,0} + N_{2,0})}{2n}} \left( \frac{\sum_{t=1}^2 \sum_{i=1}^n A_{t,i}R_{t,i}}{N_{1,1} + N_{2,1}} - \frac{\sum_{t=1}^2 \sum_{i=1}^n (1 - A_{t,i})R_{t,i}}{N_{1,0} + N_{2,0}} - \Delta \right)$$

$$= \sqrt{\frac{(N_{1,1} + N_{2,1})(N_{1,0} + N_{2,0})}{2n}} \left( (\beta_1 - \beta_0) - \Delta + \frac{\sum_{t=1}^2 \sum_{i=1}^n A_{t,i}\epsilon_{t,i}}{N_{1,1} + N_{2,1}} - \frac{\sum_{t=1}^2 \sum_{i=1}^n (1 - A_{t,i})\epsilon_{t,i}}{N_{1,0} + N_{2,0}} \right)$$

$$= \sqrt{\frac{N_{1,0} + N_{2,0}}{2n}} \frac{\sum_{t=1}^2 \sum_{i=1}^n A_{t,i}\epsilon_{t,i}}{\sqrt{N_{1,1} + N_{2,0}}} - \sqrt{\frac{N_{1,1} + N_{2,1}}{2n}} \frac{\sum_{t=1}^2 \sum_{i=1}^n (1 - A_{t,i})\epsilon_{t,i}}{\sqrt{N_{1,0} + N_{2,0}}}$$

$$= [1, -1, 1, -1] \begin{bmatrix} \sqrt{\frac{N_{1,0}+N_{2,0}}{2n}} \frac{\sum_{i=1}^n A_{1,i}\epsilon_{1,i}}{\sqrt{N_{1,1}+N_{2,1}}} \\ \sqrt{\frac{N_{1,1}+N_{2,1}}{2n}} \frac{\sum_{i=1}^n (1-A_{1,i})\epsilon_{1,i}}{\sqrt{N_{1,0}+N_{2,0}}} \\ \sqrt{\frac{N_{1,0}+N_{2,0}}{2n}} \frac{\sum_{i=1}^n A_{2,i}\epsilon_{2,i}}{\sqrt{N_{1,1}+N_{2,1}}} \\ \sqrt{\frac{N_{1,1}+N_{2,1}}{2n}} \frac{\sum_{i=1}^n (1-A_{2,i})\epsilon_{2,i}}{\sqrt{N_{1,0}+N_{2,0}}} \end{bmatrix}$$

$$= [1, -1, 1, -1] \begin{bmatrix} \sqrt{\frac{N_{1,0}+N_{2,0}}{2(N_{1,1}+N_{2,1})}} \sqrt{\frac{N_{1,1}}{n}} \frac{\sum_{i=1}^n A_{1,i}\epsilon_{1,i}}{\sqrt{N_{1,1}}} \\ \sqrt{\frac{N_{1,1}+N_{2,1}}{2(N_{1,0}+N_{2,0})}} \sqrt{\frac{N_{1,0}}{n}} \frac{\sum_{i=1}^n (1-A_{1,i})\epsilon_{1,i}}{\sqrt{N_{1,0}}} \\ \sqrt{\frac{N_{1,0}+N_{2,0}}{2(N_{1,1}+N_{2,1})}} \sqrt{\frac{N_{2,1}}{n}} \frac{\sum_{i=1}^n A_{2,i}\epsilon_{2,i}}{\sqrt{N_{2,1}}} \\ \sqrt{\frac{N_{1,1}+N_{2,1}}{2(N_{1,0}+N_{2,0})}} \sqrt{\frac{N_{2,0}}{n}} \frac{\sum_{i=1}^n (1-A_{2,i})\epsilon_{2,i}}{\sqrt{N_{2,0}}} \end{bmatrix} \tag{19}$$

By Theorem 3, $\left( \frac{\sum_{i=1}^n A_{1,i}\epsilon_{1,i}}{\sqrt{N_{1,1}}}, \frac{\sum_{i=1}^n (1-A_{1,i})\epsilon_{1,i}}{\sqrt{N_{1,0}}}, \frac{\sum_{i=1}^n A_{2,i}\epsilon_{2,i}}{\sqrt{N_{2,1}}}, \frac{\sum_{i=1}^n (1-A_{2,i})\epsilon_{2,i}}{\sqrt{N_{2,0}}} \right) \xrightarrow{D} \mathcal{N}(\mathbf{0}, \mathbf{I}_4)$. By

Lemma 1 and Slutsky's Theorem, $\sqrt{\frac{2n(N_{1,1}+N_{2,1})}{N_{1,1}(N_{1,0}+N_{2,0})}} \sqrt{\frac{\frac{1}{2}(\frac{1}{2}+[1-\pi_2])}{2(\frac{1}{2}+\pi_2)}} = 1 + o_p(1)$, thus,

$$\sqrt{\frac{N_{1,0}+N_{2,0}}{2(N_{1,1}+N_{2,1})}} \sqrt{\frac{N_{1,1}}{n}} \frac{\sum_{i=1}^n A_{1,i}\epsilon_{1,i}}{\sqrt{N_{1,1}}}$$

$$= \left( \sqrt{\frac{2n(N_{1,1}+N_{2,1})}{N_{1,1}(N_{1,0}+N_{2,0})}} \sqrt{\frac{\frac{1}{2}(\frac{1}{2}+[1-\pi_2])}{2(\frac{1}{2}+\pi_2)}} + o_p(1) \right) \sqrt{\frac{N_{1,0}+N_{2,0}}{2(N_{1,1}+N_{2,1})}} \sqrt{\frac{N_{1,1}}{n}} \frac{\sum_{i=1}^n A_{1,i}\epsilon_{1,i}}{\sqrt{N_{1,1}}}$$

$$= \sqrt{\frac{\frac{1}{2}(\frac{1}{2}+[1-\pi_2])}{2(\frac{1}{2}+\pi_2)}} \frac{\sum_{i=1}^n A_{1,i}\epsilon_{1,i}}{\sqrt{N_{1,1}}} + o_p(1) \sqrt{\frac{N_{1,0}+N_{2,0}}{2(N_{1,1}+N_{2,1})}} \sqrt{\frac{N_{1,1}}{n}} \frac{\sum_{i=1}^n A_{1,i}\epsilon_{1,i}}{\sqrt{N_{1,1}}}$$

Note that $\sqrt{\frac{N_{1,0}+N_{2,0}}{2(N_{1,1}+N_{2,1})}}$ is stochastically bounded because for any $K > 2$,

$$\mathbb{P}\left( \frac{N_{1,0}+N_{2,0}}{2(N_{1,1}+N_{2,1})} > K \right) \leq \mathbb{P}\left( \frac{n}{N_{1,1}} > K \right) = \mathbb{P}\left( \frac{1}{K} > \frac{N_{1,1}}{n} \right) \to 0$$

where the limit holds by the law of large numbers since $N_{1,1}^{(n)} \sim \text{Binomial}(n, \frac{1}{2})$. Thus, since $\frac{N_{1,1}}{n} \leq 1$ and $\frac{\sum_{i=1}^n A_{1,i}\epsilon_{1,i}}{\sqrt{N_{1,1}}} \xrightarrow{D} \mathcal{N}(0,1)$,

$$o_p(1) \sqrt{\frac{N_{1,0}+N_{2,0}}{2(N_{1,1}+N_{2,1})}} \sqrt{\frac{N_{1,1}}{n}} \frac{\sum_{i=1}^n A_{1,i}\epsilon_{1,i}}{\sqrt{N_{1,1}}} = o_p(1)$$

We can perform the above procedure on the other three terms. Thus, equation (19) is equal to the following:

$$[1, -1, 1, -1] \begin{bmatrix} \sqrt{\frac{1/2+1-\pi_2}{4(1/2+\pi_2)}} \frac{\sum_{i=1}^n A_{1,i}\epsilon_{1,i}}{\sqrt{N_{1,1}}} \\ \sqrt{\frac{1/2+\pi_2}{4(1/2+1-\pi_2)}} \frac{\sum_{i=1}^n (1-A_{1,i})\epsilon_{1,i}}{\sqrt{N_{1,0}}} \\ \sqrt{\frac{(1/2+1-\pi_2)\pi_2}{2(1/2+\pi_2)}} \frac{\sum_{i=1}^n A_{2,i}\epsilon_{2,i}}{\sqrt{N_{2,1}}} \\ \sqrt{\frac{(1/2+\pi_2)(1-\pi_2)}{2(1/2+1-\pi_2)}} \frac{\sum_{i=1}^n (1-A_{2,i})\epsilon_{2,i}}{\sqrt{N_{2,0}}} \end{bmatrix} + o_p(1)$$

Recall that we showed earlier in Proposition 1 that

$$\pi_2^{(n)} = \pi_{\min} \vee \left[ \pi_{\max} \wedge \Phi\left( \frac{\mu_n}{\sigma_n} \right) \right] = \pi_{\min} \vee \left[ \pi_{\max} \wedge \Phi\left( B_n + C_n \right) \right]$$

$$= \pi_{\min} \vee \left[ \pi_{\max} \wedge \Phi \left( \frac{\sum_{i=1}^{n} A_{1,i}\epsilon_{1,i}}{\sqrt{2N_{1,1}}} - \frac{\sum_{i=1}^{n}(1 - A_{1,i})\epsilon_{1,i}}{\sqrt{2N_{1,0}}} + \sqrt{n}\Delta \left[ \frac{1}{2} + o(1) \right] + o(1) \right) \right]$$

When $\Delta > 0$, $\pi_2^{(n)} \xrightarrow{P} \pi_{\max}$ and when $\Delta < 0$, $\pi_2^{(n)} \xrightarrow{P} \pi_{\min}$. We now consider the $\Delta = 0$ case.

$$\pi_2^{(n)} = \pi_{\min} \vee \left[ \pi_{\max} \wedge \Phi \left( \frac{1}{\sqrt{2}} \left[ \frac{\sum_{i=1}^{n} A_{1,i}\epsilon_{1,i}}{\sqrt{N_{1,1}}} - \frac{\sum_{i=1}^{n}(1 - A_{1,i})\epsilon_{1,i}}{\sqrt{N_{1,0}}} \right] + o(1) \right) \right]$$

$$= \pi_{\min} \vee \left[ \pi_{\max} \wedge \Phi \left( \frac{1}{\sqrt{2}} \left[ \frac{\sum_{i=1}^{n} A_{1,i}\epsilon_{1,i}}{\sqrt{N_{1,1}}} - \frac{\sum_{i=1}^{n}(1 - A_{1,i})\epsilon_{1,i}}{\sqrt{N_{1,0}}} \right] \right) \right] + o(1)$$

By Slutsky's Theorem, for $Z_1, Z_2, Z_3, Z_4 \overset{i.i.d.}{\sim} \mathcal{N}(0,1)$,

$$[1,-1,1,-1] \begin{bmatrix} \sqrt{\frac{1/2+1-\pi_2}{4(1/2+\pi_2)}} \frac{\sum_{i=1}^{n} A_{1,i}\epsilon_{1,i}}{\sqrt{N_{1,1}}} \\ \sqrt{\frac{1/2+\pi_2}{4(1/2+1-\pi_2)}} \frac{\sum_{i=1}^{n}(1-A_{1,i})\epsilon_{1,i}}{\sqrt{N_{1,0}}} \\ \sqrt{\frac{(1/2+1-\pi_2)\pi_2}{2(1/2+\pi_2)}} \frac{\sum_{i=1}^{n} A_{2,i}\epsilon_{2,i}}{\sqrt{N_{2,1}}} \\ \sqrt{\frac{(1/2+\pi_2)(1-\pi_2)}{2(1/2+1-\pi_2)}} \frac{\sum_{i=1}^{n}(1-A_{2,i})\epsilon_{2,i}}{\sqrt{N_{2,0}}} \end{bmatrix} + o_p(1) \xrightarrow{D} [1,-1,1,-1] \begin{bmatrix} \sqrt{\frac{1/2+1-\pi_*}{4(1/2+\pi_*)}} Z_1 \\ \sqrt{\frac{1/2+\pi_*}{4(1/2+1-\pi_*)}} Z_2 \\ \sqrt{\frac{(1/2+1-\pi_*)\pi_*}{2(1/2+\pi_*)}} Z_3 \\ \sqrt{\frac{(1/2+\pi_*)(1-\pi_*)}{2(1/2+1-\pi_*)}} Z_4 \end{bmatrix}$$

$$= \sqrt{\frac{1/2 + 1 - \pi_*}{2(1/2 + \pi_*)}} \left( \sqrt{1/2} Z_1 + \sqrt{\pi_*} Z_3 \right) - \sqrt{\frac{1/2 + \pi_*}{2(1/2 + 1 - \pi_*)}} \left( \sqrt{1/2} Z_2 + \sqrt{1 - \pi_*} Z_4 \right)$$

where $\pi_* = \begin{cases} \pi_{\max} & \text{if } \Delta > 0 \\ \pi_{\min} & \text{if } \Delta < 0 \\ \pi_{\min} \vee (\pi_{\max} \wedge \Phi[\sqrt{1/2}(Z_1 - Z_2)]) & \text{if } \Delta = 0 \end{cases}$ $\square$

## C.2 $\epsilon$-Greedy

**Proposition 2** (Non-concentration of the sampling probabilities under zero treatment effect for $\epsilon$-greedy). *Let $T = 2$ and $\pi_1^{(n)} = \frac{1}{2}$ for all $n$. We assume that $\{\epsilon_{t,i}\}_{i=1}^n \overset{i.i.d.}{\sim} \mathcal{N}(0,1)$, and*

$$\pi_2^{(n)} = \begin{cases} 1 - \frac{\epsilon}{2} & \text{if } \frac{\sum_{i=1}^n A_{1,i} R_{1,i}}{N_{1,1}} > \frac{\sum_{i=1}^n (1-A_{1,i}) R_{1,i}}{N_{1,0}} \\ \frac{\epsilon}{2} & \text{otherwise} \end{cases}$$

*Thus, the sampling probability $\pi_2^{(n)}$ does not concentrate when $\beta_1 = \beta_0$.*

**Proof:** We define $M_n := \mathbb{I}_{\left(\frac{\sum_{i=1}^n A_{1,i} R_{1,i}}{N_{1,1}} > \frac{\sum_{i=1}^n (1-A_{1,i}) R_{1,i}}{N_{1,0}}\right)} = \mathbb{I}_{\left((\beta_1 - \beta_0) + \frac{\sum_{i=1}^n A_{1,i} \epsilon_{1,i}}{N_{1,1}} > \frac{\sum_{i=1}^n (1-A_{1,i}) \epsilon_{1,i}}{N_{1,0}}\right)}$. Note that when $M_n = 1$, $\pi_2^{(n)} = 1 - \frac{\epsilon}{2}$ and when $M_n = 0$, $\pi_2^{(n)} = \frac{\epsilon}{2}$.

When the margin is zero, $M_n$ does not concentrate because for all $N_{1,1}, N_{1,0}$, since $\epsilon_{1,i} \overset{i.i.d.}{\sim} \mathcal{N}(0,1)$,

$$\mathbb{P}\left(\frac{\sum_{i=1}^n A_{1,i} \epsilon_{1,i}}{N_{1,1}} > \frac{\sum_{i=1}^n (1-A_{1,i}) \epsilon_{1,i}}{N_{1,0}}\right) = \mathbb{P}\left(\frac{1}{\sqrt{N_{1,1}}} Z_1 - \frac{1}{\sqrt{N_{1,0}}} Z_2 > 0\right) = \frac{1}{2}$$

for $Z_1, Z_2 \overset{i.i.d.}{\sim} \mathcal{N}(0,1)$. Thus, we have shown that $\pi_2^{(n)}$ does not concentrate when $\beta_1 - \beta_0 = 0$. $\square$

**Theorem 6** (Non-uniform convergence of the OLS estimator of the treatment effect for $\epsilon$-greedy). *Assuming the setup and conditions of Proposition 2, and that $\beta_1 = b$, we show that the normalized errors of the OLS estimator converges in distribution as follows:*

$$\sqrt{N_{1,1} + N_{2,1}}(\hat{\beta}_1^{\text{OLS}} - b) \overset{D}{\to} Y$$

$$Y = \begin{cases} Z_1 & \text{if } \beta_1 - \beta_0 \neq 0 \\ \sqrt{\frac{1}{3-\epsilon}}(Z_1 - \sqrt{2-\epsilon} Z_3) \mathbb{I}_{(Z_1 > Z_2)} + \sqrt{\frac{1}{1+\epsilon}}(Z_1 - \sqrt{\epsilon} Z_3) \mathbb{I}_{(Z_1 < Z_2)} & \text{if } \beta_1 - \beta_0 = 0 \end{cases}$$

*for $Z_1, Z_2, Z_3 \overset{i.i.d.}{\sim} N(0,1)$. Note the $\beta_1 - \beta_0 = 0$ case, $Y$ is non-normal.*

**Proof:** The normalized errors of the OLS estimator for $\beta_1$ are

$$\sqrt{N_{1,1} + N_{2,1}}\left(\frac{\sum_{t=1}^2 \sum_{i=1}^n A_{t,i} R_{t,i}}{N_{1,1} + N_{2,1}} - b\right) = \frac{\sum_{t=1}^2 \sum_{i=1}^n A_{t,i} \epsilon_{t,i}}{\sqrt{N_{1,1} + N_{2,1}}}$$

$$= [1,1] \begin{bmatrix} \frac{\sum_{i=1}^n A_{1,i} \epsilon_{1,i}}{\sqrt{N_{1,1} + N_{2,1}}} \\ \frac{\sum_{i=1}^n A_{2,i} \epsilon_{2,i}}{\sqrt{N_{1,1} + N_{2,1}}} \end{bmatrix} = [1,1] \begin{bmatrix} \sqrt{\frac{N_{1,1}}{N_{1,1} + N_{2,1}}} \frac{\sum_{i=1}^n A_{1,i} \epsilon_{1,i}}{\sqrt{N_{1,1}}} \\ \sqrt{\frac{N_{2,1}}{N_{1,1} + N_{2,1}}} \frac{\sum_{i=1}^n A_{2,i} \epsilon_{2,i}}{\sqrt{N_{2,1}}} \end{bmatrix}$$

By Slutsky's Theorem and Lemma 1, $\left(\sqrt{\frac{1/2}{1/2 + \pi_2^{(n)}}} \sqrt{\frac{N_{1,1} + N_{2,1}}{N_{1,1}}}, \sqrt{\frac{\pi_2^{(n)}}{1/2 + \pi_2^{(n)}}} \sqrt{\frac{N_{1,1} + N_{2,1}}{N_{2,1}}}\right) \overset{P}{\to} (1,1)$, so

$$= [1,1] \begin{bmatrix} \left(\sqrt{\frac{1/2}{1/2 + \pi_2^{(n)}}} \sqrt{\frac{N_{1,1} + N_{2,1}}{N_{1,1}}} + o_p(1)\right) \sqrt{\frac{N_{1,1}}{N_{1,1} + N_{2,1}}} \frac{\sum_{i=1}^n A_{1,i} \epsilon_{1,i}}{\sqrt{N_{1,1}}} \\ \left(\sqrt{\frac{\pi_2^{(n)}}{1/2 + \pi_2^{(n)}}} \sqrt{\frac{N_{1,1} + N_{2,1}}{N_{2,1}}} + o_p(1)\right) \sqrt{\frac{N_{2,1}}{N_{1,1} + N_{2,1}}} \frac{\sum_{i=1}^n A_{2,i} \epsilon_{2,i}}{\sqrt{N_{2,1}}} \end{bmatrix}$$

$$= [1,1] \begin{bmatrix} \sqrt{\frac{1/2}{1/2 + \pi_2^{(n)}}} \frac{\sum_{i=1}^n A_{1,i} \epsilon_{1,i}}{\sqrt{N_{1,1}}} + o_p(1) \\ \sqrt{\frac{\pi_2^{(n)}}{1/2 + \pi_2^{(n)}}} \frac{\sum_{i=1}^n A_{2,i} \epsilon_{2,i}}{\sqrt{N_{2,1}}} + o_p(1) \end{bmatrix}$$

The last equality holds because by Theorem 3, $\left(\frac{\sum_{i=1}^n A_{1,i} \epsilon_{1,i}}{\sqrt{N_{1,1}}}, \frac{\sum_{i=1}^n A_{2,i} \epsilon_{2,i}}{\sqrt{N_{2,1}}}\right) \overset{D}{\to} \mathcal{N}(\mathbf{0}, \underline{\mathbf{I}}_2)$.

Let's define $M_n := \mathbb{I}_{\left(\frac{\sum_{i=1}^n A_{1,i} R_{1,i}}{N_{1,1}} > \frac{\sum_{i=1}^n (1-A_{1,i}) R_{1,i}}{N_{1,0}}\right)} = \mathbb{I}_{\left((\beta_1-\beta_0)+\frac{\sum_{i=1}^n A_{1,i}\epsilon_{1,i}}{N_{1,1}} > \frac{\sum_{i=1}^n (1-A_{1,i})\epsilon_{1,i}}{N_{1,0}}\right)}$.

Note that when $M_n = 1$, $\pi_2^{(n)} = 1 - \frac{\epsilon}{2}$ and when $M_n = 0$, $\pi_2^{(n)} = \frac{\epsilon}{2}$.

$$M_n = \mathbb{I}_{\left((\beta_1-\beta_0)+\frac{\sum_{i=1}^n A_{1,i}\epsilon_{1,i}}{N_{1,1}} > \frac{\sum_{i=1}^n (1-A_{1,i})\epsilon_{1,i}}{N_{1,0}}\right)} = \mathbb{I}_{\left(\sqrt{N_{1,0}}(\beta_1-\beta_0)+\sqrt{\frac{N_{1,0}}{N_{1,1}}}\frac{\sum_{i=1}^n A_{1,i}\epsilon_{1,i}}{\sqrt{N_{1,1}}} > \frac{\sum_{i=1}^n (1-A_{1,i})\epsilon_{1,i}}{\sqrt{N_{1,0}}}\right)}$$

$$= \mathbb{I}_{\left(\sqrt{N_{1,0}}(\beta_1-\beta_0)+[1+o_p(1)]\frac{\sum_{i=1}^n A_{1,i}\epsilon_{1,i}}{\sqrt{N_{1,1}}} > \frac{\sum_{i=1}^n (1-A_{1,i})\epsilon_{1,i}}{\sqrt{N_{1,0}}}\right)}$$

where the last equality holds because $\sqrt{\frac{N_{1,0}}{N_{1,1}}} \xrightarrow{P} 1$ by Lemma 1, Slutsky's Theorem, and continuous mapping theorem. Thus, by Proposition 2,

$$M^{(n)} \xrightarrow{P} \begin{cases} 1 & \text{if } \beta_1 - \beta_0 > 0 \\ 0 & \text{if } \beta_1 - \beta_0 < 0 \\ \text{does not concentrate} & \text{if } \beta_1 - \beta_0 = 0 \end{cases}$$

Note that

$$\begin{bmatrix} \sqrt{\frac{\frac{1}{2}}{\frac{1}{2}+\pi_2^{(n)}}} \frac{\sum_{i=1}^n A_{1,i}\epsilon_{1,i}}{\sqrt{N_{1,1}}} + o_p(1) \\ \sqrt{\frac{\pi_2^{(n)}}{\frac{1}{2}+\pi_2^{(n)}}} \frac{\sum_{i=1}^n A_{2,i}\epsilon_{2,i}}{\sqrt{N_{2,1}}} + o_p(1) \end{bmatrix}$$

$$= \begin{bmatrix} \sqrt{\frac{\frac{1}{2}}{\frac{1}{2}+1-\frac{\epsilon}{2}}} \frac{\sum_{i=1}^n A_{1,i}\epsilon_{1,i}}{\sqrt{N_{1,1}}} + o_p(1) \\ \sqrt{\frac{1-\epsilon/2}{\frac{1}{2}+1-\frac{\epsilon}{2}}} \frac{\sum_{i=1}^n A_{2,i}\epsilon_{2,i}}{\sqrt{N_{2,1}}} + o_p(1) \end{bmatrix} M_n + \begin{bmatrix} \sqrt{\frac{\frac{1}{2}}{\frac{1}{2}+\frac{\epsilon}{2}}} \frac{\sum_{i=1}^n A_{1,i}\epsilon_{1,i}}{\sqrt{N_{1,1}}} + o_p(1) \\ \sqrt{\frac{\frac{\epsilon}{2}}{\frac{1}{2}+\frac{\epsilon}{2}}} \frac{\sum_{i=1}^n A_{2,i}\epsilon_{2,i}}{\sqrt{N_{2,1}}} + o_p(1) \end{bmatrix} (1-M_n)$$

Also note that by Theorem 3, $\left(\frac{\sum_{i=1}^n A_{1,i}\epsilon_{1,i}}{\sqrt{N_{1,1}}}, \frac{\sum_{i=1}^n (1-A_{1,i})\epsilon_{1,i}}{\sqrt{N_{1,0}}}, \frac{\sum_{i=1}^n A_{2,i}\epsilon_{2,i}}{\sqrt{N_{2,1}}}, \frac{\sum_{i=1}^n (1-A_{2,i})\epsilon_{2,i}}{\sqrt{N_{2,1}}}\right) \xrightarrow{D} \mathcal{N}(\mathbf{0}, \mathbf{I}_4)$.

When $\beta_1 > \beta_0$, $M_n \xrightarrow{P} 1$ and when $\beta_1 < \beta_0$, $M_n \xrightarrow{P} 0$; in both these cases the normalized errors are asymptotically normal. We now focus on the case that $\beta_1 = \beta_0$. By continuous mapping theorem and Slutsky's theorem for $Z_1, Z_2, Z_3, Z_4 \overset{i.i.d.}{\sim} \mathcal{N}(0,1)$,

$$= [1,1] \begin{bmatrix} \sqrt{\frac{\frac{1}{2}}{\frac{1}{2}+1-\frac{\epsilon}{2}}} \frac{\sum_{i=1}^n A_{1,i}\epsilon_{1,i}}{\sqrt{N_{1,1}}} + o_p(1) \\ \sqrt{\frac{1-\epsilon/2}{\frac{1}{2}+1-\frac{\epsilon}{2}}} \frac{\sum_{i=1}^n A_{2,i}\epsilon_{2,i}}{\sqrt{N_{2,1}}} + o_p(1) \end{bmatrix} \mathbb{I}_{\left([1+o(1)]\frac{\sum_{i=1}^n A_{1,i}\epsilon_{1,i}}{\sqrt{N_{1,1}}} > \frac{\sum_{i=1}^n (1-A_{1,i})\epsilon_{1,i}}{\sqrt{N_{1,0}}}\right)}$$

$$+ [1,1] \begin{bmatrix} \sqrt{\frac{\frac{1}{2}}{\frac{1}{2}+\frac{\epsilon}{2}}} \frac{\sum_{i=1}^n A_{1,i}\epsilon_{1,i}}{\sqrt{N_{1,1}}} + o_p(1) \\ \sqrt{\frac{\frac{\epsilon}{2}}{\frac{1}{2}+\frac{\epsilon}{2}}} \frac{\sum_{i=1}^n A_{2,i}\epsilon_{2,i}}{\sqrt{N_{2,1}}} + o_p(1) \end{bmatrix} \left(1 - \mathbb{I}_{\left([1+o(1)]\frac{\sum_{i=1}^n A_{1,i}\epsilon_{1,i}}{\sqrt{N_{1,1}}} > \frac{\sum_{i=1}^n (1-A_{1,i})\epsilon_{1,i}}{\sqrt{N_{1,0}}}\right)}\right)$$

$$\xrightarrow{D} [1,1] \begin{bmatrix} \sqrt{\frac{1/2}{1/2+1-\epsilon/2}} Z_1 \\ \sqrt{\frac{1-\epsilon/2}{1/2+1-\epsilon/2}} Z_3 \end{bmatrix} \mathbb{I}_{(Z_1>Z_2)} + [1,1] \begin{bmatrix} \sqrt{\frac{1/2}{1/2+\epsilon/2}} Z_1 \\ \sqrt{\frac{\epsilon/2}{1/2+\epsilon/2}} Z_3 \end{bmatrix} \mathbb{I}_{(Z_1<Z_2)}$$

$$= \left(\sqrt{\frac{1}{3-\epsilon}} Z_1 + \sqrt{\frac{2-\epsilon}{3-\epsilon}} Z_3\right) \mathbb{I}_{(Z_1>Z_2)} + \left(\sqrt{\frac{1}{1+\epsilon}} Z_1 + \sqrt{\frac{\epsilon}{1+\epsilon}} Z_3\right) \mathbb{I}_{(Z_1<Z_2)}$$

Thus,

$$\frac{\sum_{t=1}^2 \sum_{i=1}^n A_{t,i}\epsilon_{t,i}}{\sqrt{N_{1,1}+N_{2,1}}} \xrightarrow{D} Y$$

$$Y := \begin{cases} \sqrt{\frac{1}{3-\epsilon}}(Z_1 - \sqrt{2-\epsilon}Z_3) & \text{if } \beta_1 - \beta_0 > 0 \\ \sqrt{\frac{1}{1+\epsilon}}(Z_1 - \sqrt{\epsilon}Z_3) & \text{if } \beta_1 - \beta_0 < 0 \\ \sqrt{\frac{1}{3-\epsilon}}(Z_1 - \sqrt{2-\epsilon}Z_3)\mathbb{I}_{(Z_1>Z_2)} + \sqrt{\frac{1}{1+\epsilon}}(Z_1 - \sqrt{\epsilon}Z_3)\mathbb{I}_{(Z_1<Z_2)} & \text{if } \beta_1 - \beta_0 = 0 \quad \square \end{cases}$$

## C.3 UCB

**Theorem 7** (Asymptotic non-Normality under zero treatment effect for clipped UCB). *Let $T = 2$ and $\pi_1^{(n)} = \frac{1}{2}$ for all $n$. We assume that $\{\epsilon_{t,i}\}_{i=1}^n \overset{i.i.d.}{\sim} \mathcal{N}(0,1)$, and*

$$\pi_2^{(n)} = \begin{cases} \pi_{\max} & \text{if } U_1 > U_0 \\ 1 - \pi_{\max} & \text{otherwise} \end{cases}$$

*where we define the upper confidence bounds $U$ for any confidence level $\delta$ with $0 < \delta < 1$ as follows:*

$$U_1 = \begin{cases} \infty & \text{if } N_{1,1} = 0 \\ \frac{\sum_{i=1}^n A_{1,i} R_{1,i}}{N_{1,1}} + \sqrt{\frac{2 \log 1/\delta}{N_{1,1}}} & \text{otherwise} \end{cases}$$

$$U_0 = \begin{cases} \infty & \text{if } N_{1,0} = 0 \\ \frac{\sum_{i=1}^n (1-A_{1,i}) R_{1,i}}{N_{1,1}} + \sqrt{\frac{2 \log 1/\delta}{N_{1,0}}} & \text{otherwise} \end{cases}$$

*Assuming above conditions, and that $\beta_1 = b$, we show that the normalized errors of the OLS estimator converges in distribution as follows:*

$$\sqrt{N_{1,1} + N_{2,1}}\left(\hat{\beta}_1^{\text{OLS}} - b\right) \overset{D}{\to} Y$$

$$Y = \begin{cases} Z_1 & \text{if } \Delta = 0 \\ \left(\sqrt{\frac{\frac{1}{2}}{\frac{1}{2}+\pi_{\max}}} Z_1 + \sqrt{\frac{\pi_{\max}}{\frac{1}{2}+\pi_{\max}}} Z_3\right) \mathbb{I}_{(Z_1 > Z_2)} + \left(\sqrt{\frac{\frac{1}{2}}{\frac{3}{2}-\pi_{\max}}} Z_1 + \sqrt{\frac{1-\pi_{\max}}{\frac{3}{2}-\pi_{\max}}} Z_3\right) \mathbb{I}_{(Z_1 < Z_2)} & \text{if } \Delta = 0 \end{cases}$$

*for $Z_1, Z_2, Z_3 \overset{i.i.d.}{\sim} N(0,1)$. Note the $\Delta := \beta_1 - \beta_0 = 0$ case, $Y$ is non-normal.*

**Proof:** The proof is very similar to that of asymptotic non-normality result for $\epsilon$-Greedy. By the same arguments made as in the $\epsilon$-Greedy case, we have that

$$\sqrt{N_{1,1} + N_{2,1}}\left(\frac{\sum_{t=1}^2 \sum_{i=1}^n A_{t,i} R_{t,i}}{N_{1,1} + N_{2,1}} - b\right) = [1,1]\begin{bmatrix} \sqrt{\frac{1/2}{1/2+\pi_2^{(n)}}} \frac{\sum_{i=1}^n A_{1,i}\epsilon_{1,i}}{\sqrt{N_{1,1}}} + o_p(1) \\ \sqrt{\frac{\pi_2^{(n)}}{1/2+\pi_2^{(n)}}} \frac{\sum_{i=1}^n A_{2,i}\epsilon_{2,i}}{\sqrt{N_{2,1}}} + o_p(1) \end{bmatrix}$$

Assuming $n \geq 1$, we then define
$$M_n := \mathbb{I}_{(U_1 > U_0)}$$

$$= \mathbb{I}_{(N_{1,1}>0, N_{1,0}>0)} \mathbb{I}_{\left(\frac{\sum_{i=1}^n A_{1,i} R_{1,i}}{N_{1,1}} + \sqrt{\frac{2\log 1/\delta}{N_{1,1}}} > \frac{\sum_{i=1}^n (1-A_{1,i}) R_{1,i}}{N_{1,1}} + \sqrt{\frac{2\log 1/\delta}{N_{1,0}}}\right)} + \mathbb{I}_{(N_{1,1}=0, N_{1,0}>0)}$$

$$= \mathbb{I}_{(N_{1,1}>0, N_{1,0}>0)} \mathbb{I}_{\left((\beta_1-\beta_0) + \frac{\sum_{i=1}^n A_{1,i}\epsilon_{1,i}}{N_{1,1}} + \sqrt{\frac{2\log 1/\delta}{N_{1,1}}} > \frac{\sum_{i=1}^n (1-A_{1,i})\epsilon_{1,i}}{N_{1,1}} + \sqrt{\frac{2\log 1/\delta}{N_{1,0}}}\right)} + \mathbb{I}_{(N_{1,1}=0, N_{1,0}>0)}$$

$$= \mathbb{I}_{(N_{1,1}>0, N_{1,0}>0)} \mathbb{I}_{\left(\sqrt{N_{1,0}}(\beta_1-\beta_0) + \sqrt{\frac{N_{1,0}}{N_{1,1}}}\left[\frac{\sum_{i=1}^n A_{1,i}\epsilon_{1,i}}{\sqrt{N_{1,1}}} + \sqrt{2\log 1/\delta}\right] > \frac{\sum_{i=1}^n (1-A_{1,i})\epsilon_{1,i}}{\sqrt{N_{1,1}}} + \sqrt{2\log 1/\delta}\right)}$$
$$+ \mathbb{I}_{(N_{1,1}=0, N_{1,0}>0)}$$

Note that $\frac{N_{1,0}}{N_{1,1}} \overset{P}{\to} 1$ by Lemma 1. Thus by Slutsky's Theorem and continuous mapping theorem,

$$= \mathbb{I}_{\left(\sqrt{N_{1,0}}(\beta_1-\beta_0) + [1+o_p(1)]\frac{\sum_{i=1}^n A_{1,i}\epsilon_{1,i}}{\sqrt{N_{1,1}}} + o_p(1) > \frac{\sum_{i=1}^n (1-A_{1,i})\epsilon_{1,i}}{\sqrt{N_{1,1}}}\right)} + o_p(1) \qquad (20)$$

Note that

$$\begin{bmatrix} \sqrt{\frac{\frac{1}{2}}{\frac{1}{2}+\pi_2^{(n)}}} \frac{\sum_{i=1}^n A_{1,i}\epsilon_{1,i}}{\sqrt{N_{1,1}}} + o_p(1) \\ \sqrt{\frac{\pi_2^{(n)}}{\frac{1}{2}+\pi_2^{(n)}}} \frac{\sum_{i=1}^n A_{2,i}\epsilon_{2,i}}{\sqrt{N_{2,1}}} + o_p(1) \end{bmatrix}$$

$$= \begin{bmatrix} \sqrt{\frac{\frac{1}{2}}{\frac{1}{2}+\pi_{\max}}} \frac{\sum_{i=1}^n A_{1,i}\epsilon_{1,i}}{\sqrt{N_{1,1}}} + o_p(1) \\ \sqrt{\frac{\pi_{\max}}{\frac{1}{2}+\pi_{\max}}} \frac{\sum_{i=1}^n A_{2,i}\epsilon_{2,i}}{\sqrt{N_{2,1}}} + o_p(1) \end{bmatrix} M_n + \begin{bmatrix} \sqrt{\frac{\frac{1}{2}}{\frac{1}{2}+1-\pi_{\max}}} \frac{\sum_{i=1}^n A_{1,i}\epsilon_{1,i}}{\sqrt{N_{1,1}}} + o_p(1) \\ \sqrt{\frac{1-\pi_{\max}}{\frac{1}{2}+1-\pi_{\max}}} \frac{\sum_{i=1}^n A_{2,i}\epsilon_{2,i}}{\sqrt{N_{2,1}}} + o_p(1) \end{bmatrix} (1 - M_n)$$

Let $(Z_1^{(n)}, Z_2^{(n)}, Z_3^{(n)}, Z_4^{(n)}) := \left( \frac{\sum_{i=1}^{n} A_{1,i} \epsilon_{1,i}}{\sqrt{N_{1,1}}}, \frac{\sum_{i=1}^{n} (1-A_{1,i}) \epsilon_{1,i}}{\sqrt{N_{1,0}}}, \frac{\sum_{i=1}^{n} A_{2,i} \epsilon_{2,i}}{\sqrt{N_{2,1}}}, \frac{\sum_{i=1}^{n} (1-A_{2,i}) \epsilon_{2,i}}{\sqrt{N_{2,1}}} \right).$

Note that by Theorem 3, $(Z_1^{(n)}, Z_2^{(n)}, Z_3^{(n)}, Z_4^{(n)}) \xrightarrow{D} \mathcal{N}(\mathbf{0}, \underline{\mathbf{I}}_4)$.

When $\beta_1 > \beta_0$, $M_n \xrightarrow{P} 1$ and when $\beta_1 < \beta_0$, $M_n \xrightarrow{P} 0$; in both these cases the normalized errors are asymptotically normal. We now focus on the case that $\beta_1 = \beta_0$. By continuous mapping theorem and Slutsky's theorem,

$$
= [1,1] \begin{bmatrix} \sqrt{\frac{\frac{1}{2}}{\frac{1}{2}+\pi_{\max}}} Z_1^{(n)} + o_p(1) \\ \sqrt{\frac{\pi_{\max}}{\frac{1}{2}+\pi_{\max}}} Z_3^{(n)} + o_p(1) \end{bmatrix} \left[ \mathbb{I}_{\left( [1+o_p(1)] Z_1^{(n)} + o_p(1) > Z_2^{(n)} \right)} + o_p(1) \right]
$$

$$
+ [1,1] \begin{bmatrix} \sqrt{\frac{\frac{1}{2}}{\frac{1}{2}+1-\pi_{\max}}} Z_1^{(n)} + o_p(1) \\ \sqrt{\frac{1-\pi_{\max}}{\frac{1}{2}+1-\pi_{\max}}} Z_3^{(n)} + o_p(1) \end{bmatrix} \left[ 1 - \mathbb{I}_{\left( [1+o_p(1)] Z_1^{(n)} + o_p(1) > Z_2^{(n)} \right)} + o_p(1) \right] \Bigg)
$$

$$
\xrightarrow{D} [1,1] \begin{bmatrix} \sqrt{\frac{\frac{1}{2}}{\frac{1}{2}+\pi_{\max}}} Z_1 \\ \sqrt{\frac{\pi_{\max}}{\frac{1}{2}+\pi_{\max}}} Z_3 \end{bmatrix} \mathbb{I}_{(Z_1 > Z_2)} + [1,1] \begin{bmatrix} \sqrt{\frac{\frac{1}{2}}{\frac{1}{2}+1-\pi_{\max}}} Z_1 \\ \sqrt{\frac{1-\pi_{\max}}{\frac{1}{2}+1-\pi_{\max}}} Z_3 \end{bmatrix} \mathbb{I}_{(Z_1 < Z_2)}
$$

$$
= \left( \sqrt{\frac{\frac{1}{2}}{\frac{1}{2}+\pi_{\max}}} Z_1 + \sqrt{\frac{\pi_{\max}}{\frac{1}{2}+\pi_{\max}}} Z_3 \right) \mathbb{I}_{(Z_1 > Z_2)} + \left( \sqrt{\frac{\frac{1}{2}}{\frac{3}{2}-\pi_{\max}}} Z_1 + \sqrt{\frac{1-\pi_{\max}}{\frac{3}{2}-\pi_{\max}}} Z_3 \right) \mathbb{I}_{(Z_1 < Z_2)}. \quad \square
$$

Note that (20) implies that if $\beta_1 = \beta_0$, that $\pi_2^{(n)}$ will not concentrate.

# D  Asymptotic Normality of the Batched OLS Estimator: Multi-Arm Bandits

**Theorem 3**  (Asymptotic normality of Batched OLS estimator for multi-arm bandits) *Assuming Conditions 6 (weak moments) and 3 (conditionally i.i.d. actions), and a clipping rate of $f(n) = \omega(\frac{1}{n})$ (Definition 1),*

$$
\begin{bmatrix}
\begin{bmatrix} N_{1,0} & 0 \\ 0 & N_{1,1} \end{bmatrix}^{1/2} (\hat{\boldsymbol{\beta}}_1^{\text{BOLS}} - \boldsymbol{\beta}_1) \\[2mm]
\begin{bmatrix} N_{2,0} & 0 \\ 0 & N_{2,1} \end{bmatrix}^{1/2} (\hat{\boldsymbol{\beta}}_2^{\text{BOLS}} - \boldsymbol{\beta}_2) \\[2mm]
\vdots \\[2mm]
\begin{bmatrix} N_{T,0} & 0 \\ 0 & N_{T,1} \end{bmatrix}^{1/2} (\hat{\boldsymbol{\beta}}_T^{\text{BOLS}} - \boldsymbol{\beta}_T)
\end{bmatrix}
\xrightarrow{D} \mathcal{N}(0, \sigma^2 \boldsymbol{I}_{2T})
$$

*where $\boldsymbol{\beta}_t = (\beta_{t,0}, \beta_{t,1})$, $N_{t,1} = \sum_{i=1}^n A_{t,i}$, and $N_{t,0} = \sum_{i=1}^n (1 - A_{t,i})$. Note in the body of this paper, we state Theorem 3 with conditions that are are sufficient for the weaker conditions we use here.*

**Lemma 1.** *Assuming the conditions of Theorem 3, for any batch $t \in [1:T]$,*

$$
\frac{N_{t,1}}{n\pi_t^{(n)}} = \frac{\sum_{i=1}^n A_{t,i}}{n\pi_t^{(n)}} \xrightarrow{P} 1 \quad \text{and} \quad \frac{N_{t,0}}{n(1-\pi_t^{(n)})} = \frac{\sum_{i=1}^n (1 - A_{t,i})}{n(1-\pi_t^{(n)})} \xrightarrow{P} 1
$$

**Proof of Lemma 1:**  To prove that $\frac{N_{t,1}}{n\pi_t^{(n)}} \xrightarrow{P} 1$, it is equivalent to show that $\frac{1}{n\pi_t^{(n)}} \sum_{i=1}^n (A_{t,i} - \pi_t^{(n)}) \xrightarrow{P} 0$. Let $\epsilon > 0$.

$$
\mathbb{P}\left( \left| \frac{1}{n\pi_t^{(n)}} \sum_{i=1}^n (A_{t,i} - \pi_t^{(n)}) \right| > \epsilon \right)
$$

$$
= \mathbb{P}\left( \left| \frac{1}{n\pi_t^{(n)}} \sum_{i=1}^n (A_{t,i} - \pi_t^{(n)}) \right| \left[ \mathbb{I}_{(\pi_t^{(n)} \in [f(n), 1-f(n)])} + \mathbb{I}_{(\pi_t^{(n)} \notin [f(n), 1-f(n)])} \right] > \epsilon \right)
$$

$$
\leq \mathbb{P}\left( \left| \frac{1}{n\pi_t^{(n)}} \sum_{i=1}^n (A_{t,i} - \pi_t^{(n)}) \right| \mathbb{I}_{(\pi_t^{(n)} \in [f(n), 1-f(n)])} > \frac{\epsilon}{2} \right)
$$

$$
+ \mathbb{P}\left( \left| \frac{1}{n\pi_t^{(n)}} \sum_{i=1}^n (A_{t,i} - \pi_t^{(n)}) \right| \mathbb{I}_{(\pi_t^{(n)} \notin [f(n), 1-f(n)])} > \frac{\epsilon}{2} \right)
$$

Since by our clipping assumption, $\mathbb{I}_{(\pi_t^{(n)} \in [f(n), 1-f(n)])} \xrightarrow{P} 1$, the second probability in the summation above converges to 0 as $n \to \infty$. We will now show that the first probability in the summation above also goes to zero. Note that $\mathbb{E}\left[ \frac{1}{n\pi_t^{(n)}} \sum_{i=1}^n (A_{t,i} - \pi_t^{(n)}) \right] = \mathbb{E}\left[ \frac{1}{n\pi_t^{(n)}} \sum_{i=1}^n (\mathbb{E}[A_{t,i}|H_{t-1}^{(n)}] - \pi_t^{(n)}) \right] = 0$. So by Chebychev inequality, for any $\epsilon > 0$,

$$
\mathbb{P}\left( \left| \frac{1}{n\pi_t^{(n)}} \sum_{i=1}^n (A_{t,i} - \pi_t^{(n)}) \right| \mathbb{I}_{(\pi_t^{(n)} \in [f(n), 1-f(n)])} > \epsilon \right)
$$

$$
\leq \frac{1}{\epsilon^2 n^2} \mathbb{E}\left[ \frac{1}{(\pi_t^{(n)})^2} \left( \sum_{i=1}^n (A_{t,i} - \pi_t^{(n)}) \right)^2 \mathbb{I}_{(\pi_t^{(n)} \in [f(n), 1-f(n)])} \right]
$$

$$
\leq \frac{1}{\epsilon^2 n^2} \sum_{i=1}^n \sum_{j=1}^n \mathbb{E}\left[ \frac{1}{(\pi_t^{(n)})^2} (A_{t,i} - \pi_t^{(n)})(A_{t,j} - \pi_t^{(n)}) \mathbb{I}_{(\pi_t^{(n)} \in [f(n), 1-f(n)])} \right]
$$

$$
= \frac{1}{\epsilon^2 n^2} \sum_{i=1}^n \sum_{j=1}^n \mathbb{E}\left[ \frac{1}{(\pi_t^{(n)})^2} \mathbb{I}_{(\pi_t^{(n)} \in [f(n), 1-f(n)])} \mathbb{E}\left[ A_{t,i} A_{t,j} - \pi_t^{(n)}(A_{t,i} + A_{t,j}) + (\pi_t^{(n)})^2 \big| H_{t-1}^{(n)} \right] \right]
$$

$$= \frac{1}{\epsilon^2 n^2} \sum_{i=1}^{n} \sum_{j=1}^{n} \mathbb{E}\left[ \frac{1}{(\pi_t^{(n)})^2} \mathbb{I}_{(\pi_t^{(n)} \in [f(n), 1-f(n)])} \left( \mathbb{E}[A_{t,i} A_{t,j} | H_{t-1}^{(n)}] - (\pi_t^{(n)})^2 \right) \right] \quad (21)$$

Note that if $i \neq j$, since $A_{t,i} \overset{i.i.d.}{\sim}$ Bernoulli($\pi_t^{(n)}$), $\mathbb{E}[A_{t,i} A_{t,j} | H_{t-1}^{(n)}] = \mathbb{E}[A_{t,i} | H_{t-1}^{(n)}] E[A_{t,j} | H_{t-1}^{(n)}] = (\pi_t^{(n)})^2$, so (21) above equals the following

$$= \frac{1}{\epsilon^2 n^2} \sum_{i=1}^{n} \mathbb{E}\left[ \frac{1}{(\pi_t^{(n)})^2} \mathbb{I}_{(\pi_t^{(n)} \in [f(n), 1-f(n)])} \left( \mathbb{E}[A_{t,i} | H_{t-1}^{(n)}] - (\pi_t^{(n)})^2 \right) \right]$$

$$= \frac{1}{\epsilon^2 n^2} \sum_{i=1}^{n} \mathbb{E}\left[ \frac{1 - \pi_t^{(n)}}{\pi_t^{(n)}} \mathbb{I}_{(\pi_t^{(n)} \in [f(n), 1-f(n)])} \right] = \frac{1}{\epsilon^2 n} \mathbb{E}\left[ \frac{1 - \pi_t^{(n)}}{\pi_t^{(n)}} \mathbb{I}_{(\pi_t^{(n)} \in [f(n), 1-f(n)])} \right] \leq \frac{1}{\epsilon^2 n} \frac{1}{f(n)} \to 0$$

where the limit holds because we assume $f(n) = \omega(\frac{1}{n})$ so $f(n) n \to \infty$. We can make a very similar argument for $\frac{N_{t,0}}{n(1-\pi_t^{(n)})} \overset{P}{\to} 1$. $\quad \square$

**Proof for Theorem 3 (Asymptotic normality of Batched OLS estimator for multi-arm bandits):**
For readability, for this proof we drop the $(n)$ superscript on $\pi_t^{(n)}$. Note that

$$\begin{bmatrix} N_{t,0} & 0 \\ 0 & N_{t,1} \end{bmatrix}^{1/2} (\hat{\boldsymbol{\beta}}_t^{\text{BOLS}} - \boldsymbol{\beta}_t) = \begin{bmatrix} N_{t,0} & 0 \\ 0 & N_{t,1} \end{bmatrix}^{-1/2} \sum_{i=1}^{n} \begin{bmatrix} 1 - A_{t,i} \\ A_{t,i} \end{bmatrix} \epsilon_{t,i}.$$

We want to show that

$$\begin{bmatrix} \begin{bmatrix} N_{0,1} & 0 \\ 0 & N_{1,1} \end{bmatrix}^{-1/2} \sum_{i=1}^{n} \begin{bmatrix} 1 - A_{1,i} \\ A_{1,i} \end{bmatrix} \epsilon_{1,i} \\ \begin{bmatrix} N_{0,2} & 0 \\ 0 & N_{1,2} \end{bmatrix}^{-1/2} \sum_{i=1}^{n} \begin{bmatrix} 1 - A_{2,i} \\ A_{2,i} \end{bmatrix} \epsilon_{2,i} \\ \vdots \\ \begin{bmatrix} N_{t,0} & 0 \\ 0 & N_{t,1} \end{bmatrix}^{-1/2} \sum_{i=1}^{n} \begin{bmatrix} 1 - A_{T,i} \\ A_{T,i} \end{bmatrix} \epsilon_{T,i} \end{bmatrix} = \begin{bmatrix} N_{0,1}^{-1/2} \sum_{i=1}^{n} (1 - A_{1,i}) \epsilon_{1,i} \\ N_{1,1}^{-1/2} \sum_{i=1}^{n} A_{1,i} \epsilon_{1,i} \\ N_{0,2}^{-1/2} \sum_{i=1}^{n} (1 - A_{2,i}) \epsilon_{2,i} \\ N_{1,2}^{-1/2} \sum_{i=1}^{n} A_{2,i} \epsilon_{2,i} \\ \vdots \\ N_{t,0}^{-1/2} \sum_{i=1}^{n} (1 - A_{T,i}) \epsilon_{T,i} \\ N_{t,1}^{-1/2} \sum_{i=1}^{n} A_{T,i} \epsilon_{T,i} \end{bmatrix} \overset{D}{\to} \mathcal{N}(0, \sigma^2 \mathbf{I}_{2T}).$$

By Lemma 1 and Slutsky's Theorem it is sufficient to show that as $n \to \infty$,

$$\begin{bmatrix} \frac{1}{\sqrt{n(1-\pi_1)}} \sum_{i=1}^{n} (1 - A_{1,i}) \epsilon_{1,i} \\ \frac{1}{\sqrt{n\pi_1}} \sum_{i=1}^{n} A_{1,i} \epsilon_{1,i} \\ \frac{1}{\sqrt{n(1-\pi_2)}} \sum_{i=1}^{n} (1 - A_{2,i}) \epsilon_{2,i} \\ \frac{1}{\sqrt{n\pi_2}} \sum_{i=1}^{n} A_{2,i} \epsilon_{2,i} \\ \vdots \\ \frac{1}{\sqrt{n(1-\pi_T)}} \sum_{i=1}^{n} (1 - A_{T,i}) \epsilon_{T,i} \\ \frac{1}{\sqrt{n\pi_T}} \sum_{i=1}^{n} A_{T,i} \epsilon_{T,i} \end{bmatrix} = \begin{bmatrix} \frac{1}{\sqrt{n}} \begin{bmatrix} 1 - \pi_{1,1} & 0 \\ 0 & \pi_{1,1} \end{bmatrix}^{-1/2} \sum_{i=1}^{n} \begin{bmatrix} 1 - A_{1,i} \\ A_{1,i} \end{bmatrix} \epsilon_{1,i} \\ \frac{1}{\sqrt{n}} \begin{bmatrix} 1 - \pi_2^{(n)} & 0 \\ 0 & \pi_2^{(n)} \end{bmatrix}^{-1/2} \sum_{i=1}^{n} \begin{bmatrix} 1 - A_{2,i} \\ A_{2,i} \end{bmatrix} \epsilon_{2,i} \\ \vdots \\ \frac{1}{\sqrt{n}} \begin{bmatrix} 1 - \pi_t^{(n)} & 0 \\ 0 & \pi_t^{(n)} \end{bmatrix}^{-1/2} \sum_{i=1}^{n} \begin{bmatrix} 1 - A_{T,i} \\ A_{T,i} \end{bmatrix} \epsilon_{T,i} \end{bmatrix} \overset{D}{\to} \mathcal{N}(0, \sigma^2 \mathbf{I}_{2T})$$

By Cramer-Wold device, it is sufficient to show that for any fixed vector $\mathbf{c} \in \mathbb{R}^{2T}$ s.t. $\|\mathbf{c}\|_2 = 1$ that as $n \to \infty$,

$$\mathbf{c}^\top \begin{bmatrix} n^{-1/2} \begin{bmatrix} 1 - \pi_{1,1} & 0 \\ 0 & \pi_{1,1} \end{bmatrix}^{-1/2} \sum_{i=1}^{n} \begin{bmatrix} 1 - A_{1,i} \\ A_{1,i} \end{bmatrix} \epsilon_{1,i} \\ n^{-1/2} \begin{bmatrix} 1 - \pi_2^{(n)} & 0 \\ 0 & \pi_2^{(n)} \end{bmatrix}^{-1/2} \sum_{i=1}^{n} \begin{bmatrix} 1 - A_{2,i} \\ A_{2,i} \end{bmatrix} \epsilon_{2,i} \\ \vdots \\ n^{-1/2} \begin{bmatrix} 1 - \pi_t^{(n)} & 0 \\ 0 & \pi_t^{(n)} \end{bmatrix}^{-1/2} \sum_{i=1}^{n} \begin{bmatrix} 1 - A_{T,i} \\ A_{T,i} \end{bmatrix} \epsilon_{T,i} \end{bmatrix} \overset{D}{\to} \mathcal{N}(0, \sigma^2)$$

Let us break up $\mathbf{c}$ so that $\mathbf{c} = [\mathbf{c}_1, \mathbf{c}_2, ..., \mathbf{c}_T]^\top \in \mathbb{R}^{2T}$ with $\mathbf{c}_t \in \mathbb{R}^2$ for $t \in [1\colon T]$. The above is equivalent to

$$\sum_{t=1}^{T} n^{-1/2} \mathbf{c}_t^\top \begin{bmatrix} 1 - \pi_t^{(n)} & 0 \\ 0 & \pi_t^{(n)} \end{bmatrix}^{-1/2} \sum_{i=1}^{n} \begin{bmatrix} 1 - A_{t,i} \\ A_{t,i} \end{bmatrix} \epsilon_{t,i} \xrightarrow{D} \mathcal{N}(0, \sigma^2)$$

Let us define $Y_{t,i} := n^{-1/2} \mathbf{c}_t^\top \begin{bmatrix} 1 - \pi_{t,i} & 0 \\ 0 & \pi_{t,i} \end{bmatrix}^{-1/2} \begin{bmatrix} 1 - A_{t,i} \\ A_{t,i} \end{bmatrix} \epsilon_{t,i}$.

The sequence $\{Y_{1,1}, Y_{1,2}, ..., Y_{1,n}, ..., Y_{T,1}, Y_{T,2}, ..., Y_{T,n}\}$ is a martingale with respect to sequence of histories $\{H_t^{(n)}\}_{t=1}^T$, since

$$\mathbb{E}[Y_{t,i}|H_{t-1}^{(n)}] = n^{-1/2} \mathbf{c}_t^\top \begin{bmatrix} 1 - \pi_t^{(n)} & 0 \\ 0 & \pi_t^{(n)} \end{bmatrix}^{-1/2} \mathbb{E}\left[ \begin{bmatrix} 1 - A_{t,i} \\ A_{t,i} \end{bmatrix} \epsilon_{t,i} \Big| H_{t-1}^{(n)} \right]$$

$$= n^{-1/2} \mathbf{c}_t^\top \begin{bmatrix} 1 - \pi_t^{(n)} & 0 \\ 0 & \pi_t^{(n)} \end{bmatrix}^{-1/2} \mathbb{E}\left[ \begin{bmatrix} (1 - \pi_t^{(n)})\mathbb{E}[\epsilon_{t,i}|H_{t-1}^{(n)}, A_{t,i} = 0] \\ \pi_{t,i}\mathbb{E}[\epsilon_{t,i}|H_{t-1}^{(n)}, A_{t,i} = 1] \end{bmatrix} \Big| H_{t-1}^{(n)} \right] = 0$$

for all $i \in [1\colon n]$ and all $t \in [1\colon T]$. We then apply [8] martingale central limit theorem to $Y_{t,i}$ to show the desired result (see the proof of Theorem 5 in Appendix B for the statement of the martingale CLT conditions).

**Condition(a): Martingale Condition**    The first condition holds because $\mathbb{E}[Y_{t,i}|H_{t-1}^{(n)}] = 0$ for all $i \in [1\colon n]$ and all $t \in [1\colon T]$.

**Condition(b): Conditional Variance**

$$\sum_{t=1}^{T}\sum_{i=1}^{n} E[Y_{n,t,i}^2|H_{t-1}^{(n)}] = \sum_{t=1}^{T}\sum_{i=1}^{n} \mathbb{E}\left[ \left( \frac{1}{\sqrt{n}}\mathbf{c}_t^\top \begin{bmatrix} 1 - \pi_t^{(n)} & 0 \\ 0 & \pi_t^{(n)} \end{bmatrix}^{-1/2} \begin{bmatrix} 1 - A_{t,i} \\ A_{t,i} \end{bmatrix} \epsilon_{t,i} \right)^2 \Big| H_{t-1}^{(n)} \right]$$

$$= \sum_{t=1}^{T}\sum_{i=1}^{n} \mathbb{E}\left[ \frac{1}{n}\mathbf{c}_t^\top \begin{bmatrix} 1 - \pi_t^{(n)} & 0 \\ 0 & \pi_t^{(n)} \end{bmatrix}^{-1/2} \begin{bmatrix} 1 - A_{t,i} & 0 \\ 0 & A_{t,i} \end{bmatrix} \begin{bmatrix} 1 - \pi_t^{(n)} & 0 \\ 0 & \pi_t^{(n)} \end{bmatrix}^{-1/2} \mathbf{c}_t \epsilon_{t,i}^2 \Big| H_{t-1}^{(n)} \right]$$

$$= \sum_{t=1}^{T}\sum_{i=1}^{n} \frac{1}{n}\mathbf{c}_t^\top \begin{bmatrix} 1 - \pi_t^{(n)} & 0 \\ 0 & \pi_t^{(n)} \end{bmatrix}^{-1/2} \begin{bmatrix} \mathbb{E}[(1 - A_{t,i})\epsilon_{t,i}^2|H_{t-1}^{(n)}] & 0 \\ 0 & \mathbb{E}[A_{t,i}\epsilon_{t,i}^2|H_{t-1}^{(n)}] \end{bmatrix} \begin{bmatrix} 1 - \pi_t^{(n)} & 0 \\ 0 & \pi_t^{(n)} \end{bmatrix}^{-1/2} \mathbf{c}_t$$

Since $\mathbb{E}[A_{t,i}\epsilon_{t,i}^2|H_{t-1}^{(n)}] = \pi_t^{(n)}\mathbb{E}[\epsilon_{t,i}^2|H_{t-1}^{(n)}, A_{t,i} = 1] = \sigma^2 \pi_t$ and $\mathbb{E}[(1 - A_{t,i})\epsilon_{t,i}^2|H_{t-1}^{(n)}] = (1 - \pi_t)\mathbb{E}[\epsilon_{t,i}^2|H_{t-1}^{(n)}, A_{t,i} = 0] = \sigma^2(1 - \pi_t)$,

$$= \sum_{t=1}^{T}\sum_{i=1}^{n} n^{-1}\mathbf{c}_t^\top \mathbf{c}_t \sigma^2 = \sum_{t=1}^{T} \mathbf{c}_t^\top \mathbf{c}_t \sigma^2 = \sigma^2$$

**Condition(c): Lindeberg Condition**    Let $\delta > 0$.

$$\sum_{t=1}^{T}\sum_{i=1}^{n} E[Y_{t,i}^2 \mathbb{I}_{(Y_{t,i}^2 > \delta^2)}|H_{t-1}^{(n)}] = \sum_{t=1}^{T}\sum_{i=1}^{n} \mathbb{E}\left[ \left( n^{-1/2}\mathbf{c}_t^\top \begin{bmatrix} 1 - \pi_t^{(n)} & 0 \\ 0 & \pi_t^{(n)} \end{bmatrix}^{-1/2} \begin{bmatrix} 1 - A_{t,i} \\ A_{t,i} \end{bmatrix} \epsilon_{t,i} \right)^2 \mathbb{I}_{(Y_{t,i}^2 > \delta^2)} \Big| H_{t-1}^{(n)} \right]$$

$$= \sum_{t=1}^{T} \frac{1}{n}\sum_{i=1}^{n} \mathbb{E}\left[ \mathbf{c}_t^\top \begin{bmatrix} 1 - \pi_t^{(n)} & 0 \\ 0 & \pi_t^{(n)} \end{bmatrix}^{-1/2} \begin{bmatrix} 1 - A_{t,i} & 0 \\ 0 & A_{t,i} \end{bmatrix} \begin{bmatrix} 1 - \pi_t^{(n)} & 0 \\ 0 & \pi_t^{(n)} \end{bmatrix}^{-1/2} \mathbf{c}_t \epsilon_{t,i}^2 \mathbb{I}_{(Y_{t,i}^2 > \delta^2)} \Big| H_{t-1}^{(n)} \right]$$

$$= \sum_{t=1}^{T} \frac{1}{n}\sum_{i=1}^{n} \mathbf{c}_t^\top \begin{bmatrix} 1 - \pi_t^{(n)} & 0 \\ 0 & \pi_t^{(n)} \end{bmatrix}^{-\frac{1}{2}}$$

$$\begin{bmatrix} \mathbb{E}[(1 - A_{t,i})\epsilon_{t,i}^2 \mathbb{I}_{(Y_{t,i}^2 > \delta^2)}|H_{t-1}^{(n)}] & 0 \\ 0 & \mathbb{E}[A_{t,i}\epsilon_{t,i}^2 \mathbb{I}_{(Y_{t,i}^2 > \delta^2)}|H_{t-1}^{(n)}] \end{bmatrix}$$

$$\begin{bmatrix} 1 - \pi_t^{(n)} & 0 \\ 0 & \pi_t^{(n)} \end{bmatrix}^{-\frac{1}{2}} \mathbf{c}_t$$

Note that for $\mathbf{c}_t = [c_{t,0}, c_{t,1}]^\top$, $\mathbb{E}\big[(1 - A_{t,i})\epsilon_{t,i}^2 \mathbb{I}_{(Y_{t,i}^2 > \delta^2)}\big|H_{t-1}^{(n)}\big] = \mathbb{E}\Big[\epsilon_{t,i}^2 \mathbb{I}_{\big(\frac{c_{t,0}^2}{1-\pi_t^{(n)}}\epsilon_{t,i}^2 > n\delta^2\big)}\Big|H_{t-1}^{(n)}, A_{t,i} = 0\Big](1 - \pi_t)$ and $\mathbb{E}\big[A_{t,i}\epsilon_{t,i}^2 \mathbb{I}_{(Y_{t,i}^2 > \delta^2)}\big|H_{t-1}^{(n)}\big] = \mathbb{E}\Big[\epsilon_{t,i}^2 \mathbb{I}_{\big(\frac{c_{t,1}^2}{\pi_t^{(n)}}\epsilon_{t,i}^2 > n\delta^2\big)}\Big|H_{t-1}^{(n)}, A_{t,i} = 1\Big]\pi_t$. Thus, we have that

$$= \sum_{t=1}^{T} \frac{1}{n} \sum_{i=1}^{n} c_{t,0}^2 \mathbb{E}\Big[\epsilon_{t,i}^2 \mathbb{I}_{\big(\epsilon_{t,i}^2 > \frac{n\delta^2(1-\pi_t)}{c_{t,0}^2}\big)}\Big|H_{t-1}^{(n)}, A_{t,i} = 0\Big] + c_{t,1}^2 \mathbb{E}\Big[\epsilon_{t,i}^2 \mathbb{I}_{\big(\epsilon_{t,i}^2 > \frac{n\delta^2\pi_t^{(n)}}{c_{t,1}^2}\big)}\Big|H_{t-1}^{(n)}, A_{t,i} = 1\Big]$$

$$\leq \sum_{t=1}^{T} \max_{i \in [1:\, n]} \Big\{ c_{t,0}^2 \mathbb{E}\Big[\epsilon_{t,i}^2 \mathbb{I}_{\big(\epsilon_{t,i}^2 > \frac{n\delta^2(1-\pi_t)}{c_{t,0}^2}\big)}\Big|H_{t-1}^{(n)}, A_{t,i} = 0\Big] + c_{t,1}^2 \mathbb{E}\Big[\epsilon_{t,i}^2 \mathbb{I}_{\big(\epsilon_{t,i}^2 > \frac{n\delta^2\pi_t^{(n)}}{c_{t,1}^2}\big)}\Big|H_{t-1}^{(n)}, A_{t,i} = 1\Big]\Big\}$$

Note that for any $t \in [1:T]$ and $i \in [1:n]$,

$$\mathbb{E}\Big[\epsilon_{t,i}^2 \mathbb{I}_{\big(\epsilon_{t,i}^2 > \frac{n\delta^2\pi_t^{(n)}}{c_{t,1}^2}\big)}\Big|H_{t-1}^{(n)}, A_{t,i} = 1\Big]$$

$$= \mathbb{E}\Big[\epsilon_{t,i}^2 \mathbb{I}_{\big(\epsilon_{t,i}^2 > \frac{n\delta^2\pi_t^{(n)}}{c_{t,1}^2}\big)}\Big|H_{t-1}^{(n)}, A_{t,i} = 1\Big]\Big(\mathbb{I}_{(\pi_t^{(n)} \in [f(n),1-f(n)])} + \mathbb{I}_{(\pi_t^{(n)} \notin [f(n),1-f(n)])}\Big)$$

$$\leq \mathbb{E}\Big[\epsilon_{t,i}^2 \mathbb{I}_{\big(\epsilon_{t,i}^2 > \frac{n\delta^2 f(n)}{c_{t,1}^2}\big)}\Big|H_{t-1}^{(n)}, A_{t,i} = 1\Big] + \sigma^2 \mathbb{I}_{(\pi_t^{(n)} \notin [f(n),1-f(n)])}$$

The second term converges in probability to zero as $n \to \infty$ by our clipping assumption. We now show how the first term goes to zero in probability. Since we assume $f(n) = \omega(\frac{1}{n})$, $nf(n) \to \infty$. So, it is sufficient to show that for all $t, n$,

$$\lim_{m\to\infty} \max_{i \in [1:\, n]} \Big\{ \mathbb{E}\Big[\epsilon_{t,i}^2 \mathbb{I}_{(\epsilon_{t,i}^2 > m)}\Big|H_{t-1}^{(n)}, A_{t,i} = 1\Big]\Big\} = 0$$

By Condition 6, we have that for all $n \geq 1$,

$$\max_{t \in [1:\, T], i \in [1:\, n]} \mathbb{E}[\varphi(\epsilon_{t,i}^2)|H_{t-1}^{(n)}, A_{t,i} = 1] < M$$

Since we assume that $\lim_{x\to\infty} \frac{\varphi(x)}{x} = \infty$, for all $m$, there exists a $b_m$ s.t. $\varphi(x) \geq mMx$ for all $x \geq b_m$. So, for all $n, t, i$,

$$M \geq \mathbb{E}[\varphi(\epsilon_{t,i}^2)|H_{t-1}^{(n)}, A_{t,i} = 1] \geq \mathbb{E}[\varphi(\epsilon_{t,i}^2)\mathbb{I}_{(\epsilon_{t,i}^2 \geq b_m)}|H_{t-1}^{(n)}, A_{t,i} = 1]$$

$$\geq mM \mathbb{E}[\epsilon_{t,i}^2 \mathbb{I}_{(\epsilon_{t,i}^2 \geq b_m)}|H_{t-1}^{(n)}, A_{t,i} = 1]$$

Thus,

$$\max_{t \in [1:\, T], i \in [1:\, n]} \mathbb{E}[\epsilon_{t,i}^2 \mathbb{I}_{(\epsilon_{t,i}^2 \geq b_m)}|H_{t-1}^{(n)}, A_{t,i} = 1] \leq \frac{1}{m}$$

We can make a very similar argument that for all $t \in [1:T]$, as $n \to \infty$,

$$\max_{i \in [1:\, n]} \mathbb{E}\Big[\epsilon_{t,i}^2 \mathbb{I}_{\big(\epsilon_{t,i}^2 > \frac{n\delta^2(1-\pi_t)}{c_{t,0}^2}\big)}\Big|H_{t-1}^{(n)}, A_{t,i} = 0\Big] \xrightarrow{P} 0 \qquad \square$$

**Corollary 3** (Asymptotic Normality of the Batched OLS Estimator of Margin; two-arm bandit setting). *Assume the same conditions as Theorem 3. For each $t \in [1:T]$, we have the BOLS estimator of the margin $\beta_1 - \beta_0$:*

$$\hat{\Delta}_t^{\text{BOLS}} = \frac{\sum_{i=1}^{n}(1 - A_{t,i})R_{t,i}}{N_{t,0}} - \frac{\sum_{i=1}^{n} A_{t,i}R_{t,i}}{N_{t,1}}$$

*We show that as $n \to \infty$,*

$$\begin{bmatrix} \sqrt{\frac{N_{1,0}N_{1,1}}{n}}(\hat{\Delta}_1^{\text{BOLS}} - \Delta_1) \\ \sqrt{\frac{N_{2,0}N_{2,1}}{n}}(\hat{\Delta}_2^{\text{BOLS}} - \Delta_2) \\ \vdots \\ \sqrt{\frac{N_{T,0}N_{T,1}}{n}}(\hat{\Delta}_T^{\text{BOLS}} - \Delta_T) \end{bmatrix} \xrightarrow{D} \mathcal{N}(0, \sigma^2 \underline{\mathbf{I}}_T)$$

**Proof:**

$$\sqrt{\frac{N_{t,0}N_{t,1}}{n}}(\hat{\Delta}_t^{\text{BOLS}} - \Delta_t) = \sqrt{\frac{N_{t,0}N_{t,1}}{n}}\left(\frac{\sum_{i=1}^n (1-A_{t,i})\epsilon_{t,i}}{N_{t,0}} - \frac{\sum_{i=1}^n A_{t,i}\epsilon_{t,i}}{N_{t,1}}\right)$$

$$= \sqrt{\frac{N_{t,1}}{n}}\frac{\sum_{i=1}^n (1-A_{t,i})\epsilon_{t,i}}{\sqrt{N_{t,0}}} - \sqrt{\frac{N_{t,0}}{n}}\frac{\sum_{i=1}^n A_{t,i}\epsilon_{t,i}}{\sqrt{N_{t,1}}}$$

$$= \left[\sqrt{\frac{N_{t,1}}{n}} \quad -\sqrt{\frac{N_{t,0}}{n}}\right]\begin{bmatrix} N_{t,0} & 0 \\ 0 & N_{t,1} \end{bmatrix}^{-1/2}\sum_{i=1}^n \begin{bmatrix} 1-A_{t,i} \\ A_{t,i} \end{bmatrix}\epsilon_{t,i}$$

By Slutsky's Theorem and Lemma 1, it is sufficient to show that as $n \to \infty$,

$$\begin{bmatrix} \frac{1}{\sqrt{n}}\left[\sqrt{\pi_1^{(n)}} \quad -\sqrt{1-\pi_1^{(n)}}\right]\begin{bmatrix} 1-\pi_1^{(n)} & 0 \\ 0 & \pi_1^{(n)} \end{bmatrix}^{-1/2}\sum_{i=1}^n \begin{bmatrix} 1-A_{1,i} \\ A_{1,i} \end{bmatrix}\epsilon_{1,i} \\ \frac{1}{\sqrt{n}}\left[\sqrt{\pi_2^{(n)}} \quad -\sqrt{1-\pi_2^{(n)}}\right]\begin{bmatrix} 1-\pi_2^{(n)} & 0 \\ 0 & \pi_2^{(n)} \end{bmatrix}^{-1/2}\sum_{i=1}^n \begin{bmatrix} 1-A_{2,i} \\ A_{2,i} \end{bmatrix}\epsilon_{2,i} \\ \vdots \\ \frac{1}{\sqrt{n}}\left[\sqrt{\pi_t^{(n)}} \quad -\sqrt{1-\pi_t^{(n)}}\right]\begin{bmatrix} 1-\pi_t^{(n)} & 0 \\ 0 & \pi_t^{(n)} \end{bmatrix}^{-1/2}\sum_{i=1}^n \begin{bmatrix} 1-A_{T,i} \\ A_{T,i} \end{bmatrix}\epsilon_{T,i} \end{bmatrix} \xrightarrow{D} \mathcal{N}(0, \sigma^2 \mathbf{I}_T)$$

By Cramer-Wold device, it is sufficient to show that for any fixed vector $\mathbf{d} \in \mathbb{R}^T$ s.t. $\|\mathbf{d}\|_2 = 1$ that

$$\mathbf{d}^\top \begin{bmatrix} \frac{1}{\sqrt{n}}\left[\sqrt{\pi_1^{(n)}} \quad -\sqrt{1-\pi_1^{(n)}}\right]\begin{bmatrix} 1-\pi_1^{(n)} & 0 \\ 0 & \pi_1^{(n)} \end{bmatrix}^{-1/2}\sum_{i=1}^n \begin{bmatrix} 1-A_{1,i} \\ A_{1,i} \end{bmatrix}\epsilon_{1,i} \\ \frac{1}{\sqrt{n}}\left[\sqrt{\pi_2^{(n)}} \quad -\sqrt{1-\pi_2^{(n)}}\right]\begin{bmatrix} 1-\pi_2^{(n)} & 0 \\ 0 & \pi_2^{(n)} \end{bmatrix}^{-1/2}\sum_{i=1}^n \begin{bmatrix} 1-A_{2,i} \\ A_{2,i} \end{bmatrix}\epsilon_{2,i} \\ \vdots \\ \frac{1}{\sqrt{n}}\left[\sqrt{\pi_t^{(n)}} \quad -\sqrt{1-\pi_t^{(n)}}\right]\begin{bmatrix} 1-\pi_t^{(n)} & 0 \\ 0 & \pi_t^{(n)} \end{bmatrix}^{-1/2}\sum_{i=1}^n \begin{bmatrix} 1-A_{T,i} \\ A_{T,i} \end{bmatrix}\epsilon_{T,i} \end{bmatrix} \xrightarrow{D} \mathcal{N}(0, \sigma^2)$$

Let $[d_1, d_2, ..., d_T]^\top := \mathbf{d} \in \mathbb{R}^T$. The above is equivalent to

$$\sum_{t=1}^T \frac{1}{\sqrt{n}}d_t\left[\sqrt{\pi_t^{(n)}} \quad -\sqrt{1-\pi_t^{(n)}}\right]\begin{bmatrix} 1-\pi_t^{(n)} & 0 \\ 0 & \pi_t^{(n)} \end{bmatrix}^{-1/2}\sum_{i=1}^n \begin{bmatrix} 1-A_{t,i} \\ A_{t,i} \end{bmatrix}\epsilon_{t,i} \xrightarrow{D} \mathcal{N}(0, \sigma^2)$$

Define $Y_{t,i} := \frac{1}{\sqrt{n}}d_t\left[\sqrt{\pi_t^{(n)}} \quad -\sqrt{1-\pi_t^{(n)}}\right]\begin{bmatrix} 1-\pi_t^{(n)} & 0 \\ 0 & \pi_t^{(n)} \end{bmatrix}^{-1/2}\begin{bmatrix} 1-A_{t,i} \\ A_{t,i} \end{bmatrix}\epsilon_{t,i}.$

$\{Y_{1,1}, Y_{1,2}, ..., Y_{1,n}, ..., Y_{T,1}, Y_{T,2}, ..., Y_{T,n}\}$ is a martingale difference array with respect to the sequence of histories $\{H_t^{(n)}\}_{t=1}^T$ because for all $i \in [1:n]$ and $t \in [1:T]$,

$$\mathbb{E}[Y_{t,i}|H_{t-1}^{(n)}] = \frac{1}{\sqrt{n}}d_t\left[\sqrt{\pi_t^{(n)}} \quad -\sqrt{1-\pi_t^{(n)}}\right]\begin{bmatrix} 1-\pi_t^{(n)} & 0 \\ 0 & \pi_t^{(n)} \end{bmatrix}^{-1/2}\mathbb{E}\left[\begin{bmatrix} 1-A_{t,i} \\ A_{t,i} \end{bmatrix}\epsilon_{t,i}\Big|H_{t-1}^{(n)}\right]$$

$$= \frac{d_t}{\sqrt{n}}\left[\sqrt{\pi_t^{(n)}} \quad -\sqrt{1-\pi_t^{(n)}}\right]\begin{bmatrix} 1-\pi_t^{(n)} & 0 \\ 0 & \pi_t^{(n)} \end{bmatrix}^{-\frac{1}{2}}\mathbb{E}\left[\begin{bmatrix} (1-\pi_t^{(n)})\mathbb{E}[\epsilon_{t,i}|H_{t-1}^{(n)}, A_{t,i}=0] \\ \pi_{t,i}\mathbb{E}[\epsilon_{t,i}|H_{t-1}^{(n)}, A_{t,i}=1] \end{bmatrix}\Big|H_{t-1}^{(n)}\right] = 0$$

We now apply [8] martingale central limit theorem to $Y_{t,i}$ to show the desired result. Verifying the conditions for the martingale CLT is equivalent to what we did to verify the conditions in the conditions in the proof of Theorem 3—the only difference is that we replace $\mathbf{c}_t^\top$ in the Theorem 3 proof with $d_t\left[\sqrt{1-\pi_t^{(n)}} \quad -\sqrt{\pi_t^{(n)}}\right]$ in this proof. Even though $\mathbf{c}_t$ is a constant vector and $d_t\left[\sqrt{1-\pi_t^{(n)}} \quad -\sqrt{\pi_t^{(n)}}\right]$ is a random vector, the proof still goes through with this adjusted $\mathbf{c}_t$ vector, since (i) $d_t\left[\sqrt{1-\pi_t^{(n)}} \quad -\sqrt{\pi_t^{(n)}}\right] \in H_{t-1}^{(n)}$, (ii) $\|\left[\sqrt{1-\pi_t^{(n)}} \quad -\sqrt{\pi_t^{(n)}}\right]\|_2 = 1$, and (iii) $\frac{n\delta^2 \pi_t^{(n)}}{c_{t,1}^2} = \frac{n\delta^2 \pi_t^{(n)}}{d_t^2 \pi_t^{(n)}} \to \infty$ and $\frac{n\delta^2(1-\pi_t)}{c_{t,0}^2} = \frac{n\delta^2(1-\pi_t)}{d_t^2(1-\pi_t)} \to \infty$. $\square$

**Corollary 4** (Consistency of BOLS Variance Estimator). *Assuming Conditions 1 (moments) and 3 (conditionally i.i.d. actions), and a clipping rate of $f(n) = \omega(\frac{1}{n})$ (Definition 1), for all $t \in [1:T]$, as $n \to \infty$,*

$$\hat{\sigma}_t^2 = \frac{1}{n-2} \sum_{i=1}^{n} \left( R_{t,i} - A_{t,i}\hat{\beta}_{t,1}^{\text{BOLS}} - (1-A_{t,i})\hat{\beta}_{t,0}^{\text{BOLS}} \right)^2 \xrightarrow{P} \sigma^2$$

**Proof:**

$$\hat{\sigma}_t^2 = \frac{1}{n-2} \sum_{i=1}^{n} \left( R_{t,i} - A_{t,i}\hat{\beta}_{t,1}^{\text{BOLS}} - (1-A_{t,i})\hat{\beta}_{t,0}^{\text{BOLS}} \right)^2$$

$$= \frac{1}{n-2} \sum_{i=1}^{n} \left( \left[ A_{t,i}\beta_{t,1} + (1-A_{t,i})\beta_{t,0} + \epsilon_{t,i} \right] - A_{t,i}\left[ \beta_{t,1} + \frac{\sum_{i=1}^{n} A_{t,i}\epsilon_{t,i}}{N_{t,1}} \right] - (1-A_{t,i})\left[ \beta_{t,0} + \frac{\sum_{i=1}^{n} (1-A_{t,i})\epsilon_{t,i}}{N_{t,0}} \right] \right)^2$$

$$= \frac{1}{n-2} \sum_{i=1}^{n} \left( \epsilon_{t,i} - A_{t,i}\frac{\sum_{i=1}^{n} A_{t,i}\epsilon_{t,i}}{N_{t,1}} - (1-A_{t,i})\frac{\sum_{i=1}^{n}(1-A_{t,i})\epsilon_{t,i}}{N_{t,0}} \right)^2$$

$$= \frac{1}{n-2} \sum_{i=1}^{n} \left( \epsilon_{t,i}^2 - 2A_{t,i}\epsilon_{t,i}\frac{\sum_{i=1}^{n} A_{t,i}\epsilon_{t,i}}{N_{t,1}} - 2(1-A_{t,i})\epsilon_{t,i}\frac{\sum_{i=1}^{n}(1-A_{t,i})\epsilon_{t,i}}{N_{t,0}} \right.$$
$$\left. + A_{t,i}\left[ \frac{\sum_{i=1}^{n} A_{t,i}\epsilon_{t,i}}{N_{t,1}} \right]^2 + (1-A_{t,i})\left[ \frac{\sum_{i=1}^{n}(1-A_{t,i})\epsilon_{t,i}}{N_{t,0}} \right]^2 \right)$$

$$= \left( \frac{1}{n-2} \sum_{i=1}^{n} \epsilon_{t,i}^2 \right) - 2\frac{(\sum_{i=1}^{n} A_{t,i}\epsilon_{t,i})^2}{(n-2)N_{t,1}} - 2\frac{(\sum_{i=1}^{n}(1-A_{t,i})\epsilon_{t,i})^2}{(n-2)N_{t,0}}$$
$$+ \frac{N_{t,1}}{n-2}\left[ \frac{\sum_{i=1}^{n} A_{t,i}\epsilon_{t,i}}{N_{t,1}} \right]^2 + \frac{N_{t,0}}{n-2}\left[ \frac{\sum_{i=1}^{n}(1-A_{t,i})\epsilon_{t,i}}{N_{t,0}} \right]^2$$

$$= \left( \frac{1}{n-2} \sum_{i=1}^{n} \epsilon_{t,i}^2 \right) - \frac{(\sum_{i=1}^{n} A_{t,i}\epsilon_{t,i})^2}{(n-2)N_{t,1}} - \frac{(\sum_{i=1}^{n}(1-A_{t,i})\epsilon_{t,i})^2}{(n-2)N_{t,0}}$$

Note that $\frac{1}{n-2} \sum_{i=1}^{n} \epsilon_{t,i}^2 \xrightarrow{P} \sigma^2$ because for all $\delta > 0$,

$$\mathbb{P}\left( \left| \left[ \frac{1}{n-2} \sum_{i=1}^{n} \epsilon_{t,i}^2 \right] - \sigma^2 \right| > \delta \right) \leq \mathbb{P}\left( \left| \left[ \frac{1}{n-2} \sum_{i=1}^{n} \epsilon_{t,i}^2 \right] - \frac{\sigma^2(n-2)}{n} \right| > \delta/2 \right) + \mathbb{P}\left( \left| \frac{\sigma^2(n-2)}{n} - \sigma^2 \right| > \delta/2 \right)$$

$$= \mathbb{P}\left( \left| \frac{1}{n-2} \sum_{i=1}^{n} (\epsilon_{t,i}^2 - \sigma^2) \right| > \delta/2 \right) + \mathbb{P}\left( \sigma^2 \left| \frac{-2}{n} \right| > \delta/2 \right)$$

Since the second term in the summation above goes to zero for sufficiently large $n$, we now focus on the first term in the summation above. By Chebychev inequality,

$$\mathbb{P}\left( \left| \frac{1}{n-2} \sum_{i=1}^{n} (\epsilon_{t,i}^2 - \sigma^2) \right| > \delta/2 \right) \leq \frac{4}{\delta^2(n-2)^2} \mathbb{E}\left[ \sum_{i=1}^{n} \sum_{j=1}^{n} (\epsilon_{t,i}^2 - \sigma^2)(\epsilon_{t,j}^2 - \sigma^2) \right] = \frac{4}{\delta^2(n-2)^2} \mathbb{E}\left[ \sum_{i=1}^{n} (\epsilon_{t,i}^2 - \sigma^2)^2 \right]$$

where the equality above holds because for $i \neq j$, $\mathbb{E}[(\epsilon_{t,i}^2 - \sigma^2)(\epsilon_{t,j}^2 - \sigma^2)] = \mathbb{E}\left[ \mathbb{E}[(\epsilon_{t,i}^2 - \sigma^2)(\epsilon_{t,j}^2 - \sigma^2)|\mathcal{H}_{t-1}^{(n)}] \right] = \mathbb{E}\left[ \mathbb{E}[\epsilon_{t,i}^2 - \sigma^2|\mathcal{H}_{t-1}^{(n)}]\mathbb{E}[\epsilon_{t,j}^2 - \sigma^2|\mathcal{H}_{t-1}^{(n)}] \right] = 0$. By Condition 1 $\mathbb{E}[\epsilon_{t,i}^4|\mathcal{H}_{t=1}^{(n)}] < M < \infty$,

$$= \frac{4}{\delta^2(n-2)^2} \mathbb{E}\left[ \sum_{i=1}^{n} \mathbb{E}[(\epsilon_{t,i}^4 - 2\epsilon_{t,i}^2\sigma^2 + \sigma^4)|\mathcal{H}_{t-1}^{(n)}] \right] \leq \frac{4n(M + \sigma^4)}{\delta^2(n-2)^2} \to 0$$

Thus by Slutsky's Theorem it is sufficient to show that $\frac{(\sum_{i=1}^{n} A_{t,i}\epsilon_{t,i})^2}{(n-2)N_{t,1}} + \frac{(\sum_{i=1}^{n}(1-A_{t,i})\epsilon_{t,i})^2}{(n-2)N_{t,0}} \xrightarrow{P} 0$. We will only show that $\frac{(\sum_{i=1}^{n} A_{t,i}\epsilon_{t,i})^2}{(n-2)N_{t,1}} \xrightarrow{P} 0$; $\frac{(\sum_{i=1}^{n}(1-A_{t,i})\epsilon_{t,i})^2}{(n-2)N_{t,0}} \xrightarrow{P} 0$ holds by a very similar argument.

Note that by Lemma 1, $\frac{N_{t,1}}{n\pi_t^{(n)}} \xrightarrow{P} 1$. Thus, to show that $\frac{(\sum_{i=1}^n A_{t,i}\epsilon_{t,i})^2}{(n-2)N_{t,1}} \xrightarrow{P} 0$ by Slutsky's Theorem it is sufficient to show that $\frac{(\sum_{i=1}^n A_{t,i}\epsilon_{t,i})^2}{(n-2)n\pi_t^{(n)}} \xrightarrow{P} 0$. Let $\delta > 0$. By Markov inequality,

$$\mathbb{P}\left(\left|\frac{(\sum_{i=1}^n A_{t,i}\epsilon_{t,i})^2}{(n-2)n\pi_t^{(n)}}\right| > \delta\right) \leq \mathbb{E}\left[\frac{1}{\delta(n-2)n\pi_t^{(n)}}\left(\sum_{i=1}^n A_{t,i}\epsilon_{t,i}\right)^2\right] = \mathbb{E}\left[\frac{1}{\delta(n-2)n\pi_t^{(n)}}\sum_{j=1}^n\sum_{i=1}^n A_{t,j}A_{t,i}\epsilon_{t,i}\epsilon_{t,j}\right]$$

Since $\pi_t^{(n)} \in \mathcal{H}_{t-1}^{(n)}$,

$$= \mathbb{E}\left[\frac{1}{\delta(n-2)n\pi_t^{(n)}}\sum_{j=1}^n\sum_{i=1}^n \mathbb{E}[A_{t,j}A_{t,i}\epsilon_{t,i}\epsilon_{t,j}|\mathcal{H}_{t-1}^{(n)}]\right]$$

Since for $i \neq j$, $\mathbb{E}[A_{t,j}A_{t,i}\epsilon_{t,j}\epsilon_{t,i}|\mathcal{H}_{t-1}^{(n)}] = \mathbb{E}[A_{t,j}\epsilon_{t,j}|\mathcal{H}_{t-1}^{(n)}]\mathbb{E}[A_{t,i}\epsilon_{t,i}|\mathcal{H}_{t-1}^{(n)}] = 0$,

$$= \mathbb{E}\left[\frac{1}{\delta(n-2)n\pi_t^{(n)}}\sum_{i=1}^n \mathbb{E}[A_{t,i}\epsilon_{t,i}^2|\mathcal{H}_{t-1}^{(n)}]\right]$$

Since $\mathbb{E}[A_{t,i}\epsilon_{t,i}^2|\mathcal{H}_{t-1}^{(n)}] = \mathbb{E}[\epsilon_{t,i}^2|\mathcal{H}_{t-1}^{(n)}, A_{t,i} = 1]\pi_t^{(n)} = \sigma^2\pi_t^{(n)}$,

$$= \mathbb{E}\left[\frac{1}{\delta(n-2)n\pi_t^{(n)}}n\sigma^2\pi_t^{(n)}\right] = \frac{\sigma^2}{\delta(n-2)} \to 0 \qquad \square$$

# E  Asymptotic Normality of the Batched OLS Estimator: Contextual Bandits

**Theorem 4 (Asymptotic Normality of the Batched OLS Statistic)**  *For a $K$-armed contextual bandit, we for each $t \in [1\colon T]$, we have the BOLS estimator:*

$$
\hat{\boldsymbol{\beta}}_t^{\text{BOLS}} = \begin{bmatrix} \underline{\boldsymbol{C}}_{t,0} & \mathbf{0} & \mathbf{0} & \dots & \mathbf{0} \\ \mathbf{0} & \underline{\boldsymbol{C}}_{t,1} & \mathbf{0} & \dots & \mathbf{0} \\ \mathbf{0} & \mathbf{0} & \underline{\boldsymbol{C}}_{t,2} & \dots & \mathbf{0} \\ \vdots & \vdots & \vdots & \ddots & \vdots \\ \mathbf{0} & \mathbf{0} & \mathbf{0} & \dots & \underline{\boldsymbol{C}}_{t,K-1} \end{bmatrix}^{-1} \sum_{i=1}^{n} \begin{bmatrix} \mathbb{I}_{A_{t,i}=0}\boldsymbol{C}_{t,i} \\ \mathbb{I}_{A_{t,i}=1}\boldsymbol{C}_{t,i} \\ \vdots \\ \mathbb{I}_{A_{t,i}=K-1}\boldsymbol{C}_{t,i} \end{bmatrix} R_{t,i} \in \mathbb{R}^{Kd}
$$

*where $\underline{\boldsymbol{C}}_{t,k} := \sum_{i=1}^{n} \mathbb{I}_{A_{t,i}^{(n)}=k}\boldsymbol{C}_{t,i}(\boldsymbol{C}_{t,i})^{\top} \in \mathbb{R}^{d \times d}$. Assuming Conditions 6 (weak moments), 3 (conditionally i.i.d. actions), 4 (conditionally i.i.d. contexts), and 5 (bounded contexts), and a conditional clipping rate $f(n) = c$ for some $0 \le c < \frac{1}{2}$ (see Definition 2), we show that as $n \to \infty$,*

$$
\begin{bmatrix} \text{Diagonal}\big[\underline{\boldsymbol{C}}_{1,0}, \underline{\boldsymbol{C}}_{1,1}, ..., \underline{\boldsymbol{C}}_{1,K-1}\big]^{1/2}(\hat{\boldsymbol{\beta}}_1^{\text{BOLS}} - \boldsymbol{\beta}_1) \\ \text{Diagonal}\big[\underline{\boldsymbol{C}}_{2,0}, \underline{\boldsymbol{C}}_{2,1}, ..., \underline{\boldsymbol{C}}_{2,K-1}\big]^{1/2}(\hat{\boldsymbol{\beta}}_2^{\text{BOLS}} - \boldsymbol{\beta}_2) \\ \vdots \\ \text{Diagonal}\big[\underline{\boldsymbol{C}}_{T,0}, \underline{\boldsymbol{C}}_{T,1}, ..., \underline{\boldsymbol{C}}_{T,K-1}\big]^{1/2}(\hat{\boldsymbol{\beta}}_T^{\text{BOLS}} - \boldsymbol{\beta}_T) \end{bmatrix} \xrightarrow{D} \mathcal{N}(0, \sigma^2 \underline{\boldsymbol{I}}_{TKd})
$$

**Lemma 2.** *Assuming the conditions of Theorem 4, for any batch $t \in [1\colon T]$ and any arm $k \in [0\colon K-1]$, as $n \to \infty$,*

$$
\left[ \sum_{i=1}^{n} \mathbb{I}_{A_{t,i}=k}\boldsymbol{C}_{t,i}\boldsymbol{C}_{t,i}^{\top} \right] \left[ n\underline{\boldsymbol{Z}}_{t,k} P_{t,k} \right]^{-1} \xrightarrow{P} \underline{\boldsymbol{I}}_d \tag{22}
$$

$$
\left[ \sum_{i=1}^{n} \mathbb{I}_{A_{t,i}=k}\boldsymbol{C}_{t,i}\boldsymbol{C}_{t,i}^{\top} \right]^{1/2} \left[ n\underline{\boldsymbol{Z}}_{t,k} P_{t,k} \right]^{-1/2} \xrightarrow{P} \underline{\boldsymbol{I}}_d \tag{23}
$$

*where $P_{t,k} := \mathbb{P}(A_{t,i} = k | H_{t-1}^{(n)})$ and $\underline{\boldsymbol{Z}}_{t,k} := \mathbb{E}\big[\boldsymbol{C}_{t,i}\boldsymbol{C}_{t,i}^{\top} | H_{t-1}^{(n)}, A_{t,i} = k\big]$.*

**Proof of Lemma 2:**  We first show that as $n \to \infty$, $\frac{1}{n} \sum_{i=1}^{n} \big( \mathbb{I}_{A_{t,i}=k}\boldsymbol{C}_{t,i}\boldsymbol{C}_{t,i}^{\top} - \underline{\boldsymbol{Z}}_{t,k} P_{t,k} \big) \xrightarrow{P} \underline{\mathbf{0}}$. It is sufficient to show that convergence holds entry-wise so for any $r, s \in [0\colon d-1]$, as $n \to \infty$, $\frac{1}{n} \sum_{i=1}^{n} \big( \mathbb{I}_{A_{t,i}=k}\boldsymbol{C}_{t,i}\boldsymbol{C}_{t,i}^{\top}(r,s) - P_{t,k}\underline{\boldsymbol{Z}}_{t,k}(r,s) \big) \xrightarrow{P} 0$. Note that

$$
\mathbb{E}\Big[ \mathbb{I}_{A_{t,i}=k}\boldsymbol{C}_{t,i}\boldsymbol{C}_{t,i}^{\top}(r,s) - P_{t,k}\underline{\boldsymbol{Z}}_{t,k}(r,s) \Big] = \mathbb{E}\Big[ \mathbb{E}\big[\boldsymbol{C}_{t,i}\boldsymbol{C}_{t,i}^{\top}(r,s) | H_{t-1}, A_{t,i} = k \big] P_{t,k} - P_{t,k}\underline{\boldsymbol{Z}}_{t,k}(r,s) \Big] = 0
$$

By Chebychev inequality, for any $\epsilon > 0$,

$$
\mathbb{P}\left( \left| \frac{1}{n} \sum_{i=1}^{n} \mathbb{I}_{A_{t,i}=k}\boldsymbol{C}_{t,i}\boldsymbol{C}_{t,i}^{\top}(r,s) - P_{t,k}\underline{\boldsymbol{Z}}_{t,k}(r,s) \right| > \epsilon \right) \le \frac{1}{\epsilon^2 n^2} \mathbb{E}\left[ \left( \sum_{i=1}^{n} \mathbb{I}_{A_{t,i}=k}\boldsymbol{C}_{t,i}\boldsymbol{C}_{t,i}^{\top} - P_{t,k}\underline{\boldsymbol{Z}}_{t,k}(r,s) \right)^2 \right]
$$

$$
= \frac{1}{\epsilon^2 n^2} \sum_{i=1}^{n}\sum_{j=1}^{n} \mathbb{E}\Big[ \big[ \mathbb{I}_{A_{t,i}=k}\boldsymbol{C}_{t,i}\boldsymbol{C}_{t,i}^{\top}(r,s) - P_{t,k}\underline{\boldsymbol{Z}}_{t,k}(r,s) \big] \big[ \mathbb{I}_{A_{t,i}=k}\boldsymbol{C}_{t,j}\boldsymbol{C}_{t,j}^{\top}(r,s) - P_{t,k}\underline{\boldsymbol{Z}}_{t,k}(r,s) \big] \Big] \tag{24}
$$

By conditional independence and by law of iterated expectations (conditioning on $H_{t-1}^{(n)}$), for $i \ne j$, $\mathbb{E}\big[ \big(\mathbb{I}_{A_{t,i}=k}\boldsymbol{C}_{t,i}\boldsymbol{C}_{t,i}^{\top}(r,s) - P_{t,k}\underline{\boldsymbol{Z}}_{t,k}(r,s)\big)\big(\mathbb{I}_{A_{t,j}=k}\boldsymbol{C}_{t,j}\boldsymbol{C}_{t,j}^{\top}(r,s) - P_{t,k}\underline{\boldsymbol{Z}}_{t,k}(r,s)\big) \big] = 0$. Thus, (24) above equals the following:

$$
= \frac{1}{\epsilon^2 n^2} \sum_{i=1}^{n} \mathbb{E}\left[ \left( \mathbb{I}_{A_{t,i}=k}\boldsymbol{C}_{t,i}\boldsymbol{C}_{t,i}^{\top}(r,s) - P_{t,k}\underline{\boldsymbol{Z}}_{t,k}(r,s) \right)^2 \right]
$$

$$
= \frac{1}{\epsilon^2 n^2} \sum_{i=1}^{n} \mathbb{E}\left[ \mathbb{I}_{A_{t,i}=k}\big[\boldsymbol{C}_{t,i}\boldsymbol{C}_{t,i}^{\top}(r,s)\big]^2 - 2\mathbb{I}_{A_{t,i}=k}\boldsymbol{C}_{t,i}\boldsymbol{C}_{t,i}^{\top}(r,s)P_{t,k}\underline{\boldsymbol{Z}}_{t,k}(r,s) + P_{t,k}^2\big[\underline{\boldsymbol{Z}}_{t,k}(r,s)\big]^2 \right]
$$

$$= \frac{1}{\epsilon^2 n^2} \sum_{i=1}^{n} \mathbb{E}\left[ \mathbb{I}_{A_{t,i}=k} \big[ \mathbf{C}_{t,i} \mathbf{C}_{t,i}^{\top}(r,s) \big]^2 - P_{t,k}^2 \big[ \mathbf{Z}_{t,k}(r,s) \big]^2 \right]$$

$$= \frac{1}{\epsilon^2 n} \mathbb{E}\left[ \mathbb{I}_{A_{t,i}=k} \big[ \mathbf{C}_{t,i} \mathbf{C}_{t,i}^{\top}(r,s) \big]^2 - P_{t,k}^2 \big[ \mathbf{Z}_{t,k}(r,s) \big]^2 \right] \leq \frac{2d \max(u^2, 1)}{\epsilon^2 n} \to 0$$

as $n \to \infty$. The last inequality above holds by Condition 5.

**Proving Equation (22):**

It is sufficient to show that

$$\left\| \frac{2 \max(du^2, 1)}{\epsilon^2 n} \big[ n \mathbf{Z}_{t,k} P_{t,k} \big]^{-1} \right\|_{op} = \left\| \frac{2 \max(du^2, 1)}{\epsilon^2 n^2 P_{t,k}} \mathbf{Z}_{t,k}^{-1} \right\|_{op} \xrightarrow{P} 0 \tag{25}$$

We define random variable $M_t^{(n)} = \mathbb{I}_{(\forall \, \mathbf{c} \in \mathbb{R}^d, \, \mathcal{A}_t(H_{t-1}^{(n)}, \mathbf{c}) \in [f(n), 1-f(n)]^K)}$, representing whether the conditional clipping condition is satisfied. Note that by our conditional clipping assumption, $M_t^{(n)} \xrightarrow{P} 1$ as $n \to \infty$. The left hand side of (25) is equal to the following

$$\left\| \frac{2 \max(du^2, 1)}{\epsilon^2 n^2 P_{t,k}} \mathbf{Z}_{t,k}^{-1} (M_t^{(n)} + (1 - M_t^{(n)})) \right\|_{op} = \left\| \frac{2 \max(du^2, 1)}{\epsilon^2 n^2 P_{t,k}} \mathbf{Z}_{t,k}^{-1} M_t^{(n)} \right\|_{op} + o_p(1) \tag{26}$$

By our conditional clipping condition and Bayes rule we have that for all $\mathbf{c} \in [-u, u]^d$,

$$\mathbb{P}(\mathbf{C}_{t,i} = \mathbf{c} | A_{t,i} = k, H_{t-1}^{(n)}, M_t^{(n)} = 1)$$

$$= \frac{\mathbb{P}(A_{t,i} = k | \mathbf{C}_{t,i} = \mathbf{c}, H_{t-1}^{(n)}, M_t^{(n)} = 1) \mathbb{P}(\mathbf{C}_{t,i} = \mathbf{c} | H_{t-1}^{(n)}, M_t^{(n)} = 1)}{\mathbb{P}(A_{t,i} = k | H_{t-1}^{(n)}, M_t^{(n)} = 1)}$$

$$\geq \frac{f(n) \, \mathbb{P}(\mathbf{C}_{t,i} = \mathbf{c} | H_{t-1}^{(n)}, M_t^{(n)} = 1)}{1}.$$

Thus, we have that

$$\mathbf{Z}_{t,k} M_t^{(n)} = \mathbb{E}\big[ \mathbf{C}_{t,i} \mathbf{C}_{t,i}^{\top} \big| H_{t-1}^{(n)}, A_{t,i} = k \big] M_t^{(n)} = \mathbb{E}\big[ \mathbf{C}_{t,i} \mathbf{C}_{t,i}^{\top} \big| H_{t-1}^{(n)}, A_{t,i} = k, M_t^{(n)} = 1 \big] M_t^{(n)}$$

$$\succcurlyeq f(n) \mathbb{E}\big[ \mathbf{C}_{t,i} \mathbf{C}_{t,i}^{\top} \big| H_{t-1}^{(n)}, M_t^{(n)} = 1 \big] M_t^{(n)} = f(n) \mathbb{E}\big[ \mathbf{C}_{t,i} \mathbf{C}_{t,i}^{\top} \big| H_{t-1}^{(n)} \big] M_t^{(n)} = f(n) \mathbf{\Sigma}_t^{(n)} M_t^{(n)}.$$

By apply matrix inverses to both sides of the above inequality, we get that

$$\lambda_{\max}(\mathbf{Z}_{t,k}^{-1} M_t^{(n)}) \leq \frac{1}{f(n)} \lambda_{\max}\left( (\mathbf{\Sigma}_t^{(n)})^{-1} \right) M_t^{(n)} \leq \frac{1}{l \, f(n)} \tag{27}$$

where the last inequality above holds for constant $l$ by Condition 5. Recall that $P_{t,k} = \mathbb{P}(A_{t,i} = k \mid H_{t-1}^{(n)})$, so $P_{t,k} \mid (M_t^{(n)} = 1) \geq f(n)$. Thus, equation (26) is bounded above by the following

$$\leq \frac{2 \max(du^2, 1)}{\epsilon^2 n^2 l f(n)^2} + o_p(1) \xrightarrow{P} 0$$

where the limit above holds because we assume that $f(n) = c$ for some $0 < c \leq \frac{1}{2}$. $\quad \square$

**Proving Equation (23):** By Condition 5, $\|\frac{1}{n} \mathbf{C}_{t,k}\|_{\max} \leq u$ and $\|\mathbf{Z}_{t,k} P_{t,k}\|_{\max} \leq u$. Thus, any continuous function of $\frac{1}{n} \mathbf{C}_{t,k}$ and $\mathbf{Z}_{t,k} P_{t,k}$ will have compact support and thus be uniformly continuous. For any uniformly continuous function $f : \mathbb{R}^{d \times d} \to \mathbb{R}^{d \times d}$, for any $\epsilon > 0$, there exists a $\delta > 0$ such that for any matrices $\mathbf{A}, \mathbf{B} \in \mathbb{R}^{d \times d}$, whenever $\|\mathbf{A} - \mathbf{B}\|_{op} < \delta$, then $\|f(\mathbf{A}) - f(\mathbf{B})\|_{op} < \epsilon$. Thus, for any $\epsilon > 0$, there exists some $\delta > 0$ such that

$$\mathbb{P}\left( \left\| \left( \frac{1}{n} \sum_{i=1}^{n} \mathbb{I}_{(A_{t,k}=k)} \mathbf{C}_{t,i} \mathbf{C}_{t,i}^{\top} \right) - \mathbf{Z}_{t,k} P_{t,k} \right\|_{op} > \delta \right) \to 0$$

implies

$$\mathbb{P}\left( \left\| f\left( \frac{1}{n} \sum_{i=1}^{n} \mathbb{I}_{(A_{t,k}=k)} \mathbf{C}_{t,i} \mathbf{C}_{t,i}^{\top} \right) - f(\mathbf{Z}_{t,k} P_{t,k}) \right\|_{op} > \epsilon \right) \to 0$$

Thus, by letting $f$ be the matrix square-root function,

$$\left(\frac{1}{n}\sum_{i=1}^{n}\mathbb{I}_{(A_{t,k}=k)}\mathbf{C}_{t,i}\mathbf{C}_{t,i}^{\top}\right)^{1/2} - (\underline{\mathbf{Z}}_{t,k}P_{t,k})^{1/2} \overset{P}{\to} \underline{\mathbf{0}}.$$

We now want to show that for some constant $r > 0$, $\mathbb{P}\big(\big\|\underline{\mathbf{Z}}_{t,k}^{-1}\frac{1}{P_{t,k}}\big\|_{\mathrm{op}} > r\big)$, because this would imply that

$$\left[\left(\frac{1}{n}\sum_{i=1}^{n}\mathbb{I}_{(A_{t,k}=k)}\mathbf{C}_{t,i}\mathbf{C}_{t,i}^{\top}\right)^{1/2} - (\underline{\mathbf{Z}}_{t,k}P_{t,k})^{1/2}\right](\underline{\mathbf{Z}}_{t,k}P_{t,k})^{-1/2} \overset{P}{\to} \underline{\mathbf{0}}.$$

Recall that for $M_t^{(n)} = \mathbb{I}_{(\forall\, \mathbf{c}\in\mathbb{R}^d,\, \mathcal{A}_t(H_{t-1}^{(n)},\mathbf{c})\in[f(n),1-f(n)]^K)}$, representing whether the conditional clipping condition is satisfied,

$$\underline{\mathbf{Z}}_{t,k}^{-1} = \underline{\mathbf{Z}}_{t,k}^{-1}(M_t^{(n)} + (1 - M_t^{(n)})) = \underline{\mathbf{Z}}_{t,k}^{-1}M_t^{(n)} + o_p(1).$$

Thus it is sufficient to show that $\mathbb{P}\big(\big\|\underline{\mathbf{Z}}_{t,k}^{-1}\frac{1}{P_{t,k}}M_t^{(n)}\big\|_{\mathrm{op}} > r\big)$. Recall that by equation (27) we have that

$$\lambda_{\max}(\underline{\mathbf{Z}}_{t,k}^{-1}M_t^{(n)}) \leq \frac{1}{f(n)}\lambda_{\max}\left(\left(\mathbf{\Sigma}_t^{(n)}\right)^{-1}\right)M_t^{(n)} \leq \frac{1}{l\,f(n)}$$

Also note that $P_{t,k} = \mathbb{P}(A_{t,i} = k \mid H_{t-1}^{(n)})$, so $P_{t,k} \mid (M_t^{(n)} = 1) \geq f(n)$. Thus we have that

$$\mathbb{P}\left(\left\|\underline{\mathbf{Z}}_{t,k}^{-1}\frac{1}{P_{t,k}}M_t^{(n)}\right\|_{\mathrm{op}} > r\right) \leq \mathbb{I}_{(\frac{1}{l\,f(n)^2}>r)} = 0$$

for $r > \frac{1}{l\,f(n)^2} = \frac{1}{lc^2}$, since we assume that $f(n) = c$ for some $0 < c \leq \frac{1}{2}$. $\quad\square$

**Proof of Theorem 4:** We define $P_{t,k} := \mathbb{P}(A_{t,i} = k|H_{t-1}^{(n)})$ and $\underline{\mathbf{Z}}_{t,k} := \mathbb{E}\big[\mathbf{C}_{t,i}\mathbf{C}_{t,i}^{\top}\big|H_{t-1}^{(n)}, A_{t,i} = k\big]$. We also define

$$\mathbf{D}_t^{(n)} := \mathrm{Diagonal}\big[\underline{\mathbf{C}}_{t,0}, \underline{\mathbf{C}}_{t,1}, ..., \underline{\mathbf{C}}_{t,K-1}\big]^{1/2}(\hat{\beta}_t - \beta_t) = \sum_{i=1}^{n}\begin{bmatrix}\underline{\mathbf{C}}_{t,0}^{-1/2}\,\mathbf{C}_{t,i}\mathbb{I}_{A_{t,i}=0} \\ \underline{\mathbf{C}}_{t,1}^{-1/2}\,\mathbf{C}_{t,i}\mathbb{I}_{A_{t,i}=1} \\ \vdots \\ \underline{\mathbf{C}}_{t,K-1}^{-1/2}\,\mathbf{C}_{t,i}\mathbb{I}_{A_{t,i}=K-1}\end{bmatrix}\epsilon_{t,i}$$

We want to show that $[\mathbf{D}_1^{(n)}, \mathbf{D}_2^{(n)}, ..., \mathbf{D}_T^{(n)}]^{\top} \overset{D}{\to} \mathcal{N}(\mathbf{0}, \sigma^2\underline{\mathbf{I}}_{TKd})$. By Lemma 2 and Slutsky's Theorem, it sufficient to show that as $n \to \infty$, $[\mathbf{Q}_1^{(n)}, \mathbf{Q}_2^{(n)}, ..., \mathbf{Q}_T^{(n)}]^{\top} \overset{D}{\to} \mathcal{N}(\mathbf{0}, \sigma^2\underline{\mathbf{I}}_{TKd})$ for

$$\mathbf{Q}_t^{(n)} := \sum_{i=1}^{n}\begin{bmatrix}\frac{1}{\sqrt{nP_{t,0}}}\mathbf{Z}_{t,0}^{-1/2}\mathbf{C}_{t,i}\mathbb{I}_{A_{t,i}=0} \\ \frac{1}{\sqrt{nP_{t,1}}}\mathbf{Z}_{t,1}^{-1/2}\mathbf{C}_{t,i}\mathbb{I}_{A_{t,i}=1} \\ \vdots \\ \frac{1}{\sqrt{nP_{t,K-1}}}\mathbf{Z}_{t,K-1}^{-1/2}\mathbf{C}_{t,i}\mathbb{I}_{A_{t,i}=K-1}\end{bmatrix}\epsilon_{t,i}$$

By Cramer Wold device, it is sufficient to show that for any $\mathbf{b} \in \mathbb{R}^{TKd}$ with $\|\mathbf{b}\|_2 = 1$, where $\mathbf{b} = [\mathbf{b}_1, \mathbf{b}_2, ..., \mathbf{b}_T]$ for $\mathbf{b}_t \in \mathbb{R}^{Kd}$, as $n \to \infty$.

$$\sum_{t=1}^{T}\mathbf{b}_t^{\top}\mathbf{Q}_t^{(n)} \overset{D}{\to} \mathcal{N}(0, \sigma^2) \tag{28}$$

We can further define for all $t \in [1\colon T]$, $\mathbf{b}_t = [\mathbf{b}_{t,0}, \mathbf{b}_{t,1}, ..., \mathbf{b}_{t,K-1}]$ with $\mathbf{b}_{t,k} \in \mathbb{R}^d$. Thus to show (28) it is equivalent to show that

$$\sum_{t=1}^{T}\sum_{k=0}^{K-1}\mathbf{b}_{t,k}^{\top}\frac{1}{\sqrt{nP_{t,k}}}\underline{\mathbf{Z}}_{t,k}^{-1/2}\sum_{i=1}^{n}\mathbb{I}_{A_{t,i}=k}\mathbf{C}_{t,i}\epsilon_{t,i} \overset{D}{\to} \mathcal{N}(0, \sigma^2)$$

We define $Y_{t,i}^{(n)} := \sum_{k=0}^{K-1} \mathbf{b}_{t,k}^\top \frac{1}{\sqrt{nP_{t,k}}} \mathbb{I}_{A_{t,i}=k} \underline{\mathbf{Z}}_{t,k}^{-1/2} \mathbf{C}_{t,i} \epsilon_{t,i}$. The sequence $Y_{1,1}^{(n)}, Y_{1,2}^{(n)}, ..., Y_{1,n}^{(n)}, ... Y_{T,1}^{(n)}, Y_{T,2}^{(n)}, ..., Y_{T,n}^{(n)}$ is a martingale difference array with respect to the sequence of histories $\{H_{t-1}^{(n)}\}_{t=1}^T$ because $\mathbb{E}[Y_{t,i}^{(n)}|H_{t-1}^{(n)}] = \mathbb{E}\left[\mathbb{E}[Y_{t,i}^{(n)}|H_{t-1}^{(n)}, A_{t,i}, \mathbf{C}_{t,i}]\Big|H_{t-1}^{(n)}\right] = 0$ for all $i \in [1\colon n]$ and all $t \in [1\colon T]$. We then apply the martingale central limit theorem of [8] to $Y_{t,i}^{(n)}$ to show the desired result (see the proof of Theorem 5 in Appendix B for the statement of the martingale CLT conditions). Note that the first condition (a) of the martingale CLT is already satisfied, as we just showed that $Y_{t,i}^{(n)}$ form a martingale difference array with respect to $H_{t-1}^{(n)}$.

**Condition(b): Conditional Variance**

$$\sum_{t=1}^T \sum_{i=1}^n \mathbb{E}[Y_{t,i}^2|H_{t-1}^{(n)}] = \sum_{t=1}^T \sum_{i=1}^n \mathbb{E}\left[\left(\sum_{k=0}^{K-1} \mathbf{b}_{t,k}^\top \frac{1}{\sqrt{nP_{t,k}}} \mathbf{Z}_{t,k}^{-1/2} \mathbb{I}_{A_{t,i}=k} \mathbf{C}_{t,i}\epsilon_{t,i}\right)^2 \Big|H_{t-1}^{(n)}\right]$$

$$= \sum_{t=1}^T \sum_{i=1}^n \sum_{k=0}^{K-1} \frac{1}{nP_{t,k}} \mathbf{b}_{t,k}^\top \underline{\mathbf{Z}}_{t,k}^{-1/2} \mathbb{E}\left[\mathbb{I}_{A_{t,i}=k}\mathbf{C}_{t,i}\mathbf{C}_{t,i}^\top \epsilon_{t,i}^2\Big|H_{t-1}^{(n)}\right] \underline{\mathbf{Z}}_{t,k}^{-1/2} \mathbf{b}_{t,k}$$

By law of iterated expectations (conditioning on $H_{t-1}^{(n)}, A_{t,i}, \mathbf{C}_{t,i}$) and Condition 6,

$$= \frac{1}{n}\sum_{t=1}^T \sum_{i=1}^n \sum_{k=0}^{K-1} \frac{1}{P_{t,k}} \mathbf{b}_{t,k}^\top \underline{\mathbf{Z}}_{t,k}^{-1/2} \mathbb{E}\left[\mathbb{I}_{A_{t,i}=k}\mathbf{C}_{t,i}\mathbf{C}_{t,i}^\top \Big|H_{t-1}^{(n)}\right] \underline{\mathbf{Z}}_{t,k}^{-1/2} \mathbf{b}_{t,k}\sigma^2$$

$$= \frac{1}{n}\sum_{t=1}^T \sum_{i=1}^n \sum_{k=0}^{K-1} \frac{1}{P_{t,k}} \mathbf{b}_{t,k}^\top \underline{\mathbf{Z}}_{t,k}^{-1/2} \mathbb{E}\left[\mathbf{C}_{t,i}\mathbf{C}_{t,i}^\top \Big|H_{t-1}^{(n)}, A_{t,i}=k\right] P_{t,k}\underline{\mathbf{Z}}_{t,k}^{-1/2} \mathbf{b}_{t,k}\sigma^2$$

$$= \frac{1}{n}\sum_{t=1}^T \sum_{i=1}^n \sum_{k=0}^{K-1} \mathbf{b}_{t,k}^\top \underline{\mathbf{I}}_d \mathbf{b}_{t,k}\sigma^2 = \sigma^2 \sum_{t=1}^T \sum_{k=0}^{K-1} \mathbf{b}_{t,k}^\top \mathbf{b}_{t,k} = \sigma^2$$

**Condition(c): Lindeberg Condition**   Let $\delta > 0$.

$$\sum_{t=1}^T \sum_{i=1}^n \mathbb{E}\left[Y_{t,i}^2\mathbb{I}_{(|Y_{t,i}|>\delta)}\big|H_{t-1}^{(n)}\right] = \sum_{t=1}^T \sum_{i=1}^n \mathbb{E}\left[\left(\sum_{k=0}^{K-1} \mathbf{b}_{t,k}^\top \frac{1}{\sqrt{nP_{t,k}}} \mathbf{Z}_{t,i}^{-1/2}\mathbb{I}_{A_{t,i}=k}\mathbf{C}_{t,i}\epsilon_{t,i}\right)^2 \mathbb{I}_{(Y_{t,i}^2>\delta^2)}\Big|H_{t-1}^{(n)}\right]$$

$$= \sum_{t=1}^T \sum_{i=1}^n \sum_{k=0}^{K-1} \frac{1}{nP_{t,k}} \mathbf{b}_{t,k}^\top \mathbf{Z}_{t,i}^{-1/2} \mathbb{E}\left[\mathbb{I}_{A_{t,i}=k}\mathbf{C}_{t,i}\mathbf{C}_{t,i}^\top \epsilon_{t,i}^2\mathbb{I}_{(Y_{t,i}^2>\delta^2)}\Big|H_{t-1}^{(n)}\right] \mathbf{Z}_{t,i}^{-1/2}\mathbf{b}_{t,k}$$

It is sufficient to show that for any $t \in [1\colon T]$ and any $k \in [0\colon K-1]$ the following converges in probability to zero:

$$\sum_{i=1}^n \frac{1}{nP_{t,k}} \mathbf{b}_{t,k}^\top \mathbf{Z}_{t,i}^{-1/2} \mathbb{E}\left[\mathbb{I}_{A_{t,i}=k}\mathbf{C}_{t,i}\mathbf{C}_{t,i}^\top \epsilon_{t,i}^2\mathbb{I}_{(Y_{t,i}^2>\delta^2)}\Big|H_{t-1}^{(n)}\right] \mathbf{Z}_{t,i}^{-1/2}\mathbf{b}_{t,k}$$

Recall that $Y_{t,i} = \sum_{k=0}^{K-1} \mathbf{b}_{t,k}^\top \frac{1}{\sqrt{nP_{t,k}}} \mathbb{I}_{A_{t,i}=k}\underline{\mathbf{Z}}_{t,k}^{-1/2}\mathbf{C}_{t,i}\epsilon_{t,i}$.

$$= \frac{1}{n}\sum_{i=1}^n \mathbf{b}_{t,k}^\top \mathbf{Z}_{t,i}^{-1/2}\mathbb{E}\left[\mathbf{C}_{t,i}\mathbf{C}_{t,i}^\top \epsilon_{t,i}^2\mathbb{I}_{(\frac{1}{nP_{t,k}}\mathbf{b}_{t,k}^\top \underline{\mathbf{Z}}_{t,k}^{-1/2}\mathbf{C}_{t,i}\mathbf{C}_{t,i}^\top \underline{\mathbf{Z}}_{t,k}^{-1/2}\mathbf{b}_{t,k}\epsilon_{t,i}^2>\delta^2)}\Big|H_{t-1}^{(n)}, A_{t,i}=k\right] \mathbf{Z}_{t,i}^{-1/2}\mathbf{b}_{t,k}$$

Since $\mathbf{c} \in [-u, u]$, by the Gershgorin circle theorem, we can bound the maximum eigenvalue of $\mathbf{c}\mathbf{c}^\top$ by some constant $a > 0$.

$$\leq \frac{a}{n}\sum_{i=1}^n \mathbf{b}_{t,k}^\top \mathbf{Z}_{t,i}^{-1}\mathbf{b}_{t,k}\mathbb{E}\left[\epsilon_{t,i}^2\mathbb{I}_{(\frac{a}{nP_{t,k}}\mathbf{b}_{t,k}^\top \mathbf{Z}_{t,i}^{-1}\mathbf{b}_{t,k}\epsilon_{t,i}^2>\delta^2)}\Big| H_{t-1}^{(n)}, A_{t,i}=k\right]$$

We define random variable $M_t^{(n)} = \mathbb{I}_{(\forall\, \mathbf{c}\in\mathbb{R}^d,\, \mathcal{A}_t(H_{t-1}^{(n)},\mathbf{c})\in[f(n),1-f(n)]^K)}$, representing whether the conditional clipping condition is satisfied. Note that by our conditional clipping assumption, $M_t^{(n)} \xrightarrow{P} 1$ as $n \to \infty$.

$$= \frac{a}{n}\sum_{i=1}^{n}\mathbf{b}_{t,k}^{\top}\underline{\mathbf{Z}}_{t,k}^{-1}\mathbf{b}_{t,k}\mathbb{E}\left[\epsilon_{t,i}^2\mathbb{I}_{(\frac{a}{nP_{t,k}}\mathbf{b}_{t,k}^{\top}\underline{\mathbf{Z}}_{t,k}^{-1}\mathbf{b}_{t,k}\epsilon_{t,i}^2 > \delta^2)}\Big| H_{t-1}^{(n)}, A_{t,i}=k\right]\left(M_t^{(n)}+(1-M_t^{(n)})\right)$$

$$= \frac{a}{n}\sum_{i=1}^{n}\mathbf{b}_{t,k}^{\top}\underline{\mathbf{Z}}_{t,k}^{-1}\mathbf{b}_{t,k}\mathbb{E}\left[\epsilon_{t,i}^2\mathbb{I}_{(\frac{a}{nP_{t,k}}\mathbf{b}_{t,k}^{\top}\underline{\mathbf{Z}}_{t,k}^{-1}\mathbf{b}_{t,k}\epsilon_{t,i}^2 > \delta^2)}\Big| H_{t-1}^{(n)}, A_{t,i}=k\right]M_t^{(n)}+o_p(1) \qquad (29)$$

By equation (27), have that

$$\lambda_{\max}(\underline{\mathbf{Z}}_{t,k}^{-1}) \leq \frac{1}{f(n)}\lambda_{\max}\left((\underline{\mathbf{\Sigma}}_t^{(n)})^{-1}\right) \leq \frac{1}{l\,f(n)}$$

Recall that $P_{t,k} = \mathbb{P}(A_{t,i}=k \mid H_{t-1}^{(n)})$, so $P_{t,k}\mid(M_t^{(n)}=1)\geq f(n)$. Thus we have that equation (29) is upper bounded by the following:

$$\leq \frac{1}{n}\sum_{i=1}^{n}\frac{\mathbf{b}_{t,k}^{\top}\mathbf{b}_{t,k}}{l\,f(n)}\mathbb{E}\left[\epsilon_{t,i}^2\mathbb{I}_{(\frac{a}{nf(n)}\frac{\mathbf{b}_{t,k}^{\top}\mathbf{b}_{t,k}}{l\,f(n)}\epsilon_{t,i}^2 > \delta^2)}\Big| H_{t-1}^{(n)}, A_{t,i}=k\right]+o_p(1)$$

$$= \frac{1}{n}\sum_{i=1}^{n}\frac{\mathbf{b}_{t,k}^{\top}\mathbf{b}_{t,k}}{l\,f(n)}\mathbb{E}\left[\epsilon_{t,i}^2\mathbb{I}_{(\epsilon_{t,i}^2 > \delta^2\frac{l\,nf(n)^2}{a\mathbf{b}_{t,k}^{\top}\mathbf{b}_{t,k}})}\Big| H_{t-1}^{(n)}, A_{t,i}=k\right]+o_p(1)$$

It is sufficient to show that

$$\lim_{n\to\infty}\max_{i\in[1:\,n]}\frac{1}{f(n)}\mathbb{E}\left[\epsilon_{t,i}^2\mathbb{I}_{(\epsilon_{t,i}^2 > \delta^2\frac{l\,nf(n)^2}{a\mathbf{b}_{t,k}^{\top}\mathbf{b}_{t,k}})}\Big| H_{t-1}^{(n)}, A_{t,i}=k\right]=0. \qquad (30)$$

By Condition 6, we have that for all $n \geq 1$, $\max_{t\in[1:\,T],i\in[1:\,n]}\mathbb{E}[\varphi(\epsilon_{t,i}^2)|H_{t-1}^{(n)}, A_{t,i}=k] < M$.

Since we assume that $\lim_{x\to\infty}\frac{\varphi(x)}{x}=\infty$, for all $m \geq 1$, there exists a $b_m$ s.t. $\varphi(x) \geq mMx$ for all $x \geq b_m$. So, for all $n,t,i$,

$$M \geq \mathbb{E}[\varphi(\epsilon_{t,i}^2)|H_{t-1}^{(n)}, A_{t,i}=k] \geq \mathbb{E}[\varphi(\epsilon_{t,i}^2)\mathbb{I}_{(\epsilon_{t,i}^2\geq b_m)}|H_{t-1}^{(n)}, A_{t,i}=k]$$

$$\geq mM\mathbb{E}[\epsilon_{t,i}^2\mathbb{I}_{(\epsilon_{t,i}^2\geq b_m)}|H_{t-1}^{(n)}, A_{t,i}=k]$$

Thus, $\max_{i\in[1:\,n]}\mathbb{E}[\epsilon_{t,i}^2\mathbb{I}_{(\epsilon_{t,i}^2\geq b_m)}|H_{t-1}^{(n)}, A_{t,i}=k] \leq \frac{1}{m}$; so $\lim_{m\to\infty}\max_{i\in[1:\,n]}\mathbb{E}[\epsilon_{t,i}^2\mathbb{I}_{(\epsilon_{t,i}^2\geq b_m)}|H_{t-1}^{(n)}, A_{t,i}=k]=0$. Since by our conditional clipping assumption, $f(n)=c$ for some $0 < c \leq \frac{1}{2}$ thus $nf(n)^2 \to \infty$. So equation (30) holds. $\square$

**Corollary 5** (Asymptotic Normality of the Batched OLS for Margin with Context Statistic). *Assume the same conditions as Theorem 4. For any two arms $x, y \in [0\colon K-1]$ for all $t \in [1\colon T]$, we have the BOLS estimator for $\boldsymbol{\Delta}_{t,x-y} := \boldsymbol{\beta}_{t,x} - \boldsymbol{\beta}_{t,y}$. We show that as $n \to \infty$,*

$$
\begin{bmatrix}
\left[\underline{\mathbf{C}}_{1,x}^{-1} + \underline{\mathbf{C}}_{1,y}^{-1}\right]^{1/2} (\hat{\boldsymbol{\Delta}}_{1,x-y}^{\text{BOLS}} - \boldsymbol{\Delta}_{1,x-y}) \\
\left[\underline{\mathbf{C}}_{2,x}^{-1} + \underline{\mathbf{C}}_{2,y}^{-1}\right]^{1/2} (\hat{\boldsymbol{\Delta}}_{2,x-y}^{\text{BOLS}} - \boldsymbol{\Delta}_{2,x-y}) \\
\vdots \\
\left[\underline{\mathbf{C}}_{T,x}^{-1} + \underline{\mathbf{C}}_{T,y}^{-1}\right]^{1/2} (\hat{\boldsymbol{\Delta}}_{T,x-y}^{\text{BOLS}} - \boldsymbol{\Delta}_{T,x-y})
\end{bmatrix}
\xrightarrow{D} \mathcal{N}(0, \sigma^2 \underline{\mathbf{I}}_{Td})
$$

*where*

$$
\hat{\boldsymbol{\Delta}}_{t,x-y}^{\text{BOLS}} = \left[\mathbf{C}_{t,x}^{-1} + \underline{\mathbf{C}}_{t,y}^{-1}\right]^{-1} \left(\underline{\mathbf{C}}_{t,y}^{-1} \sum_{i=1}^{n} A_{t,i} \mathbf{C}_{t,i} R_{t,i} - \underline{\mathbf{C}}_{t,x}^{-1} \sum_{i=1}^{n} (1 - A_{t,i}) \mathbf{C}_{t,i} R_{t,i}\right).
$$

**Proof:** By Cramer-Wold device, it is sufficient to show that for any fixed vector $\mathbf{d} \in \mathbb{R}^{Td}$ s.t. $\|\mathbf{d}\|_2 = 1$, where $\mathbf{d} = [\mathbf{d}_1, \mathbf{d}_2, ..., \mathbf{d}_T]$ for $\mathbf{d}_t \in \mathbb{R}^d$, $\sum_{t=1}^{T} \mathbf{d}_t^\top \left[\mathbf{C}_{t,x}^{-1} + \underline{\mathbf{C}}_{t,y}^{-1}\right]^{1/2} (\hat{\boldsymbol{\Delta}}_{t,x-y}^{\text{BOLS}} - \boldsymbol{\Delta}_{t,x-y}) \xrightarrow{D} \mathcal{N}(0, \sigma^2)$, as $n \to \infty$.

$$
\sum_{t=1}^{T} \mathbf{d}_t^\top \left[\underline{\mathbf{C}}_{t,x}^{-1} + \underline{\mathbf{C}}_{t,y}^{-1}\right]^{1/2} (\hat{\boldsymbol{\Delta}}_{t,x-y}^{\text{BOLS}} - \boldsymbol{\Delta}_{t,x-y})
$$

$$
= \sum_{t=1}^{T} \mathbf{d}_t^\top \left[\underline{\mathbf{C}}_{t,x}^{-1} + \underline{\mathbf{C}}_{t,y}^{-1}\right]^{-1/2} \left(\underline{\mathbf{C}}_{t,y}^{-1} \sum_{i=1}^{n} A_{t,i} \mathbf{C}_{t,i} \epsilon_{t,i} - \underline{\mathbf{C}}_{t,x}^{-1} \sum_{i=1}^{n} (1 - A_{t,i}) \mathbf{C}_{t,i} \epsilon_{t,i}\right)
$$

By Lemma 2, as $n \to \infty$, $\frac{1}{nP_{t,x}} \mathbf{Z}_{t,x}^{-1} \underline{\mathbf{C}}_{t,x} \xrightarrow{P} \underline{\mathbf{I}}_d$ and $\frac{1}{nP_{t,y}} \mathbf{Z}_{t,y}^{-1} \underline{\mathbf{C}}_{t,y} \xrightarrow{P} \underline{\mathbf{I}}_d$, so by Slutsky's Theorem it is sufficient to that as $n \to \infty$,

$$
\sum_{t=1}^{T} \mathbf{d}_t^\top \left[\underline{\mathbf{C}}_{t,x}^{-1} + \underline{\mathbf{C}}_{t,y}^{-1}\right]^{-1/2} \left(\frac{1}{nP_{t,y}} \mathbf{Z}_{t,y}^{-1} \sum_{i=1}^{n} A_{t,i} \mathbf{C}_{t,i} \epsilon_{t,i} - \frac{1}{nP_{t,x}} \mathbf{Z}_{t,x}^{-1} \sum_{i=1}^{n} (1 - A_{t,i}) \mathbf{C}_{t,i} \epsilon_{t,i}\right) \xrightarrow{D} \mathcal{N}(0, \sigma^2)
$$

We know that $\left[\frac{1}{P_{t,x}} \mathbf{Z}_{t,x}^{-1} + \frac{1}{P_{t,y}} \mathbf{Z}_{t,y}^{-1}\right]^{-1/2} \left[\frac{1}{P_{t,x}} \mathbf{Z}_{t,x}^{-1} + \frac{1}{P_{t,y}} \mathbf{Z}_{t,y}^{-1}\right]^{1/2} \xrightarrow{P} \underline{\mathbf{I}}_d$.

By Lemma 2 and continuous mapping theorem, $nP_{t,x} \mathbf{Z}_{t,x} \underline{\mathbf{C}}_{t,x}^{-1} \xrightarrow{P} \underline{\mathbf{I}}_d$ and $nP_{t,y} \mathbf{Z}_{t,y} \underline{\mathbf{C}}_{t,y}^{-1} \xrightarrow{P} \underline{\mathbf{I}}_d$. So by Slutsky's Theorem,

$$
\left[\frac{1}{P_{t,x}} \mathbf{Z}_{t,x}^{-1} + \frac{1}{P_{t,y}} \mathbf{Z}_{t,y}^{-1}\right]^{-1/2} \left[n\underline{\mathbf{C}}_{t,x}^{-1} + n\underline{\mathbf{C}}_{t,y}^{-1}\right]^{1/2} \xrightarrow{P} \underline{\mathbf{I}}_d
$$

So, returning to our CLT, by Slutsky's Theorem, it is sufficient to show that as $n \to \infty$,

$$
\sum_{t=1}^{T} \mathbf{d}_t^\top \left[\frac{1}{nP_{t,x}} \mathbf{Z}_{t,x}^{-1} + \frac{1}{nP_{t,y}} \mathbf{Z}_{t,y}^{-1}\right]^{-1/2} \frac{1}{nP_{t,y}} \mathbf{Z}_{t,y}^{-1} \sum_{i=1}^{n} A_{t,i} \mathbf{C}_{t,i} \epsilon_{t,i}
$$

$$
- \sum_{t=1}^{T} \mathbf{d}_t^\top \left[\frac{1}{nP_{t,x}} \mathbf{Z}_{t,x}^{-1} + \frac{1}{nP_{t,y}} \mathbf{Z}_{t,y}^{-1}\right]^{-1/2} \frac{1}{nP_{t,x}} \mathbf{Z}_{t,x}^{-1} \sum_{i=1}^{n} (1 - A_{t,i}) \mathbf{C}_{t,i} \epsilon_{t,i} \xrightarrow{D} \mathcal{N}(0, \sigma^2)
$$

The above sum equals the following:

$$
= \sum_{t=1}^{T} \mathbf{d}_t^\top \left[\frac{1}{nP_{t,x}} \mathbf{Z}_{t,x}^{-1} + \frac{1}{nP_{t,y}} \mathbf{Z}_{t,y}^{-1}\right]^{-1/2} \frac{1}{\sqrt{nP_{t,x}}} \mathbf{Z}_{t,x}^{-1/2} \left(\frac{1}{\sqrt{nP_{t,x}}} \mathbf{Z}_{t,x}^{-1/2} \sum_{i=1}^{n} A_{t,i} \mathbf{C}_{t,i} \epsilon_{t,i}\right)
$$

$$
- \sum_{t=1}^{T} \mathbf{d}_t^\top \left[\frac{1}{nP_{t,x}} \mathbf{Z}_{t,x}^{-1} + \frac{1}{\sqrt{nP_{t,x}}} \mathbf{Z}_{t,y}^{-1}\right]^{-1/2} \frac{1}{\sqrt{nP_{t,y}}} \mathbf{Z}_{t,y}^{-1/2} \left(\frac{1}{\sqrt{nP_{t,y}}} \mathbf{Z}_{t,y}^{-1/2} \sum_{i=1}^{n} (1 - A_{t,i}) \mathbf{C}_{t,i} \epsilon_{t,i}\right)
$$

Asymptotic normality holds by the same martingale CLT as we used in the proof of Theorem 4. The only difference is that we adjust our $\mathbf{b}_{t,k}$ vector from Theorem 4 to the following:

$$
\mathbf{b}_{t,k} := \begin{cases} \mathbf{0} & \text{if } k \notin \{x, y\} \\ \mathbf{d}_t^\top \left[ \frac{1}{nP_{t,x}} \mathbf{Z}_{t,x}^{-1} + \frac{1}{nP_{t,y}} \mathbf{Z}_{t,y}^{-1} \right]^{-1/2} \frac{1}{\sqrt{nP_{t,x}}} \mathbf{Z}_{t,x}^{-1/2} & \text{if } k = x \\ \mathbf{d}_t^\top \left[ \frac{1}{nP_{t,x}} \mathbf{Z}_{t,x}^{-1} + \frac{1}{nP_{t,y}} \mathbf{Z}_{t,y}^{-1} \right]^{-1/2} \frac{1}{\sqrt{nP_{t,y}}} \mathbf{Z}_{t,y}^{-1/2} & \text{if } k = y \end{cases}
$$

The proof still goes through with this adjustment because for all $k \in [0:K-1]$, (i) $\mathbf{b}_{t,k} \in H_{t-1}^{(n)}$, (ii) $\sum_{t=1}^{T} \sum_{k=0}^{K-1} \mathbf{b}_{t,k}^\top \mathbf{b}_{t,k} = \sum_{t=1}^{T} \mathbf{d}_t^\top \mathbf{d}_t = 1$. and (iii) $\frac{l \, nf(n)^2}{a\mathbf{b}_{t,k}^\top \mathbf{b}_{t,k}} \to \infty$ still holds because $\mathbf{b}_{t,k}^\top \mathbf{b}_{t,k}$ is bounded above by one. $\square$

# F    W-Decorrelated Estimator [6]

To better understand why the W-decorrelated estimator has relatively low power, but is still able to guarantee asymptotic normality, we now investigate the form of the W-decorrelated estimator in the two-arm bandit setting.

## F.1    Decorrelation Approach

We now assume we are in the unbatched setting (i.e., batch size of one), as the W-decorrelated estimator was developed for this setting; however, these results easily translate to the batched setting. We now let $n$ index the number of samples total (previously this was $nT$) and examine asymptotics as $n \to \infty$. We assume the following model:

$$\mathbf{R}_n = \underline{\mathbf{X}}_n^\top \boldsymbol{\beta} + \boldsymbol{\epsilon}_n$$

where $\mathbf{R}_n, \boldsymbol{\epsilon}_n \in \mathbb{R}^n$ and $\underline{\mathbf{X}}_n \in \mathbb{R}^{n \times p}$ and $\boldsymbol{\beta} \in \mathbb{R}^p$. The W-decorrelated OLS estimator is defined as follows:

$$\hat{\boldsymbol{\beta}}^d = \hat{\boldsymbol{\beta}}_{\text{OLS}} + \underline{\mathbf{W}}_n(\mathbf{R}_n - \underline{\mathbf{X}}_n\hat{\boldsymbol{\beta}}_{\text{OLS}})$$

With this definition we have that,

$$\hat{\boldsymbol{\beta}}^d - \boldsymbol{\beta} = \hat{\boldsymbol{\beta}}_{\text{OLS}} + \underline{\mathbf{W}}_n(\mathbf{R}_n - \underline{\mathbf{X}}_n\hat{\boldsymbol{\beta}}_{\text{OLS}}) - \boldsymbol{\beta}$$
$$= \hat{\boldsymbol{\beta}}_{\text{OLS}} + \underline{\mathbf{W}}_n(\underline{\mathbf{X}}_n\boldsymbol{\beta} + \boldsymbol{\epsilon}_n) - \underline{\mathbf{W}}_n\underline{\mathbf{X}}_n\hat{\boldsymbol{\beta}}_{\text{OLS}} - \boldsymbol{\beta}$$
$$= (\mathbf{I}_p - \underline{\mathbf{W}}_n\underline{\mathbf{X}}_n)(\hat{\boldsymbol{\beta}}_{\text{OLS}} - \boldsymbol{\beta}) + \underline{\mathbf{W}}_n\boldsymbol{\epsilon}_n$$

Note that if $\mathbb{E}[\underline{\mathbf{W}}_n\boldsymbol{\epsilon}_n] = \mathbb{E}\left[\sum_{i=1}^n \mathbf{W}_i\epsilon_i\right] = 0$ (where $\mathbf{W}_i$ is the $i^{th}$ column of $\underline{\mathbf{W}}_n$), then $\mathbb{E}[(\mathbf{I}_p - \underline{\mathbf{W}}_n\underline{\mathbf{X}}_n)(\hat{\boldsymbol{\beta}}_{\text{OLS}} - \boldsymbol{\beta})]$ would be the bias of the estimator. We assume $\{\epsilon_i\}$ is a martingale difference sequence w.r.t. filtration $\{\mathcal{G}_i\}_{i=1}^n$. Thus, if we constrain $\mathbf{W}_i$ to be $\mathcal{G}_{i-1}$ measurable,

$$\mathbb{E}[\underline{\mathbf{W}}_n\boldsymbol{\epsilon}_n] = \mathbb{E}\left[\sum_{i=1}^n \mathbf{W}_i\epsilon_i\right] = \sum_{i=1}^n \mathbb{E}\left[\mathbb{E}[\mathbf{W}_i\epsilon_i|\mathcal{G}_{i-1}]\right] = \sum_{i=1}^n \mathbb{E}\left[\mathbf{W}_i\mathbb{E}[\epsilon_i|\mathcal{G}_{i-1}]\right] = 0$$

**Trading off Bias and Variance**    While decreasing $\mathbb{E}[(\mathbf{I}_p - \underline{\mathbf{W}}_n\underline{\mathbf{X}}_n)(\hat{\boldsymbol{\beta}}_{\text{OLS}} - \boldsymbol{\beta})]$ will decrease the bias, making $\underline{\mathbf{W}}_n$ larger in norm will increase the variance. So the trade-off between bias and variance can be adjusted with different values of $\lambda$ for the following optimization problem:

$$\|\mathbf{I}_p - \underline{\mathbf{W}}_n\underline{\mathbf{X}}_n\|_F^2 + \lambda\|\underline{\mathbf{W}}_n\|_F^2 = \|\mathbf{I}_p - \underline{\mathbf{W}}_n\underline{\mathbf{X}}_n\|_F^2 + \lambda\text{Tr}(\underline{\mathbf{W}}_n\underline{\mathbf{W}}_n^\top)$$

**Optimizing for $\underline{\mathbf{W}}_n$**    The authors propose to optimize for $\underline{\mathbf{W}}_n$ in a recursive fashion, so that the $i^{th}$ column, $\mathbf{W}_i$, only depends on $\{\mathbf{X}_j\}_{j \leq i} \cup \{\epsilon_j\}_{j \leq i-1}$ (so $\sum_{i=1}^n \mathbb{E}[\mathbf{W}_i\epsilon_i] = 0$). We let $\mathbf{W}_0 = 0$, $\mathbf{X}_0 = 0$, and recursively define $\underline{\mathbf{W}}_n := [\underline{\mathbf{W}}_{n-1} \mathbf{W}_n]$ where

$$\mathbf{W}_n = \text{argmin}_{\mathbf{W} \in \mathbb{R}^p}\|\mathbf{I}_p - \underline{\mathbf{W}}_{n-1}\underline{\mathbf{X}}_{n-1} - \mathbf{W}\mathbf{X}_n^\top\|_F^2 + \lambda\|\mathbf{W}\|_2^2$$

where $\underline{\mathbf{W}}_{n-1} = [\mathbf{W}_1; \mathbf{W}_2; ...; \mathbf{W}_{n-1}] \in \mathbb{R}^{p \times (n-1)}$ and $\underline{\mathbf{X}}_{n-1} = [\mathbf{X}_1; \mathbf{X}_2; ...; \mathbf{X}_{n-1}]^\top \in \mathbb{R}^{(n-1) \times p}$. Now, let us find the closed form solution for each step of this minimization:

$$\frac{d}{d\mathbf{W}}\|\mathbf{I}_p - \underline{\mathbf{W}}_{n-1}\underline{\mathbf{X}}_{n-1} - \mathbf{W}\mathbf{X}_n^\top\|_F^2 + \lambda\|\mathbf{W}\|_2^2 = 2(\mathbf{I}_p - \underline{\mathbf{W}}_{n-1}\underline{\mathbf{X}}_{n-1} - \mathbf{W}\mathbf{X}_n^\top)(-\mathbf{X}_n) + 2\lambda\mathbf{W}$$

Note that since the Hessian is positive definite, so we can find the minimizing $\mathbf{W}$ by setting the first derivative to 0:

$$\frac{d^2}{d\mathbf{W}d\mathbf{W}^\top}\|\mathbf{I}_p - \underline{\mathbf{W}}_{n-1}\underline{\mathbf{X}}_{n-1} - \mathbf{W}\mathbf{X}_n^\top\|_F^2 + \lambda\|\mathbf{W}\|_2^2 = 2\mathbf{X}_n\mathbf{X}_n^\top + 2\lambda\mathbf{I}_p \succcurlyeq 0$$

$$0 = 2(\mathbf{I}_p - \underline{\mathbf{W}}_{n-1}\underline{\mathbf{X}}_{n-1} - \mathbf{W}\mathbf{X}_n^\top)(-\mathbf{X}_n) + 2\lambda\mathbf{W}$$
$$(\mathbf{I}_p - \underline{\mathbf{W}}_{n-1}\underline{\mathbf{X}}_{n-1} - \mathbf{W}\mathbf{X}_n^\top)\mathbf{X}_n = \lambda\mathbf{W}$$
$$(\mathbf{I}_p - \underline{\mathbf{W}}_{n-1}\underline{\mathbf{X}}_{n-1})\mathbf{X}_n = \lambda\mathbf{W} + \mathbf{W}\mathbf{X}_n^\top\mathbf{X}_n = (\lambda + \|\mathbf{X}_n\|_2^2)\mathbf{W}$$
$$\mathbf{W}^* = (\mathbf{I}_p - \underline{\mathbf{W}}_{n-1}\underline{\mathbf{X}}_{n-1})\frac{\mathbf{X}_n}{\lambda + \|\mathbf{X}_n\|_2^2}$$

**Proposition 3** (W-decorrelated estimator and time discounting in the two-arm bandit setting). *Suppose we have a 2-arm bandit. $A_i$ is an indicator that equals 1 if arm 1 is chosen for the $i^{th}$ sample, and 0 if arm 0 is chosen. We define $\boldsymbol{X}_i := [1 - A_i, A_i] \in \mathbb{R}^2$. We assume the following model of rewards:*

$$R_i = \boldsymbol{X}_i^\top \boldsymbol{\beta} + \epsilon_i = A_i \beta_1 + (1 - A_i)\beta_0 + \epsilon_i$$

*We further assume that $\{\epsilon_i\}_{i=1}^n$ are a martingale difference sequence with respect to filtration $\{\mathcal{G}_i\}_{i=1}^n$. We also assume that $\boldsymbol{X}_i$ are non-anticipating with respect to filtration $\{\mathcal{G}_i\}_{i=1}^n$. Note the W-decorrelated estimator:*

$$\hat{\boldsymbol{\beta}}^d = \hat{\boldsymbol{\beta}}_{\text{OLS}} + \underline{\boldsymbol{W}}_n(\boldsymbol{R}_n - \underline{\boldsymbol{X}}_n \hat{\boldsymbol{\beta}}_{\text{OLS}})$$

*We show that for $\underline{\boldsymbol{W}}_n = [\boldsymbol{W}_1; \boldsymbol{W}_2; ...; \boldsymbol{W}_n] \in \mathbb{R}^{p \times n}$ and choice of constant $\lambda$,*

$$\boldsymbol{W}_i = \begin{bmatrix} (1 - \frac{1}{\lambda+1})^{\sum_{i=1}^n (1-A_i)} \frac{1}{\lambda+1} \\ (1 - \frac{1}{\lambda+1})^{\sum_{i=1}^n A_i} \frac{1}{\lambda+1} \end{bmatrix} \in \mathbb{R}^2$$

*Moreover, we show that the W-decorrelated estimator for the mean of arm 1, $\beta_1$, is as follows:*

$$\hat{\beta}_1^d = \left( 1 - \sum_{i=1}^n A_t \frac{1}{\lambda+1} \left( 1 - \frac{1}{\lambda+1} \right)^{N_{1,i}-1} \right) \hat{\beta}_1^{\text{OLS}} + \sum_{i=1}^n A_t R_t \cdot \frac{1}{\lambda+1} \left( 1 - \frac{1}{\lambda+1} \right)^{N_{1,i}-1}$$

*where $\hat{\beta}_1^{\text{OLS}} = \frac{\sum_{i=1}^n A_i R_i}{N_{1,n}}$ for $N_{1,n} = \sum_{i=1}^n A_i$. Since [6] require that $\lambda \geq 1$ for their CLT results to hold, thus, the W-decorrelated estimators is down-weighting samples drawn later on in the study and up-weighting earlier samples.*

**Proof:** Recall the formula for $\boldsymbol{W}_i$,

$$\boldsymbol{W}_i = (\mathbf{I}_p - \underline{\boldsymbol{W}}_{i-1}\underline{\mathbf{X}}_{i-1}) \frac{\mathbf{X}_i}{\lambda + \|\mathbf{X}_i\|_2^2}$$

We let $\boldsymbol{W}_i = [W_{0,i}, W_{1,i}]^\top$. For notational simplicity, we let $r = \frac{1}{\lambda+1}$. We now solve for $W_{1,n}$:

$$W_{1,1} = (1 - 0) \cdot rA_1 = rA_1$$

$$W_{1,2} = (1 - W_{1,1} \cdot A_1) \cdot rA_2 = (1 - rA_1)rA_2$$

$$W_{1,3} = \left( 1 - \sum_{i=1}^2 \mathbf{W}_{1,i} \cdot A_i \right) \cdot rA_3 = \left( 1 - rA_1 - (1-rA_1)rA_2 \right) \cdot rA_3 = (1-rA_1)(1-rA_2) \cdot rA_3$$

$$W_{1,4} = \left( 1 - \sum_{i=1}^3 \mathbf{W}_{1,i} \cdot A_i \right) \cdot rA_4 = \left( 1 - rA_1 - (1-rA_1)rA_2 - (1-rA_1)(1-rA_2) \cdot rA_3 \right) \cdot rA_4$$

$$= (1 - rA_1)\left( 1 - rA_2 - (1 - rA_2)rA_3 \right) \cdot rA_4 = (1 - rA_1)(1 - rA_2)(1 - rA_3) \cdot rA_4$$

We have that for arbitrary $n$,

$$W_{1,n} = \left( 1 - \sum_{i=1}^{n-1} \mathbf{W}_{1,i} \cdot A_i \right) \cdot rA_n = rA_n \prod_{i=1}^{n-1}(1 - rA_i) = rA_n(1-r)^{\sum_{i=1}^{n-1} A_i} = rA_n(1-r)^{N_{1,n-1}}$$

By symmetry, we have that

$$W_{0,n} = \left( 1 - \sum_{i=1}^{n-1} \mathbf{W}_{1,i} \cdot (1 - A_i) \right) \cdot r(1 - A_n) = r(1 - A_n)(1 - r)^{N_{0,n-1}}$$

Note the W-decorrelated estimator for $\beta_1$:

$$\hat{\beta}_1^d = \hat{\beta}_1^{\text{OLS}} + \sum_{i=1}^n A_i \left( R_i - \hat{\beta}_1^{\text{OLS}} \right) r(1 - r)^{N_{1,i-1}}$$

$$= \left( 1 - \sum_{i=1}^n A_i r(1 - r)^{N_{1,i-1}} \right) \hat{\beta}_1^{\text{OLS}} + \sum_{i=1}^n A_i R_i \cdot r(1 - r)^{N_{1,i-1}} \quad \square$$