[Reviews · NeurIPS 2020]

Review 1

Summary and Contributions: This paper studies the problem of inference from adaptively collected data, and in particular, from bandit policy. The authors first show that on data adaptively collected by a bandit policy, whether the natural ordinary least squares (OLS) estimator is asymptotically normal or not depends on whether margin is zero or not. The authors show that this is indeed true for many well-known bandit policies, such as Thompson Sampling, \epsilon-greedy, and Upper Confidence Bound. Having this motivation in mind, the authors study batched OLS estimators and show that they are normal, regardless of whether the margin is zero or not. The authors also discuss the extension of their work to the contextual bandits and non-stationary settings.

Strengths: I believe the paper tackles a very important problem and I agree that it is possible to list many applications of statistical inference on data collected adaptively by a bandit policy (or any other adaptive policy). The authors prove a result on non-asymptotic normality of OLS estimator when margin is zero, which has been observed in the prior work of Deshpande et. al. (reference [5] of the paper). Among other results stated in the paper, I particularly like this one. I also like the large range of simulations provided in the paper. I also like the way that the paper is organized and the insights that are provided.

Weaknesses: While I think the paper has a very strong motivation, I do not think that the batched model considered addresses the problem of inference on bandit data, discussed in the introduction. As authors state, the main challenge in studying data collected from bandit policies, is the adaptivity of samples collected, which creates an undesired dependency across the samples. When one assumes that the policy is fixed in one batch (that from theoretical perspective, should be very large) conditional on the history, many of the standard techniques for proving asymptotic normality of OLS estimators can be used. This can be observed from the statement of Theorem 3 that says the estimators derived from each of these T batches are normal (and also uncorrelated). So, it appears to me that the theoretical contribution, at least for the batched OLS, is very limited and could have been improved. I would be more interested to learn how to combine the estimators from all these T batches to get an asymptotically normal estimator when n (or number of samples within each batch) does not go to infinity. However, when combining weaknesses and strength, I think the weaknesses outweigh the strengths. If the authors can address my main concern above, I would be happy to change my score.

Correctness: The claims appear to be correct and the empirical methodology is sound.

Clarity: The paper is very well written. It is easy to go through the paper.

Relation to Prior Work: Yes. Indeed, I really like that the authors go through implementation of other algorithms and also show some results on W-decorrelated and AW-AIPW estimators.

Reproducibility: Yes

Additional Feedback: See above. In particular, if BOLS estimator wants to be used in practice, one needs to understand its theoretical properties in the setting where n and T are comparable in some sense. I would look into the recent literature on batched bandits (for example, Perchet. et. al, reference [21] in the paper) to find out the correct range of n and T, for which the optimal regret is achieved and see what can be said in such settings. My questions: - Is it true that the result of BOLS is not limited to data collected by a bandit policy and it applies to any non-anticipating policy? - Is it possible to combine OLS and W-decorrelated estimators? For example, use OLS within each batch and then combine the results using W-decorrelated estimators? This way, you can probably address the case of having comparable T and n. --------------------- After Rebuttal: I am satisfied with the authors response and have adjusted my score accordingly.


Review 2

Summary and Contributions: A central challenge in the use of multiarmed bandits for experimentation (whether in clinical trials, or in online experimentation) is that valid statistical inference is in general not feasible with adaptive allocation, without a great deal of additional structure. In the present paper, this structure is introduced through batching. The main result of the paper shows that if one runs a batched bandit, then OLS estimation from the bandit data is asymptotically normal, and thus usual normal confidence intervals can be constructed with valid coverage guarantees. The paper opens by first establishing that in the case of a two armed bandit with no gap, the OLS estimator is asymptotically non-normal. This result has more consequences than one would immediately think: of course, in practice, gaps are never zero so in that sense this is an edge case. However, as the authors point out, asymptotic non-normality at zero is indicative of poor convergence of the estimator, and in particular indicates that coverage and size guarantees will not hold at finite samples.

Strengths: Overall I really enjoyed this paper. The contributions are clear and valuable to the experimentation community. The proposed method is simple and easily implementable, not to mention interpretable, all of which are real virtues. I want to emphasize one nice thing about the technical argument. Even with batching, there is a challenge because allocation is adaptive across batches. The key trick is to note that conditional on the last batch, the probability of pulling an arm becomes a constant. This allows them to work out a CLT using martingale arguments across batches. I found this to be a very elegant application of martingale CLT techniques to obtain the desired result.

Weaknesses: The authors point out that in their empirical results that (in stationary settings) power can be potentially worse than other methods that don't have coverage guarantees. This highlights the fact that the paper doesn't discuss power very much. Relatedly, I would have liked to see more discussion of batch size optimization. The results are asymptotic and so they don't focus much on this, but presumably batch size interacts with power in a fundamental way, and a lack of exploration of this nexus seemed like a missed opportunity.

Correctness: As with most NeurIPS papers most of the technical discussion was deferred to the appendix/supplementary material, which I did not check in detail.

Clarity: Yes, the paper very clearly explains its goals and results. The one thing I think they would do well to emphasize is to move up the discussion in italics at line 191, as otherwise readers may not appreciate the importance of their analysis of the zero margin case.

Relation to Prior Work: One thing I found unusual is that there was not much discussion of bias in estimation via MABs. For example, there is this recent paper: https://arxiv.org/pdf/1708.01977.pdf I would like to understand how the results of that paper interact with the current paper. On finite horizons, it seems that bias will be an issue as well as coverage. How should the experimenter deal with that issue?

Reproducibility: Yes

Additional Feedback: I had a question on your discussion of clipping around line 138: "We allow the action selection probabilities π(n) to converge to 0 and/or 1 at some rate." In your results, it seems like you only use constant clipping functions, so that action selection probabilities can't go to zero. Do the results only apply if forced exploration is required for all time, or have I missed something in the formulation of clipping? EDITED AFTER REBUTTAL: I agree with some of the comments raised regarding related work on confidence sequences, but also appreciated the authors' feedback on this point. I left my score unchanged after rebuttal.


Review 3

Summary and Contributions: This paper the problem of statistical inference from data collected in *batched bandit experiments* (BBE), namely assigning confidence intervals and computing p-values from such data. They consider the relatively simple setting where there are bounded batches or 'rounds of adaptivity', and where the number of data points within each batch diverges. Within such a setting, they are able to address mild nonstationarity in the experiment environment. The main contributions of the current paper are: 1. They establish that standard least squares (LS) estimators are asymptotically normal or non-normal in BBE depending on the underlying arm mean rewards, specifically whether problem admits a unique optimal arm. In the more general context of stochastic regression, LW proved that if the data admits 'stability', then the standard least squares (LS) estimator is asymptotically normal. This work shows that stability fails in batched bandit experiments when the margin vanishes. 2. Taking advantage of the limited adaptivity, the authors propose a 'batched LS' (BLS) that computes the least squares estimates restricted to a single batch. They then use these estimates to compute Z statistics in a natural way. 3. The authors propose a natural BLS variant for batched linear contextual bandit problems and construct valid statistics analogous to point 2. Reference key LW: Lai and Wei LPP: Liu, Proschan, Pledger DMST: Deshpande, Mackey, Syrgkanis, Taddy DJM: Deshpande, Javanmard, Mehrabi

Strengths: Multi-armed bandits are well-known framework for sequential decision making and the corresponding algorithms developed for these problems are widely used in practice. One of the current paper's strengths (building on LW and DMST) is to consider frequentist statistical inference within this framework, as a problem distinct from bandit algorithm design. This is of significant interest to the NeurIPS community, particularly for reinforcement learning, policy and off-policy evaluation. Probably the strongest contribution of the paper is contribution 1, where the normal convergence of standard estimators is rigorously shown to depend on the problem environment, specifically the margin. Thanks to the limited rounds of adaptivity, or the batched settings, the proofs for this use straightforward (and easy to anticipate) arguments.

Weaknesses: The following strike me as weaknesses. - The authors consider limited (indeed bounded) rounds of adaptivity, unlike LW and DMST. While this allows for a clear first contribution, it limits the algorithmic contributions signficantly. It would be good to acknowledge this tradeoff at the outset, while pointing that earlier work (like references LPP, DJM) also use this simplification - The BLS estimator seems closely related to the batched online debiasing estimator of DJM minus l1 constraints; it would be good to point this. - I think the non-stationarity is a little artificial, especially if it perfectly aligns with the batches. In practical settings, one can perhaps expect slowly varying non-stationarity which can yield close, but possibly still imperfect alignment.

Correctness: The papers proofs are correct and easy to follow.

Clarity: The main portion of the paper is well-written, clear and easy to follow. The proofs in the appendix can make somewhat cleaner use of notation (particularly matrices) to make the arguments shorter and more natural.

Relation to Prior Work: The work references related work and discusses it appropriately in the context of the paper.

Reproducibility: Yes

Additional Feedback: Minor comments: - L.48 probably should be 'batched bandit data' - L.72 What is adaptive control? Does it mean the statistician has control of the algorithm, hence length of the batches? - Is the countability in LPP a real thing? In practice Thompson sampling is carried out using rationals on a computer, hence is countable (indeed finite). - Figure 2 is nice... .what is n here? Also useful to point out that miscoverage in the tails can be asymmetric in general. - Why are pi_min and pi_max ordered that way? Is it necessary, an artifact of the proof, or saying something real? - L.238 weird sentence 'for have non-stationary over time' -For the batch sizes (n) in simulations, does the t vs normal approximation matter significantly? In some simulations in appendix, n=25 so probably this is true. I am not sure the n is so small throughout. - L. 450, Eq (5), (6) in appendix. Possible typos in feature matrix EDIT POST RESPONSE: After reading the author response, I am maintaining my score on the paper.


Review 4

Summary and Contributions: This paper proposes new methods for inference on data collected in batches using a bandit algorithm. In theory, they first prove that OLS is not asymptotically normal and then prove the asymptotic normality of the proposed BOLS.

Strengths: 1. The considered inference problem is new and interesting in bandit algorithms. 2. The proposed method is simple to implement and theoretical study is sound.

Weaknesses: 1. Although this paper claims to consider a bandit setting with adaptive collected data, this paper in fact assumes that the actions within each batch are conditionally i.i.d, and the rewards in each batch are independent. In the theory, the sample size $n$ within each batch needs to go to infinity. That is to say, we have access to infinity i.i.d. samples within each batch. I am not fully convinced which practical problem satisfies this condition. More justification of this assumption would be helpful. 2. This paper uses an online education experiment as an example to motivate the importance of inference in bandit algorithms, see lines 19-26. The motivation is that ``to help others designing online courses we would like to be able to publish our findings about how different teaching strategies compare in their performance. Though bandit algorithms provide regret guarantees on a particular problem instance (e.g., a particular online education experiment), they alone do not provide any way for practitioners to draw generalizable knowledge from the data they have collected (e.g., how much better one teaching strategy is compared to another) for the sake of scientific discovery and informed decision making." I am not fully convinced why regret comparison between two online education methods is not sufficient to tell how much better one teaching strategy is compared to another. More justification on why inference is necessary in practical bandit algorithms would be helpful. ~~~~~~~~~~~ After rebuttal: My concern on the practical applications of the batch setting has been addressed. Since the considered statistical inference problem on bandit problems is interesting and the non-normality when $\Delta=0$ is new, I would like to increase my rating to 6.

Correctness: Yes

Clarity: Yes

Relation to Prior Work: Yes

Reproducibility: Yes

Additional Feedback:


Review 5

Summary and Contributions: N/A

Strengths: N/A

Weaknesses: N/A

Correctness: N/A

Clarity: N/A

Relation to Prior Work: The author response was satisfactory and I have changed my score to reflect this. ----- I was invited as a last minute review to comment solely on the relationship to existing prior work, so I will stick to that below. The authors miss a significant part of the literature using anytime confidence bounds to provide inference. They do not cite any papers, and mistakenly claim in one line that "Note that best arm identification is distinct from obtaining a confidence interval for the margin, as the former identifies the best arm with high probability (assuming there is a best arm), while the latter provides guarantees regarding the sign and magnitude of the margin". However, it is quite transparent that the uniform bounds used within these algorithms (like lil'UCB and followups/variants) can indeed be used to provide statistically correct guarantees on the margin, since they are simultaneously valid over all arms (including data-dependent arm indices like the best one) and simultaneously valid all times (including stopping times). For example, 0) https://arxiv.org/pdf/1312.7308.pdf (COLT'14) is the first one to use anytime confidence bounds for bandits 1) https://arxiv.org/pdf/1706.05378.pdf (NeurIPS'17) looks at inference on differences between means 2) https://arxiv.org/abs/1811.11419 (under review) also studies the "contrast" problem 3) Lemma 3 in https://arxiv.org/abs/1810.08240 (Annals of Statistics, 2020+) points out that if inference at arbitrary stopping times is desired, then there is no other option but to produce uniform bounds. (There are many other papers by the above authors and others, these are a sample.) Even though the current submission has some nice new contributions, they seem oblivious (and dismissive) of a huge swath of the literature. This certainly needs to be corrected in order to not mislead readers or oversell results. I would change my score if the authors commit to explicitly discussing the above related work, including clearly acknowledging the fact that these are the only known (and only possible) ways to provide valid inference for data-dependent arms at arbitrary data-dependent stopping times. Indeed, one may argue that in bandit algorithms when the difference between arms is apriori unknown, inference at stopping times is particularly critical --- we may stop sooner for "easier" problems (clearly one dominant arm) and later for "harder" problems (problems in which there is 0, or close to 0, difference between the arms).

Reproducibility: Yes

Additional Feedback: N/A

[Author Response · NeurIPS 2020]

**Motivation for batched setting** A common critique from the reviewers is about the relevance of the batched bandit setting we consider. Our most important point on this is that the batched setting is very common in applications, e.g., in many mobile health [4-6] and online education problems [7,8] multiple users use apps / take courses simultaneously, so a batch corresponds to the number of unique users the bandit algorithm acts on at once. The batched setting is even common in online advertising because it is impractical to update the bandit after every action if many users visit the site simultaneously [1-3]. We provide a highly-abridged list of papers on real experiments run in a batched setting:

[1] doi:10.1145/2645710.2645732 (RecSys'14)        [5] doi:10.1145/3381007 (ACM'20)
[2] doi:10.1145/2505515.2514700 (CIKM'13)          [6] doi:10.1093/abm/kay067 (Ann Behav Med'19)
[3] doi:10.1287/mksc.2016.1023 (Mark Sci'17)       [7] doi:10.5281/zenodo.3554749 (JEDM'19)
[4] doi:10.2196/jmir.7994 (JMIR'17)                [8] doi:10.1073/pnas.1921417117 (PNAS'20)

In many such experimental settings $T$ cannot be arbitrarily adjusted, e.g., in online education, courses generally cannot be made arbitrarily long, and clinical trials often run for a standard amount of time that depends on the domain science (e.g. the length of mobile health studies is a function of the scientific community's belief in how long it should take for users to form a habit). On the other hand, the number of users can in principle grow as large as funding allows, and thus for analyzing data from an experiment it is quite standard to consider asymptotics as the number of users grows.

Non-stationarity is very common in many of the problems settings [1,5,7]. In response to reviewer 3's comments, the non-stationarity is aligned with batches since batches correspond to actions selected in the same time period.

**Comments specific to reviewer 5** We thank you for your constructive comments and references; we apologize for the omission and will certainly correct it in the revision. In particular, we will not claim a distinction between best arm identification and obtaining a confidence interval for the margin, and will instead explain how the two are very much connected, especially since the intervals obtained from the uniform bounds from the best arm identification literature are particularly relevant when the experiment ends at a data-dependent time. We will reference the main works in this field (and we thank you for giving us a starter list to build on) and also include a comparison in our simulation section. We were able to obtain one preliminary comparison in time for this rebuttal: in the stationary setting under Thompson Sampling ($n = 25$, $T = 25$, 0.1 clipping), the 95% confidence interval using BOLS has average width $0.405$, while confidence intervals constructed with the anytime self-normalized martingale bound used in the regret bounds of many bandit algorithms (doi:10.5555/2986459.2986717) are more than double the width at $0.943$.

**Comments specific to reviewer 1** Regarding your comment on $n$ and $T$ needing to be comparable, we can interpret this in two ways—we will address both. First, to use BOLS for approximate inference, $n$ and $T$ can be comparable—in fact, in our experiments we often set $n = 25$ and $T = 25$; however $n$ must be sufficiently large for a reliable asymptotic approximation. The second interpretation is that $n$ and $T$ must be comparable in order to run a bandit algorithm that minimizes regret. It is correct that the batched bandit algorithms we consider cannot be optimal with respect to the standard oracle (non-batched, stationary environment). However, given the realities of the problem settings bandits are used for, it can make sense to consider other oracles. Perechet et al. consider the oracle to be the best algorithm when the number of batches is limited and the batch sizes can be adjusted adaptively. For the problems we consider it is not realistic to adjust the batch sizes adaptively, e.g., if there are a fixed number of users in the study across all time points, or a certain number of users who visit a website in some time period. In our problem setting, it makes sense to choose an oracle that is optimal in the batched setting in which the batch sizes cannot be adjusted adaptively.

**Comments specific to reviewer 4** To address your first concern, we now discuss why the adaptivity in the batched setting doesn't disappear even as $n \to \infty$. As we proved, when $\Delta = 0$, under common bandit algorithms the $\pi_t$ does not concentrate and will always depends on whether we happened to receive greater rewards on average from arm 0 or arm 1 in the previous batches. The asymptotic dependence between the $\pi_t$ and the rewards of the previous batch is what leads the OLS estimator to be asymptotically non-normal when $\Delta = 0$. Further, as we illustrate in Figure 2, when $\Delta$ is near-zero the distribution of the OLS estimator is not well approximated by a normal distribution.

To address your second concern, of course we cannot calculate the regret, since the optimal arm is unknown in real life experiments. One could propose to compare the total sum of rewards for each arm after the study is over. However, since the action selection probabilities do not concentrate when $\Delta = 0$, this means that even if the bandit happens to select one arm much more often (thus greater sum of rewards for that arm), it doesn't mean that arm is necessarily better than the other arm. For example, for $\epsilon$-greedy, if $\Delta = 0$, across many identical experiments with different random seeds, half of the experiments will choose arm 1 more often and half the experiments will choose arm 0 more often. If we look at the difference in the average reward between arms then this is exactly the OLS estimator for the margin. We proved that this OLS estimator is asymptotically non-normal when $\Delta = 0$, so we cannot use a normal approximation to obtain valid confidence intervals for this estimator.

**Comments specific to reviewer 2** (a) We agree that investigating the power is interesting, but beyond the scope of this paper. As $\pi_t$ gets closer to 0.5, the power increases, but the regret increases; trading these two objectives off is non-trivial. (b) We will mention the issue of bias in the revision. (c) We allow the clipping rate to decay in Theorem 3.

[Meta-Review · NeurIPS 2020]

The author rebuttal was deemed satisfactory, and most reviewers tended to the positive side. On my personal reading of the paper, I concur with them and recommend acceptance. In particular, one of the reviewers summarized it fairly well in the discussion: "I liked the result on the non-normality when the margin vanishes (Theorem 2), but I am still not positive about the theoretical contributions related to BOLS (Theorems 3 and 4); the limited adaptivity simplifies the theoretical analysis greatly". Another reviewer opined "Given the batched setting, with limited adaptivity, it is not difficult to anticipate the algorithmic contributions of the paper. Nevertheless I think the paper is a good contribution to NeurIPS; the proof of non-normality when margin vanishes is interesting and new. The addition of uniform bounds and the Nie et al paper should round it well". The last reviewer added "My point about related work was addressed reasonably, though I think their experiment may have been a strawman. It is obvious that when CLT bounds apply, they are tighter than time-uniform bounds, the point is that the former do not apply for adaptively stopped experiments, which is half the point of sequential analysis and bandits". Overall the reviewers were very knowledgeable and thoughtful and I concur with all their points of view. The authors would benefit from taking all of their opinions seriously, the community would benefit if the reviewers' opinions were accurately reflected in the final version. In particular, I would request to properly cite the relationship of this works to the various suggested papers, for example on the one on bias in bandits (Nie et al, Shin et al) and anytime-valid confidence bounds mentioned by the last reviewer.